# The little skate genome and the evolutionary emergence of wing-like fins

Ferdinand Marlétaz[1,2,24 ✉], Elisa de la Calle-Mustienes[3,24], Rafael D. Acemel[3,4,24], Christina Paliou[3,24], Silvia Naranjo[3], Pedro Manuel Martínez-García[3], Ildefonso Cases[3], Victoria A. Sleight[5,6], Christine Hirschberger[5], Marina Marcet-Houben[7,8], Dina Navon[9], Ali Andrescavage[9], Ksenia Skvortsova[10,11], Paul Edward Duckett[10], Álvaro González-Rajal[10,11], Ozren Bogdanovic[10,12], Johan H. Gibcus[13], Liyan Yang[13], Lourdes Gallardo-Fuentes[3], Ismael Sospedra[3], Javier Lopez-Rios[3], Fabrice Darbellay[14,23], Axel Visel[14,15,16], Job Dekker[13,17], Neil Shubin[18], Toni Gabaldón[7,8,19,20], Tetsuya Nakamura[9,25 ✉], Juan J. Tena[3,25 ✉], Darío G. Lupiáñez[4,25 ✉], Daniel S. Rokhsar[2,21,22,25 ✉] & José Luis Gómez-Skarmeta[3,25,26]

Skates are cartilaginous fish whose body plan features enlarged wing-like pectoral fins, enabling them to thrive in benthic environments[1,2]. However, the molecular underpinnings of this unique trait remain unclear. Here we investigate the origin of this phenotypic innovation by developing the little skate *Leucoraja erinacea* as a genomically enabled model. Analysis of a high-quality chromosome-scale genome sequence for the little skate shows that it preserves many ancestral jawed vertebrate features compared with other sequenced genomes, including numerous ancient microchromosomes. Combining genome comparisons with extensive regulatory datasets in developing fins—including gene expression, chromatin occupancy and three-dimensional conformation—we find skate-specific genomic rearrangements that alter the three-dimensional regulatory landscape of genes that are involved in the planar cell polarity pathway. Functional inhibition of planar cell polarity signalling resulted in a reduction in anterior fin size, confirming that this pathway is a major contributor to batoid fin morphology. We also identified a fin-specific enhancer that interacts with several *hoxa* genes, consistent with the redeployment of *hox* gene expression in anterior pectoral fins, and confirmed its potential to activate transcription in the anterior fin using zebrafish reporter assays. Our findings underscore the central role of genome reorganization and regulatory variation in the evolution of phenotypes, shedding light on the molecular origin of an enigmatic trait.

The origin and diversification of vertebrates was accompanied by the appearance of key developmental innovations[2,3]. Among them, paired appendages show an exquisite diversity of forms and adaptations not only in tetrapods, but also in chondrichthyans (cartilaginous fish) in which fin structures are considerably diverse[2]. The wing-like appendages of batoid fishes (skates and rays) (Fig. 1a) are fascinating examples, in which the pectoral fins extend anteriorly and fuse with the head.

This unique structure creates power for forward propulsion and led to the emergence of swimming mechanisms that enabled skates to colonize the sea floor[1]. Transcriptomic analysis of skate developing fins revealed a major reorganization of signalling gradients relative to other vertebrates[1]. The redeployment of developmental transcription factors, such as 3′ *hox* genes, initiates an anterior signalling centre analogous to the posterior apical ectodermal ridge (AER). These changes

[1]Centre for Life's Origin and Evolution, Department of Genetics, Evolution and Environment, University College London, London, UK. [2]Molecular Genetics Unit, Okinawa Institute of Science and Technology Graduate University, Onna, Japan. [3]Centro Andaluz de Biología del Desarrollo (CABD), Consejo Superior de Investigaciones Científicas/Universidad Pablo de Olavide/Junta de Andalucía, Seville, Spain. [4]Epigenetics and Sex Development Group, Max Delbrück Center for Molecular Medicine in the Helmholtz Association (MDC), Berlin Institute for Medical Systems Biology (BIMSB), Berlin, Germany. [5]Department of Zoology, University of Cambridge, Cambridge, UK. [6]School of Biological Sciences, University of Aberdeen, Aberdeen, UK. [7]Barcelona Supercomputing Centre (BCS-CNS), Barcelona, Spain. [8]Institute for Research in Biomedicine (IRB Barcelona), The Barcelona Institute of Science and Technology, Barcelona, Spain. [9]Department of Genetics, Rutgers the State University of New Jersey, Piscataway, NJ, USA. [10]Genomics and Epigenetics Division, Garvan Institute of Medical Research, Sydney, New South Wales, Australia. [11]Faculty of Medicine, St Vincent's Clinical School, University of New South Wales, Sydney, New South Wales, Australia. [12]School of Biotechnology and Biomolecular Sciences, University of New South Wales, Sydney, New South Wales, Australia. [13]Department of Systems Biology, University of Massachusetts Chan Medical School, Worcester, MA, USA. [14]Environmental Genomics and Systems Biology Division, Lawrence Berkeley National Laboratory, Berkeley, CA, USA. [15]US Department of Energy Joint Genome Institute, Berkeley, CA, USA. [16]School of Natural Sciences, University of California, Merced, CA, USA. [17]Howard Hughes Medical Institute, Chevy Chase, MD, USA. [18]Department of Organismal Biology and Anatomy, University of Chicago, Chicago, IL, USA. [19]Catalan Institution for Research and Advanced Studies (ICREA), Barcelona, Spain. [20]CIBER de Enfermedades Infecciosas, Instituto de Salud Carlos III, Madrid, Spain. [21]Department of Molecular and Cell Biology, University of California, Berkeley, CA, USA. [22]Chan-Zuckerberg Biohub, San Francisco, CA, USA. [23]Present address: Department of Genetic Medicine and Development, Faculty of Medicine, University of Geneva, Geneva, Switzerland. [24]These authors contributed equally: Ferdinand Marlétaz, Elisa de la Calle-Mustienes, Rafael D. Acemel, Christina Paliou. [25]These authors jointly supervised this work: Tetsuya Nakamura, Juan J. Tena, Darío G. Lupiáñez, Daniel S. Rokhsar, José Luis Gómez-Skarmeta. [26]Deceased: José Luis Gómez-Skarmeta. ✉e-mail: ferdinand.marletaz@gmail.com; nakamura@dls.rutgers.edu; jjtenagu@upo.es; Dario.Lupianez@mdc-berlin.de; dsrokhsar@gmail.com

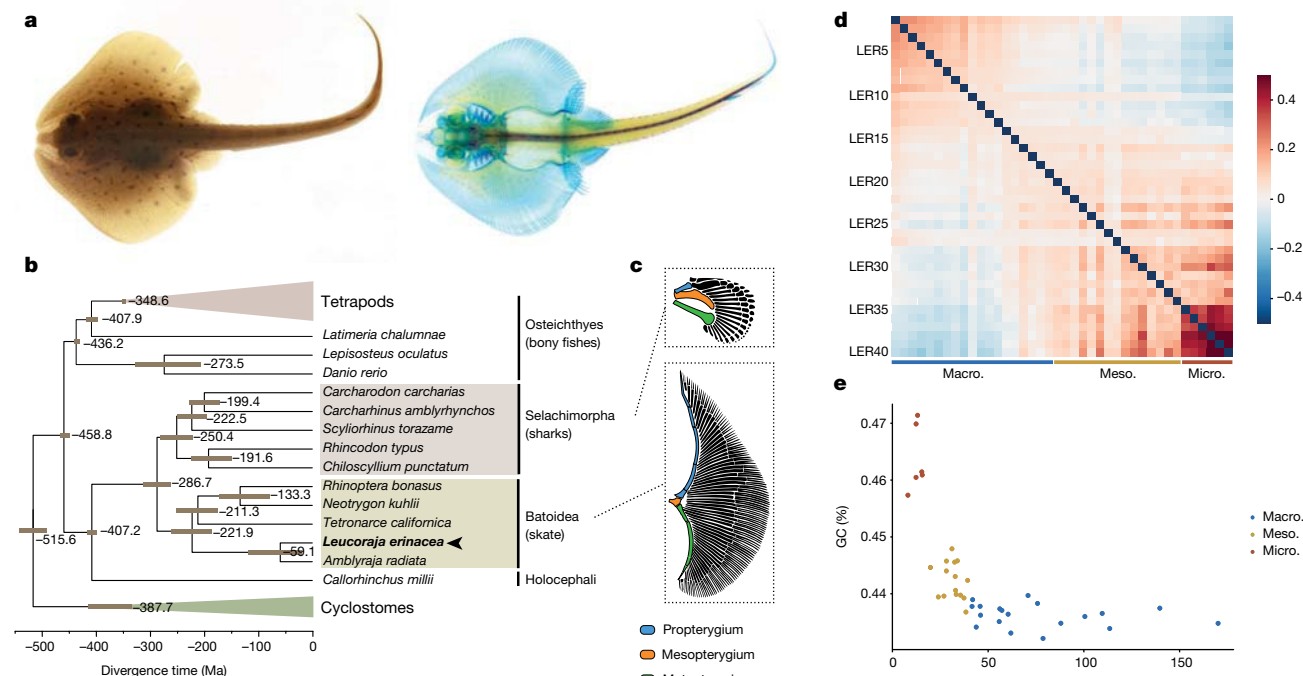

**Fig. 1 | The little skate morphology and genome evolution. a**, Adult little skate (*L. erinacea*) and skeletal staining using Alcian Blue and Alizarin Red. **b**, Chronogram showing the branching and divergence time of chondrichthyan and selected osteichthyan lineages (Supplementary Fig. 1). **c**, Morphological differences in the skeleton between the pectoral fins in shark and skate highlighting the expansion of a wing-like fin. The illustrations were reproduced from a previous publication[60]. **d**, Pairwise Hi-C contact density between 40 skate chromosomes, showing an increased interchromosomal interaction between the smallest ones (microchromosomes). The colour scale shows log-transformed observed/expected interchromosomal Hi-C contacts. Macro., macrochromosome; meso., mesochromosome; micro., microchromosome. **e**, Little skate chromosome classification based on the relationship between their size and GC percentage, highlighting the high GC content of microchromosomes.

arose ~286–221 million years ago (Fig. 1b) after the divergence between sharks and skates. Nevertheless, the genomic and regulatory changes underlying these novel expression domains have remained elusive.

Many vertebrate evolutionary innovations were influenced by the substantial genomic reorganizations caused by two rounds of whole-genome duplication (WGD). The ancestral chordate chromosomes were duplicated and rearranged to give rise to the diversity of existing karyotypes in vertebrates[4]. Concomitantly, the pervasive loss of paralogous genes after WGDs produced gene deserts enriched in regulatory elements[5]. Compellingly, those genomic alterations were paralleled by marked changes in gene regulation, contributing to an increase in pleiotropy in developmental genes[5] and to the complexity of their regulatory landscapes[6]. In vertebrates, regulatory landscapes are spatially organized into topologically associating domains (TADs)[7,8]. TADs correspond to large genomic regions with increased self-contact that promote the interaction between *cis*-regulatory elements (CREs) and cognate promoters to constitute precise transcriptional patterns. While TADs constrain the evolution of gene order[9], genomic rearrangements that alter these domains can be a source for developmental phenotypes[10] and evolutionary innovation[11,12]. Yet the importance of TAD organization for the evolution of gene regulation and the emergence of lineage-specific traits after vertebrate WGDs remains largely unexplored.

To gain insights into the evolution of the jawed vertebrate (gnathostome) karyotypes and of wing-like appendages, we generated a chromosome-scale assembly of the little skate *L. erinacea* and performed extensive functional characterization of its developing fins. Our analyses revealed a karyotype configuration resembling the gnathostome ancestor, characterized by slower paralogue loss and smaller chromosomes than other jawed vertebrates, which suggests fewer fusion events after the second round (2R) of WGD in the skate lineage.

We find evidence that three-dimensional (3D) genome organization in skate arises from an interplay between transcription-based A/B compartments and TADs formed by loop extrusion, as described in mammals[13]. The comparison of the 3D organization of α and β chromosomes after the gnathostome-specific WGD revealed a prominent loss of complete TADs, probably contributing to karyotype stabilization. By combining RNA sequencing (RNA-seq) and assay for transposase-accessible chromatin with sequencing (ATAC–seq) data, we identified the planar cell polarity (PCP) pathway and *hox* gene regulation as key contributors to skate fin morphology, which we further validated using functional assays in zebrafish and skate. Our study illustrates how comparative multi-omics approaches can be effectively used to elucidate the molecular underpinnings of evolutionary traits.

## Genome sequencing and comparative genomics

We assembled the little skate genome at the chromosome scale by integrating long- and short-read genome sequencing with chromatin conformation capture (Hi-C) data. Our assembly includes 40 chromosome-scale (>2.5 Mb) scaffolds, with 19 macrochromosomes (>40 Mb), 14 mesochromosomes (between 20 and 40 Mb) and 7 microchromosomes (<20 Mb) that together represent 91.7% of the 2.2 Gb assembly. This chromosome number is within the range reported for other Rajidae species[14]. Despite technical challenges due to high polymorphism levels (1.6% heterozygosity) and a repeat content dominated by recently expanded LINE retrotransposons (Extended Data Fig. 1), our assembly showed a similar or higher degree of completeness with respect to gene content compared to other sequenced chondrichthyans (BUSCO; Supplementary Table 1).

We annotated 26,715 protein-coding genes using extensive transcriptome resources[15], with 23,870 possessing homologues in other species.

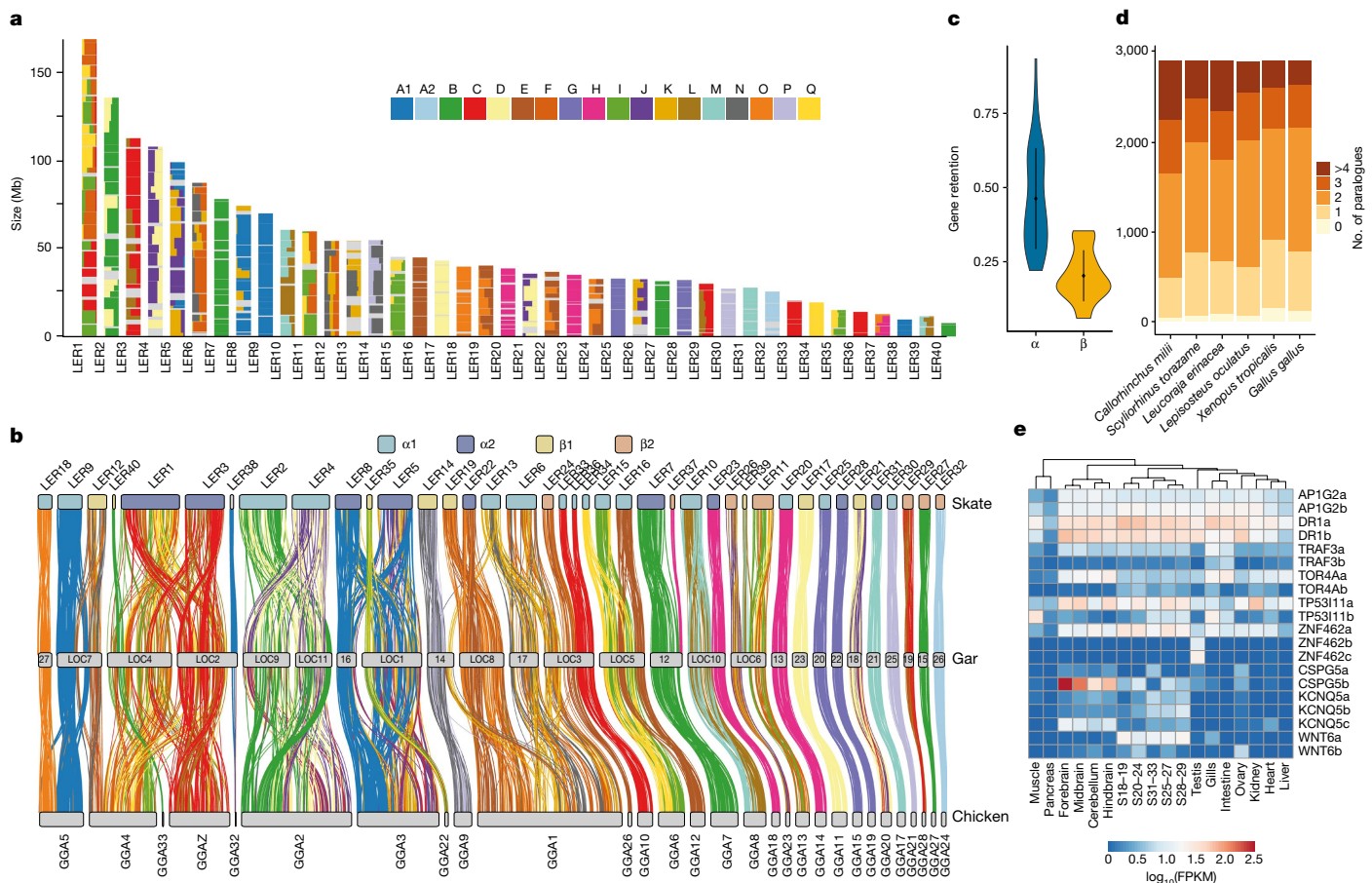

**Fig. 2 | Ancestral linkage and the architecture of early vertebrate genomes.**
**a**, The fraction of genes derived from each CLG (depicted as squares named A1–Q) in skate chromosomes represented for bins of 20 genes. **b**, The syntenic orthology relationship between skate, gar and chicken, relying on genes with a significant CLG assignment in regard to amphioxus. Skate chromosomes are coloured by segmental identity and links are coloured by CLG. **c**, Rates of gene retention for α or β segments derived from the second alloploid event of vertebrate WGD. **d**, Respective gene-family composition for ohnologues in selected jawed vertebrate species indicating differential paralogue loss. **e**, Gene expression for selected sets of differentially lost ohnologues for which a copy was lost in the gnathostome but not in the chondrichthyan lineage. FPKM, fragments per kilobase of transcript per million mapped reads; S, stage.

Using comparative analysis with 20 other sequenced vertebrates we reconstructed the complete set of skate gene evolutionary histories (the phylome) and used it to infer patterns of gene duplication and loss, as well as orthology and paralogy relationships (Supplementary Table 2; resources are available at PhylomeDB and MetaPhoRs[16,17]). We used phylogenomic methods to reconstruct jawed vertebrate phylogeny and infer divergence times, finding a more ancient divergence between sharks and skates (around 286 million years ago) than previously estimated[18] (Fig. 1b). Compared with other reported chondrichthyan genomes, *L. erinacea* displays the lowest number of species-specific gene losses (616 losses; Supplementary Fig. 1). Similar to sharks (selachians)[19,20], the little skate has larger introns than tetrapods (median size, 2,167 bp versus 1,586 bp in human), although these are not enriched in a particular repeat category (Extended Data Fig. 1).

Skate microchromosomes have an overall higher gene density compared with macro-chromosomes (Extended Data Fig. 1a–c,g), suggesting that, as in birds, these small chromosomes are prone to GC-biased gene conversion[21]. Skate microchromosomes also show a higher degree of interchromosomal contacts compared with other chromosomes (Fig. 1d,e), as also found in snakes and other tetrapods[22].

## Chromosome evolution

We surveyed the arrangement of syntenic chromosomal segments derived from ancestral chordate linkage groups (CLGs) in skate, gar and chicken, using amphioxus as an unduplicated outgroup[23], and found that the chromosomal organization of the skate genome closely resembles that of the most recent jawed vertebrate common ancestor (Fig. 2a and Extended Data Fig. 2). By analysing the chromosomal locations of single-copy orthologues, we designated chromosomal segments according to their origin at 1R (1 or 2) or 2R (α or β) vertebrate WGDs[23] (Fig. 2b). The relatively large number of elasmobranch chromosomes (≥40) reflects the ancestral condition among gnathostomes; with the exception of the losses of two ancestral segments in the skate lineages, and one secondary fusion on chromosome 1, the skate possessed 37 out of the 39 ancestral vertebrate linkages (Supplementary Table 3). The evolution of reduced chromosome number in osteichthyan (bony fishes) lineages is therefore due to subsequent chromosomal fusions.

The smaller vertebrate chromosomes often show a reciprocal correspondence across species and correspond to a single ancestral gnathostome unit[23–25] (10 chromosomes have a 1:1:1 orthology between skate, gar and chicken; Fig. 2b). The trios LER25≡LOC20≡GGA15 and LER28≡LOC22≡GGA19 represent two surviving copies of CLG-G from the 1R event. Other trios such as LER21≡LOC18≡GGA20 and LER29≡LOC19≡GGA28 derive from CLG fusions, and the occurrence of some in all gnathostome genomes implies that they happened between the pan-vertebrate 1R and the gnathostome-specific 2R[23,25] (Fig. 2b).

In many gnathostomes, larger chromosomes also derive from fusions of CLGs. The skate often represents an ancestral state among jawed vertebrates, with subsequent fusions in bony fishes, including in

chicken (for example, GGA5), in gar (for example, LOC5) or in their common ancestor (for example, LER 2 and 4; see below). For example, ancestral gnathostome chromosomes resembling skate LER9, LER12 and LER18 fused in different ways to form chromosomes in gar and chicken. Similarly, LER10≡GGA8 and LER23≡GGA18 (≡BFL8) probably represent ancestral units that fused in gar chromosome LOC10 through a centric Robertsonian fusion (Fig. 2b). Notably, these two chromosomes are also preserved in their ancestral condition in the bowfin, the sister group of gar, implying that fusion occurred specifically in this lineage[26].

Alternatively, ancestral chromosomes resembling LER2 and LER4 probably fused in the bony fish ancestor to give rise to chicken GGA2, whereas gar LOC9 and LOC11 are secondarily split from this fused ancestral chromosome. This may have involved a Robertsonian fission that split a metacentric chromosome at the centromere into two acro- or telocentric products. We also observe cases in which microchromosomes have been added to macro-chromosomes recently by terminal translocation, such as the addition of a chromosome similar to LER35≡GGA22 to the start of LOC1, or a LER12-like chromosome to the end of GGA4 (a recent translocation not found in other birds)[27].

The extensive conservation of chromosomal identity and gene order between the little skate and the bamboo shark[28], despite over 300 million years of divergence, indicates that most chondrichthyans may share this ancestral chromosomal organization (Fig. 1b,c and Extended Data Fig. 2). Notably, gene order collinearity across cartilaginous fish is more extensively conserved than within clades of comparable divergence, such as mammals and frogs[29]. By contrast, gene order is heavily disrupted between chondrichthyans (such as skate or shark) and osteichthyans (gar or chicken; Fig. 2a,b and Supplementary Fig. 2).

## Evolution of the gene complement

The gene complement of the little skate, as in other chondrichthyans, evolved slower than that of Osteichthyes with respect to gene loss (Supplementary Fig. 1). Using species-tree-aware phylogenetic methods, we found that the retention of ohnologues (paralogues derived from vertebrate-specific WGDs) was higher than that observed in bony fishes (Fig. 2c,d and Extended Data Fig. 1h). According to the auto-then-allotetraploidy scenario for jawed vertebrate evolution[23], the chromosomes derived from 2R behave distinctly, with beta segments showing increased loss and higher rates of molecular evolution (Fig. 2c,e and Extended Data Fig. 1i).

On the basis of patterns of duplication and loss, we found 68 cases in which one ohnologue was differentially retained in varying jawed vertebrate lineages, 19 genes retained in chondrichthyans but lost in bony fishes, 17 retained in chondrichthyans and coelacanth, and 24 retained in chondrichthyans and actinopterygians (ray-finned fish) but lost in lobe-finned fish (Supplementary Table 3). Some of these retained ancestral ohnologues, including previously characterized genes such as *wnt6b*[20] or novel genes such as chondroitin sulfate proteoglycan 5 (*cspg5*), show distinct expression patterns among stages and organs (Fig. 2e).

## Conservation of 3D regulatory principles

We investigated 3D chromatin organization in skates using Hi-C analysis of developing pectoral fins. We found a type II architecture[30] with chromosomes preferentially occupying individual territories within the nucleus (Supplementary Fig. 3), consistent with a complete set of condensin II subunits (*smc2*, *smc4*, *caph2*, *capg2* and *capd3*) in the genome. At higher resolution, skate chromosomes are organized into two distinct compartments, as described in other animals[31]. The A compartment displays higher gene density, chromatin accessibility and gene expression levels compared with the B compartment (Extended Data Fig. 3).

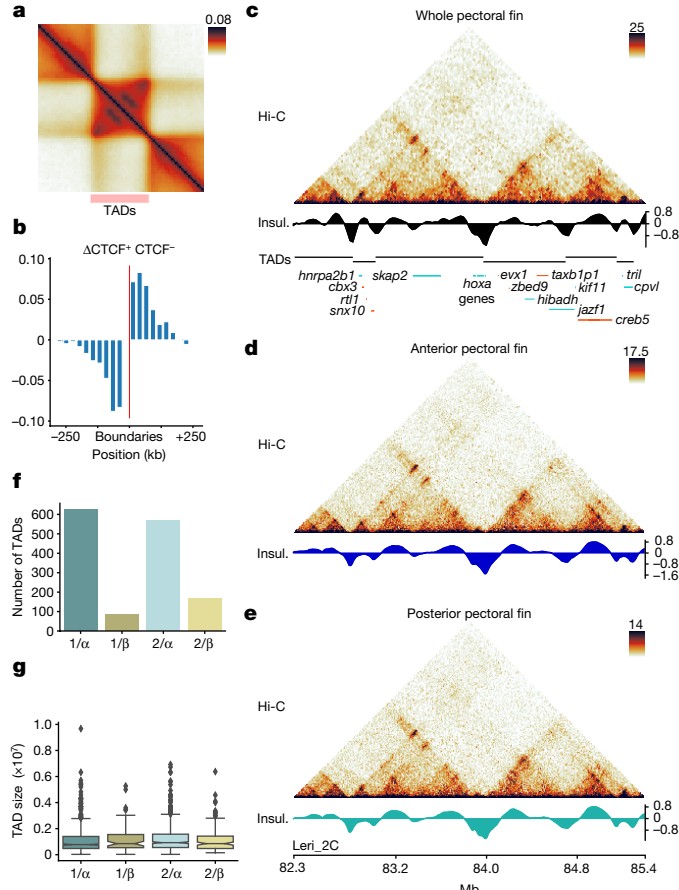

**Fig. 3 | Features of 3D chromatin organization in the little skate. a**, TAD metaplot displaying focal interactions at the apex of domains. **b**, Orientation bias of CTCF-binding site motifs inside ATAC–seq peaks at TAD boundary regions. **c**, Hi-C maps from whole pectoral fins of the skate *hoxa* locus at 25 kb resolution, denoting the presence of bipartite TAD configuration. Insul., insulation score. **d**,**e**, Hi-C maps from the same locus of **c** from dissected anterior (**d**) and posterior (**e**) portions of skate pectoral fins at 10 kb resolution. No changes in TADs or looping patterns were observed. **f**, The number of TADs detected associated to the different paralogous segments descending from the two rounds of WGD (1 or 2 for the 1R; α or β for the 2R) **g**, TAD sizes observed in the different paralogous segments from **f**. The box plots show the median (centre line) and the first and third quartiles (Q1 and Q3; box limits), and the whiskers extend to the last point within 1.5× the interquartile range below and above Q1 and Q3, respectively. The rest of the observations, including the maximum and minimum values, are shown as outliers. $n = 626$ (1/α), $n = 83$ (1/β), $n = 570$ (2/α) and $n = 169$ (2/β) TADs.

At the sub-megabase scale, the skate genome is organized into TADs with a median size of 800 kb (Extended Data Fig. 4a,b), an intermediate regime between mammals and teleosts (Supplementary Fig. 4). Aggregate analyses revealed that skate TADs are associated with chromatin loops at the upper corner of domains (Fig. 3a). Chromatin accessibility (ATAC–seq) and motif enrichment analysis revealed binding sites for the architectural factor CTCF at skate TAD boundaries (Extended Data Fig. 4c,d), in comparable proportions to mammals and teleosts (Supplementary Fig. 5). These CTCF sites display an orientation bias with motifs oriented towards the interior of TADs, suggesting that these domains are formed by loop extrusion (Fig. 3b and Extended Data Fig. 4c). Notably, the critical genes involved in loop extrusion are present in the skate genome, including *ctcf* and those encoding cohesin complex subunits (*smc1a*, *smc3*, *scc1* and two copies of *scc3*). An example of skate TAD organization can be observed at the *hoxa* and *hoxd* clusters (Fig. 3c and Extended Data Fig. 4d), which display

the characteristic bipartite TAD configuration of jawed vertebrates[32]. Manual microsynteny analysis confirmed that the 3′ and 5′ TADs found at both skate *hox* loci are orthologous to those described in mammals and teleosts. Such deeply conserved 3D organizations reflect the existence of regulatory constraints that influenced TAD evolution across the whole jawed vertebrate clade.

To investigate enhancer–promoter interactions, we used Hi-C combined with immunoprecipitation (HiChIP) to associate H3K4me3-rich active promoters with potential regulatory loci in the anterior and posterior skate pectoral fin. Notably, these fin regions display transcriptional signatures that differ from other vertebrates. In particular, several 3′ *hoxa* and *hoxd* genes are preferentially expressed in the anterior pectoral fin, whereas 5′ *hoxa* and *hoxd* genes are located in the posterior pectoral domain. This pattern of expression has been consistently found in other batoid species[1,33]. HiChIP analyses revealed 50,601 interactions associated with 7,887 different promoters (6.4 interactions per active promoter). Interactions connecting promoters with distal ATAC–seq peaks ($\chi^2$, $P < 10^{-138}$; Extended Data Fig. 5a) and intra-TAD interactions were enriched (empirical $P < 10^{-4}$; Extended Data Fig. 5b). Differential analysis revealed similar looping patterns between tissues (Pearson correlation > 0.96; Extended Data Fig. 5c), with only 9 and 5 interactions statistically enriched in anterior and posterior fins, respectively (Extended Data Fig. 5d). Promoters with differential looping included *hoxa* and *hoxb* genes and the transcription factor *alx4* (Extended Data Fig. 5e–g), which are involved in limb development. To confirm those interactions, we performed Hi-C in anterior and posterior pectoral fins, finding only minor variations. Compartment differences were subtle and restricted to less than 10% of the genome (Extended Data Fig. 6a–d). TADs were also extremely similar (Fig. 3d,e and Extended Data Fig. 6e), with insulation score correlations of above 0.98 (Extended Data Fig. 6f). Similarly, high correlations were observed for chromatin loops (Extended Data Fig. 6g) and differential analysis revealed a single significantly stronger loop in the posterior pectoral fin (Extended Data Fig. 6h,i). Notably, the differential contacts predicted by HiChIP were not noticeable (Fig. 3d,e and Extended Data Fig. 6j). The differences in HiChIP data are therefore probably derived from variations in H3K4me3 occupancy, consistent with the selective activation of the *hoxa* cluster in anterior fins. Overall, both analyses indicate that 3D chromatin folding is largely maintained in the different pectoral fin territories.

To investigate possible regulatory constraints on TAD evolution, we considered 1,464 microsyntenic pairs of genes (that is, consecutive orthologues) conserved between skate, mouse and gar. In skates, such conserved gene pairs shared TADs more often than other consecutive genes (98% versus 95%, $\chi^2$, $P = 3.7 \times 10^{-13}$; Extended Data Fig. 7a). Those pairs were present in 718 out of the 1,678 skate TADs (42%), highlighting that individual TADs are constrained but not invariant across deep evolutionary timescales (Extended Data Fig. 7b). TADs containing deeply conserved microsyntenic pairs are significantly larger and contain more distal ATAC–seq peaks and putative promoter–enhancer interactions, as defined on the basis of HiChIP analysis, compared with non-conserved TADs (Extended Data Fig. 7c; Mann–Whitney *U*-test, $P = 1.23 \times 10^{-24}$, $3.81 \times 10^{-36}$ and $1.04 \times 10^{-41}$, respectively). This suggests that the deep conservation of individual TADs emerges from regulatory constraints (Extended Data Fig. 7d,e).

Our results suggest that 3D chromatin organization in skates results from the interplay of two mechanisms—compartmentalization driven by transcriptional state and TADs formed by loop extrusion. Such organization is similar in bony fishes/tetrapods, indicating that TAD formation through loop extrusion was present in the gnathostome ancestor. As the appearance of this common ancestor was temporally close to 2R, we explored the regulatory fate of homologous TADs in relation to this duplication event. We found that, although the size and gene density of TADs is similar between α and β chromosomes, there are notably fewer TADs in beta (Fig. 3f,g and Extended Data Fig. 7f).

Regulatory landscapes derived from H3K4me3 HiChIP experiments followed a similar trend (Extended Data Fig. 7g,h). We confirmed that the lower number of TADs in beta could not be explained by TAD fusions in beta or boundary gains in α segments (Extended Data Fig. 7i). These results indicate that many TADs disappeared from the early gnathostome genome after 2R, while those that persist are comparable in size (Fig. 3g). Whether losses in beta segments were caused by the deletion of whole redundant TADs or the progressive erosion and pseudogenization of their genes is difficult to ascertain.

## PCP pathway as a driver of fin expansion

To examine whether genomic rearrangements could have driven skate pectoral fin evolution through TAD alterations, as reported for other mammalian traits[11], we identified synteny breaks by aligning six jawed vertebrate genomes (Fig. 4a). As expected, the number of (micro)syntenic changes between species increases with phylogenetic distance (Fig. 4a and Extended Data Fig. 8a), from 18 breaks in *L. erinacea* that occurred after the split of the two skate lineages to 1,801 between cartilaginous and bony fishes (around 2 breaks per million years).

As anterior expansion of the pectoral fin is a defining characteristic of skates, we focused on the 123 synteny breaks shared by the little and thorny skate genomes relative to other vertebrates. We found an enrichment of synteny breaks near TAD boundaries—42 breaks occurred within 50 kb of a TAD boundary, compared with 15 expected under a random break model (empirical $P < 1 \times 10^{-4}$; Fig. 4b). This enrichment supports the hypothesis that genome rearrangements that interrupt TADs are evolutionarily disfavoured owing to deleterious enhancer–promoter rewiring[9].

Conversely, we hypothesized that the 81 breaks that interrupt TADs could be enriched for enhancer–promoter rewiring associated with gene regulatory changes. Interrupted TADs include 2,041 genes and, by filtering those with interactions across synteny breaks on the basis of anterior fin H3K4me3 HiChIP analysis, we identified 180 genes that are potentially associated with pectoral fin expansion. Signalling pathway analysis revealed enrichment for Wnt/PCP pathway components (Fig. 4c and Extended Data Fig. 8b,c), including the important regulator *prickle1* (Fig. 4d) and other potentially relevant genes such as the *hox* gene activator *psip1*[34] (Extended Data Fig. 8d–g). Among eight candidate genes of which we determined the expression using whole-mount in situ hybridization (WISH), only *prickle1* and *psip1* exhibited clear anteriorly enriched expression patterns (Fig. 4e and Supplementary Fig. 6).

To test whether alterations in TADs drove changes in gene expression, we performed comparative WISH analysis of *prickle1* between skate and chain catshark (*S. retifer*) embryos at equivalent stages (Fig. 4e). *prickle1* expression was higher in the anterior pectoral fin of skates compared to a weak expression without spatial enrichment in shark fins (Supplementary Fig. 7). Similarly, we found differential expression for *Psip1*, suggesting a potential involvement of *Hox*-related pathways in the skate fin phenotype (Extended Data Fig. 8f,g).

Given the specific pattern of *prickle1* expression, we examined the function of the PCP pathway in anterior fin expansion using cell shape analysis, and found that anterior mesenchymal cells are more oval than those in the central and posterior regions (Supplementary Fig. 8). Treatment with a Rho-kinase (ROCK) inhibitor from stage 29 to 31 showed that the overall number of fin rays associated to each tribasal bone of the skate fin (propterygium, mesopterygium and metapterygium) was reduced in the ROCK-inhibited embryos compared with in the controls, with greater losses in the anterior than in the posterior fin region (Fig. 4f,g, Extended Data Fig. 9 and Supplementary Figs. 9 and 10). Despite significant variation across stage and treatment (Extended Data Fig. 9 and Supplementary Fig. 10), geometric morphometric analyses suggest that ROCK-inhibitor-treated embryos showed a less pronounced anterior expansion of the pectoral fin, in contrast to control embryos in which it extends anteriorly towards the eye by stage 31

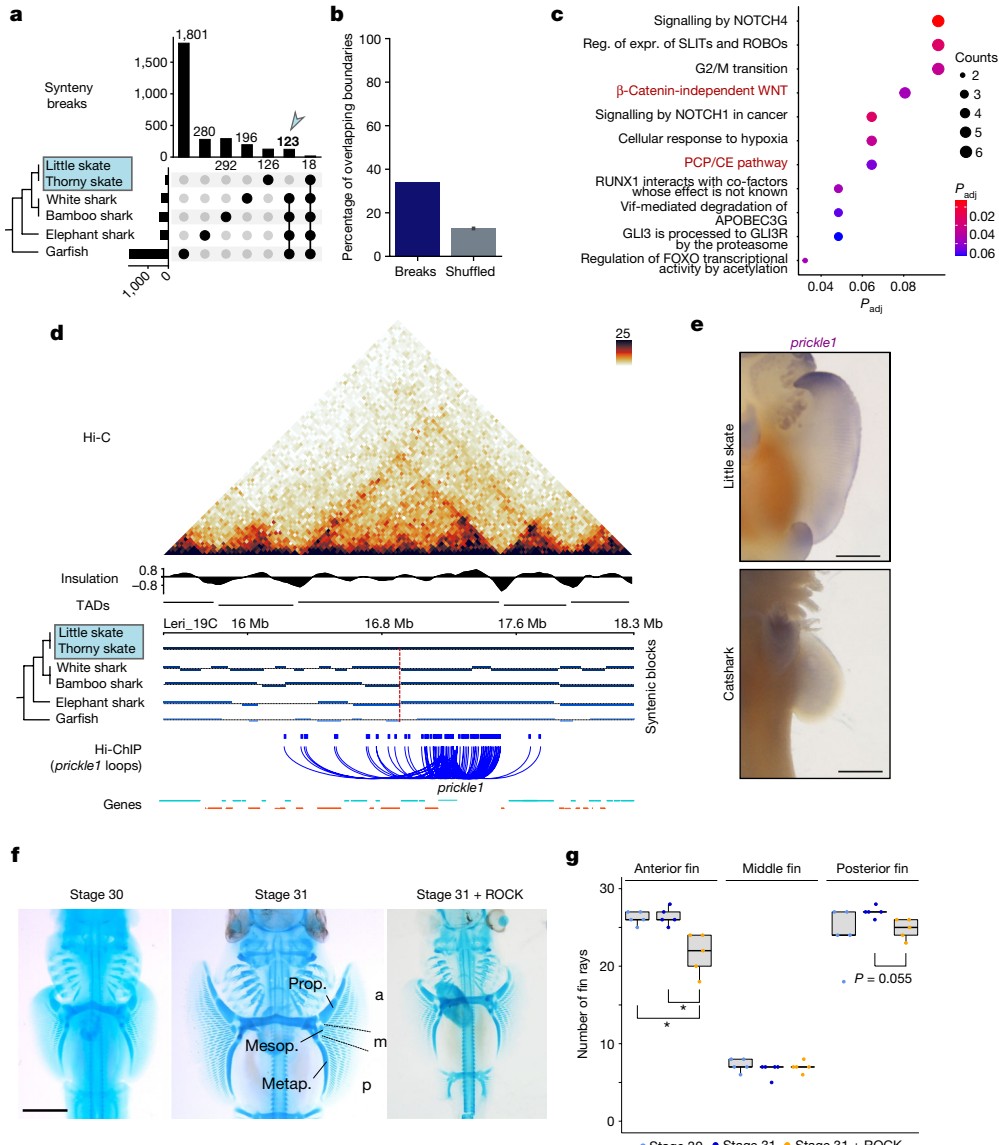

**Fig. 4 | Skate-specific genomic rearrangements and the PCP pathway. a**, Upset plot for quantification of synteny breaks in vertebrate species with the skate genome as a reference. The bar plot at the top shows quantification of synteny breaks for the species combination indicated by dots. The blue arrow highlights the 123 synteny breaks found in non-skate species, therefore probably derived in skates. The bar plot on the left shows the total quantification of synteny breaks for individual species. **b**, The percentage of synteny breaks at TAD boundaries (dark blue) and the expected percentage for shuffled boundaries (grey). **c**, Reactome signalling pathway analysis of genes contained in rearranged TADs. expr., expression; $P_{adj}$, adjusted $P$; reg., regulation. **d**, Hi-C map from pectoral fins for the *prickle1* locus. Synteny blocks, insulation scores, TAD predictions and chromatin loops detected in H3K4me3 HiChIP datasets are indicated. **e**, WISH analysis of *prickle1* in skates (*L. erinacea*, stage 30) and catshark (*Scyliorhinus retifer*, stage 30). Note that anterior expression is skate specific. $n$ = 5 animals. Scale bars, 1 mm. **f**, Cartilage staining of embryos with or without ROCK inhibitor. Compared with the stage 30 and 31 controls, the number of fin rays decreased in embryos treated with ROCK inhibitor. Note the more severe reduction in fin rays in the anterior compared with in the posterior pectoral fin. Photographs of all replicates are provided in Extended Data Fig. 11 and Supplementary Fig. 10. Scale bar, 2 mm. The pectoral fin was divided into three domains from anterior to posterior (Methods). Prop., propterygium; mesop., mesopterygium; metap., metapterygium; a, anterior; m, middle; p, posterior. **g**, Quantification of the number of rays emerging from propterygium, mesopterygium and metapterygium in samples for the conditions shown in **f**. Individual data points are shown. The box plots show the median (centre line), Q1 and Q3 (box limits), and the whiskers extend to the last point within 1.5× the interquartile range below and above Q1 and Q3, respectively. $P$ values were calculated using pairwise Wilcoxon rank-sum tests with correction for false-discovery rate (FDR); *$P$ < 0.05; $P$ = 0.018 in both significant comparisons in anterior fin.

(Extended Data Fig. 10). To rule out a general delay in body growth, we implanted acrylic beads soaked in ROCK inhibitor into the anterior pectoral fins at stage 29 and investigated fin rays at stage 31 (Extended Data Fig. 11). In contrast to control embryos with DMSO beads, specimens with ROCK inhibitor exhibited aberrant branching, fusion and loss of fin rays near beads or at potential bead implantation sites (6 out of 9 embryos for 100 μM and 6 out of 10 for 1 mM inhibitor beads). Taken together, these findings suggest that TAD rearrangements had a role in recruiting and repurposing genes and pathways during the evolution of the unique batoid fin morphology.

## HOX-driven *gli3* repression in skate fins

To examine the transcriptional drivers of skate fin morphology, we generated and compared RNA-seq datasets between pectoral fins and pelvic fins, which exhibit a characteristic tetrapod gene expression

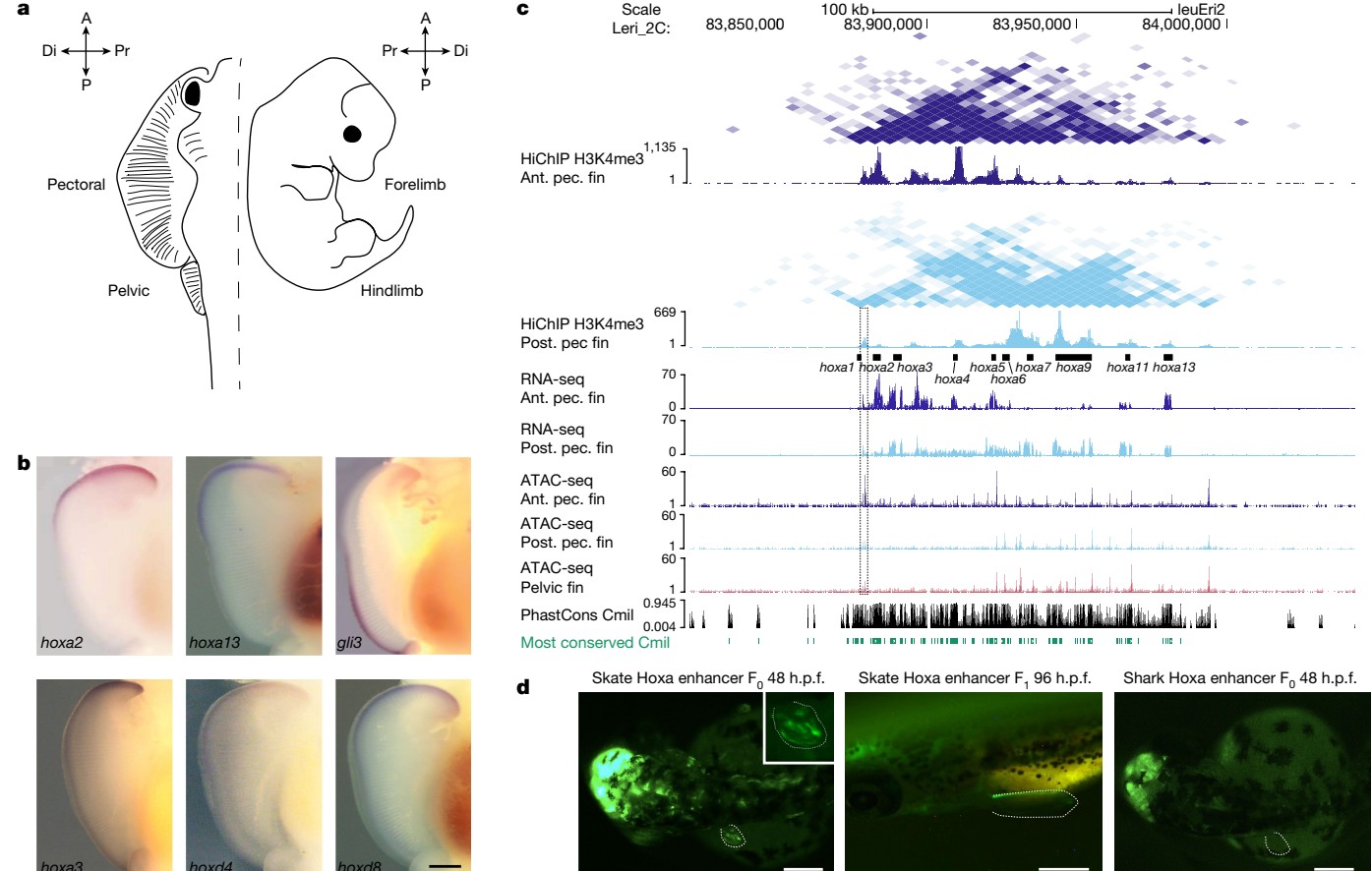

**Fig. 5 | Functional experiments in skate fin samples. a**, Outline of a skate and a mouse embryo and their homologous appendages, used in our comparative analyses. A, anterior; P, posterior. The distal (Di) and proximal (Pr) regions of the fin/limb are indicated. **b**, In situ hybridization reveals the opposite expression pattern of many *hox* genes and the *gli3* gene in the pectoral fin. *n* = 8 animals for each gene. Scale bar, 1 mm. The images of *hoxa2* and *gli3* were adapted from ref. 1. **c**, UCSC Genome Browser view showing HiChIP, RNA-seq and ATAC–seq data around the *hoxa* cluster in skate. The anterior-specific open chromatin region between the *hoxa1* and *hoxa2* genes is marked with a dotted rectangle (see 'A skate-specific *hoxa* fin enhancer' section). Green denotes the most conserved regions with the elephant shark (*Callorhinchus milii*; Cmil)

genome. Ant. pec. fin, anterior pectoral fin; post. pec. fin, posterior pectoral fin. **d**, GFP expression driven by the anterior-specific open chromatin region between the *hoxa1* and *hoxa2* genes from skate and shark in transgenic assays in zebrafish. The brain expression induced by the midbrain *enhancer:egfp* indicates a successful injection of the mini-Tol2 vector[61] with the skate or shark *hox* enhancer as a positive control. Note that only the skate enhancer drives expression on the pectoral fin (5 eGFP-positive embryos at 48 h after fertilization (h.p.f.) out of 18 $F_0$ embryos for the skate enhancer (left), in contrast to 0 out of 31 $F_0$ embryos for the shark enhancer (right)). In $F_1$ stable embryos, the GFP is driven to the pectoral fin with a clear anterior pattern at 96 h after fertilization (middle). Scale bars, 250 μm.

pattern[1]. We identified 193 and 117 genes preferentially expressed in pectoral and pelvic fins, respectively (Supplementary Table 4), including several transcription factors and components of different signalling pathways. To identify changes in the appendage gene regulatory network, we compared differentially expressed genes in skate fins with corresponding mouse fore- and hindlimb RNA-seq data[35,36] (Fig. 5a and Supplementary Fig. 11a). Key genes in determining anterior and posterior paired appendages, such as *tbx5* and *tbx4*, display a similar expression pattern, suggesting a conserved function across jawed vertebrates[33]. However, several genes, including *hox* genes or the master regulator of vertebrate hindlimb specification *pitx1*[37], displayed clear differences between skates and mice (Supplementary Figs. 11a and 12), suggesting that altered regulation of appendage-related factors may contribute to skate pectoral fin expansion.

To examine the transcriptional changes associated with skate pectoral fins, we analysed available anterior and posterior pectoral fin RNA-seq data[1]. In skates, *hox* genes show distinctive expression differences between the anterior and posterior pectoral fin (Supplementary Table 5 and Supplementary Fig. 11b). Anterior expression of the *hoxa* and *hoxd* genes forms a secondary AER-like organizer that is probably involved in the overgrowth of the skate pectoral fins[1,38,39]. Secondary

AER formation is associated with changes in the expression of *gli3*—a key regulator of hedgehog signalling in appendage patterning[40,41]. Specifically, *gli3* is expressed in the posterior pectoral fin versus predominantly anterior expression in pelvic fins, as in several vertebrate species[1] (Fig. 5b). Recently, it has been shown that (1) the *Hoxa13* and *Hoxd13* genes downregulate *Gli3* expression for proper thumb formation[42] in the mouse limb, (2) HOX13 proteins bind to and repress *Gli3* limb enhancers and (3) compound *Hox13* mutants cause anterior extension of *Gli3* expression[42]. Anterior *Hox* genes may also have a role in GLI3 transcriptional regulation, as *Hoxa2* binds to several enhancers within the *Gli3* locus (shown by ChIP–seq data[43]; Extended Data Fig. 12a). Overexpression of *hoxa2* in zebrafish pectoral fins also induces transcription of *wnt3* (an AER marker gene) potentially inhibiting *gli3* expression[1]. Some of these *hox* genes, including *hoxa13* and *hoxa2*, are strongly expressed in skate anterior pectoral fins (Supplementary Fig. 11b).

On the basis of this evidence, and considering the redundancy between Hoxd13 and Hoxa13 proteins[44–46], we explored the Hox–Gli3 relationship using a validated hoxd13a-GR overexpression construct in zebrafish[47]. After dexamethasone treatment, overexpression of Hoxd13a caused increased fin proliferation, distal expansion of

chondrogenic tissue and fin fold reduction[45]. Furthermore, 35% of the injected zebrafish embryos showed a decrease in *gli3* fin expression (Extended Data Fig. 12b). Moreover, a *gli3* loss-of-function mutant in medaka fish shows multiple radials and rays in a pattern similar to the polydactyly of mouse *gli3* mutants, but also to pectoral skate fins[48]. These findings, together with the anterior expression of 3′ *hox* genes, suggest that Gli3 downregulation, mediated by Hox repression, is a potential mechanism underlying the striking pectoral skate fin shape.

## A skate-specific *hoxa* fin enhancer

We hypothesized that the anteroposterior expression differences found in other vertebrates but not in skates could arise from changes in *cis*-regulation. To identify CREs, we performed ATAC–seq analysis in anterior and posterior pectoral fins, as well as in whole pelvic fins. DNA methylation profiling (Supplementary Fig. 16a) revealed that differentially accessible ATAC peaks are hypomethylated in developing pectoral and pelvic fins and remain hypomethylated in adult fins (Supplementary Fig. 16b,c), suggesting epigenetic memory as reported in other vertebrates[48–50]. We used our HiChIP datasets to associate CREs with target genes, and identified many differentially accessible ATAC peaks clustered around genes that are critical for appendage patterning, such as, *tbx5*, *tbx4*, *pitx1* and *hox* genes (Supplementary Tables 6 and 7). Notably, *Pitx1* displays a similar regulatory landscape in skate pectoral and pelvic fins (Supplementary Figs. 12 and 13), contrasting with the tissue-specific regularion in mouse[51].

To further investigate anterior *Hox* gene regulation in skate pectoral fins, we integrated our anterior and posterior pectoral fin ATAC–seq data with existing RNA-seq data from these tissues[1]. The few differentially accessible CREs were associated with differentially expressed genes relevant for patterning, such as *hoxa2*, *pax9*, *tbx2* and *alx4* anteriorly, as well as *chordin*, *hoxa9*, *hoxd10*, *hoxd11*, *hoxd12* and *grem1* in the posterior region (Supplementary Table 5). Notably, a region located between *hoxa1* and *hoxa2* is more accessible in anterior pectoral than in posterior pectoral or pelvic fins (Fig. 5c). Zebrafish transgenic assays confirmed enhancer activity for this open chromatin region, which drives gene expression in anterior pectoral fins (Fig. 5d). This element is conserved in cartilaginous fishes but not found in bony fishes (Supplementary Fig. 14). Importantly, the orthologous region in catshark does not promote transgene expression in zebrafish (Fig. 5d), suggesting that, although this region is conserved in different chondrichthyan species, only the skate sequence is functionally active during early development. As this potential enhancer lies close to the *hoxa2* promoter, we examined whether it is specific for *hoxa2* or shared with other *hox* genes. Using H3K4me4 HiChIP, HiC and virtual 4C data, we observed that this enhancer forms robust interactions with most genes of the *hox* cluster in the anterior pectoral fin (Fig. 5c and Supplementary Fig. 15a), including *hoxa13* located in the 5′ adjacent TAD (Figs. 3c and 5c) and expressed in the anterior pectoral fin (Fig. 5b and Supplementary Fig. 15b). Overall, these results demonstrate the existence of skate-specific CREs that can be linked to the formation of a secondary AER-like domain in the anterior pectoral fin.

## Discussion

Here we combined genomic and functional approaches to uncover fundamental principles of genome regulation in the skate lineage and provide a molecular basis for the formation of wing-like batoid fins[2]. The position of skates in the vertebrate evolutionary tree, and their slow rate of genome evolution, revealed new insights into karyotype stabilization after two rounds of WGD. Gene loss and karyotype evolution dynamics have occurred at a different pace across jawed vertebrate lineages. Analysis of the elephant shark genome found a slower rate of evolution and reduced gene loss compared with tetrapods[25,52]. Here we

showed that skate not only possesses comparably low rates of change, but also retains numerous ancestral gnathostome chromosomes, and that the smaller chromosome numbers of chicken and spotted gar arose by fusion of these ancestral units. This process was accompanied by considerable gene order rearrangement between cartilaginous and bony fishes, despite extensive conservation of TAD gene contents. Conservation of TADs in the absence of a globally colinear gene order emphasizes the impact of regulatory constraints in maintaining gene groupings.

The skate genome is functionally constrained by 3D regulatory mechanisms that parallel those described in bony fishes and tetrapods, including the presence of a CTCF-orientation code and associated loop extrusion[13]. Our findings imply that these mechanisms emerged early in vertebrate evolution, probably influencing the appearance of phenotypic novelties. These mechanisms further constrain genome evolution, as most skate-specific chromosome rearrangements occur at TAD boundaries, resulting in limited effects on gene regulation, as reported in mammals[53]. Notably, we observed the complete disappearance of TADs in the paralogous regions prone to gene loss after 2R (beta segments), with the remaining β and α TADs having the same average size and gene number. Although asymmetric paralogue loss after WGDs is considered to be a key factor in the emergence of novel gene regulation[5], the loss of TADs in beta regions indicates that entire paralogous regulatory units can be lost after WGDs and stresses the importance of regulatory constraints in shaping genome organization. It remains to be seen whether the regulatory potential of missing TADs is incorporated into other regulatory landscapes and enhances pleiotropy.

Related to novel skate morphology, we found lineage-specific TAD-disrupting rearrangements affecting genes involved in PCP signalling—an ancient developmental pathway[54] that is essential for cell orientation and patterning. We found that the main effector of this pathway, *prickle1*, has anteriorized pectoral fin expression as well as in anterior pelvic fins and in the clasper (Fig. 4e and Supplementary Fig. 6)—two structures that also extend laterally and posteriorly during skate development[55]. Importantly, unique pectoral and pelvic fin morphologies evolved simultaneously during batoid diversification, suggesting a deployment of similar/same genetic cascades during paired fin development[56], as suggested by the presence of common markers like *wnt3* and *hoxa11*[1,39]. The tissue-specific modulation of the PCP pathway through redeployment of a main pathway effector (*prickle1*) provides a compelling example of how existing gene networks can evolve new functions through genomic rearrangements.

Finally, we implicate altered regulation of 3′ *hox* genes and their activator *psip1* in novel skate pectoral fin development. Although these genes show posterior expression in most vertebrate appendages (including skate pelvic fins), they are notably expressed in skate anterior pectoral fin. Our *hoxd13a* overexpression experiments (Extended Data Fig. 12b) suggest that the increased levels of *hox* gene expression in anterior pectoral fins, together with other regulatory changes, downregulates Gli3, leading to substantially altered morphology and illustrating the plasticity of the Shh–Gli3–Ptch1 pathway in the evolution of vertebrate appendage morphology[46,56–59]. The identified skate-specific *hoxa* fin enhancer suggests a *cis*-regulatory basis for altered Shh–Gli3–Ptch1 signalling. Overall, our study shows how changes in CREs and 3D chromatin organization act as essential forces driving adaptive evolution.

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

## Methods

### Animal use

All fish work, including experiments with skate embryos, was conducted according to standard protocols approved by the Institutional Animal Care and Use Committee (IACUC) of Rutgers University (protocol number, 201702646), the IACUC of Marine Biological Laboratory (protocol number, 18-36) and the University of Chicago IACUC (protocol number, 71033). *Danio rerio* embryos were obtained from AB and Tübingen strains, and manipulated according to protocols approved by the Ethics Committee of the Andalusia Government (license number, 182-41106) and the national and European regulation established. Zebrafish procedures were reviewed and approved by the ethical committees from the University Pablo de Olavide, CSIC, and the Andalusian government, and performed in compliance with all relevant ethical regulations.

### Genomic DNA extraction and library construction

Skate DNA was isolated using extensive proteinase K digestion and phenol–chloroform extraction from the muscle of a single *L. erinacea* specimen. For genome assembly, we generated both accurate short reads and noisy long reads. A contiguous long read (CLR) library for Pacbio sequencing was prepared and sequenced at the Vincent J. Coates Genomics Sequencing Laboratory at UC Berkeley. A total of 32 cells were sequenced on the Pacbio Sequel instrument using the V7 chemistry and yielded a total 10.2 million Pacbio reads totalling 163 Gb with a median size of 10.9 kb and a read $N_{50}$ of 29 kb.

A paired-end Illumina library with a 600 bp insert was also sequenced for $2 \times 250$ bp in rapid run mode on the HiSeq 2500 instrument at BGI yielding 641 million reads and 160.3 Gb of sequence.

### Genome assembly

Genome size was estimated by analysing a $k$-mer spectrum with a mer size of 31. By fitting a multimodal distribution using Genomescope 2.0, and estimated a genome size of 2.13 Gb (as well as an heterozygosity of 1.56%)[62]. To take advantage of both short and long reads, we opted for a hybrid assembly strategy. First, we generated de Brujin graph contigs using megahit (v.1.1.1) using a multi-$k$-mer approach (31, 51, 71, 91 and 111-mers) and filtering out $k$-mers with a multiplicity lower than 5 (--min-count 5). We obtained 2,750,419 contigs with an $N_{50}$ of 1,129 bp representing a total of 2.23 Gb. We then used these contigs to prime the alignment and assembly of the Pacbio reads using dbg2olc (c.10037fa)[63] using a $k$-mer of 17 (k 17), a threshold on $k$-mer coverage of 3 (KmerCovTh 3), a minimal overlap of 30 (MinOverlap 30) and an adaptive threshold of 0.01 (AdaptiveTh 0.01) and removing chimeric reads from the dataset (RemoveChimera 1). This assembler generated an uncorrected backbone of overlapping reads with an $N_{50}$ of 4.96 Mb and a total size of 2.25 Gb. To correct sequencing errors, we processed this sequence file to two successive rounds of consensus by aligning Pacbio reads with minimap2 (v.2.12, map-pb setting)[64] and Racon (v.1.3.1) using the default parameters followed by one final round of consensus using the Illumina reads. We evaluated the progress of the polishing process with the BUSCO tool (v.3.0.2) that seeks widely represented single-copy gene families in the assembly[65]. Our final polished assembly contained 95.1% of vertebrate BUSCO genes (Supplementary Table 1). To exclude residual haploid contigs from the assembly, we aligned Illumina reads once more using bwa and computed a distribution of coverage that showed some residual positions at half coverage (31×). We used purge_haplotigs (v.1.0.2)[66] by defining a coverage threshold between haploid and diploid contigs at 40× (and a minimum of 10× and maximum of 100×). The filtered assembly has a size of 2.19 Gb, an $N_{50}$ of 5.35 Mb and 2,595 contigs in total, and the same BUSCO statistics as the unfiltered one (Supplementary Table 1).

This assembly was then scaffolded using chromatin-contact evidence obtained from Hi-C sequencing analysis of *L. erinacea* fins (see below) at Dovetail Genomics using the HiRise pipeline[67]. The accuracy of the resulting scaffolded assembly was verified and proofread by carefully inspecting the contact map in Juicebox[68] and HiGlass browser[69]. This assembly comprises 50 scaffolds larger than 1 Mb that represent 92% of the assembly size and 39 scaffolds larger than 10 Mb that show mostly internal contacts. Despite no karyotyping evidence is directly available for *L. erinacea*, closely related species show a haploid number of 49 chromosomes, which is consistent with the observed number of chromosomes[14].

As the final assembly size was smaller than the experimentally assessed genome size of 3.5 Gb, we performed gap closing on the final assembly using PBjelly[70] that proceeds through alignment of the PacBio reads on each gap border and local reassembly. The effect on the assembly statistics was marginal, but we used this assembly as our final one (Supplementary Table 1).

### Annotation

RNA-seq reads of strand-specific libraries from five bulk embryonic stages and 13 organs were aligned to the genome using STAR (v.2.5.2b)[71] and each library assembled independently using stringtie (v.1.3.3)[72]. Stringtie assemblies were then merged using TACO (v.0.7.3)[73]. RNA-seq reads were also assembled de novo using Trinity (v.2.8.4)[74]. Finally, the iso-seq protocol was applied to generate full-length transcripts using Pacbio long-reads. Both Trinity assembled transcripts and iso-seq transcripts were aligned to the genome using GMAP (v.2018-07-04)[75]. Then, both TACO assembled transcripts and aligned de novo transcripts were leveraged using Mikado (v.1.2.1)[76] to generate one consensus reference transcriptome, while predicting coding loci using Transdecoder (v.5.5.0). Using selected transcripts (2 introns or more, complete CDS, single hit against swissprot), we built an Augustus (v.3.3.3) hidden Markov model (HMM) profile for ab initio gene prediction[77]. We predicted skate genes using this profile and hints derived from (1) the mikado transcript assembly (exon hints); (2) intron hits obtained using bam2hints on a merged bam alignment of the RNA-seq data after filtering spurious junctions with portcullis (v.1.2.0)[78]; and (3) an alignment of human protein using exonerate (v.2.2.0)[79].

A repeat library was constructed using Repeatmodeler and repeats were masked in the genome using Repeatmasker (v.4.0.7). We filtered out gene models that overlap massively with mobile elements and obtained 30,489 genes models. For these genes, isoforms and untranslated regions were added by two rounds of reconciliation with an assembled transcriptome using PASA[80]. Our set of coding genes includes 5,800 PFAM domains, a similar value to other well-annotated vertebrate genomes. To further examine the validity of gene models, we assessed (1) whether their coding sequence showed similarity to that of another species using gene family reconstruction (see below); (2) whether they possessed an annotated PFAM domain; and (3) whether they are expressed above 2 FPKMs in at least one RNA-seq dataset. These criteria reduced the number of bona fide coding genes to 26,715.

### Gene family, synteny and phylogenetic analyses

We performed gene family reconstruction using OMA (v.2.4.1)[81] between selected vertebrate species to identify single-copy orthologues. These orthologues were used to infer gene phylogeny after processing as described previously[82]: HMM profiles were built for each orthologous gene family and searched against translated transcriptomes using the HMMer tool (v.3.1b2)[83]. Alignments derived from each orthologue were aligned using MAFFT (v.7.3)[84], trimmed for misaligned regions using BMGE (v.1.12)[85] and assembled in a supermatrix. Phylogeny was estimated using IQTREE (v.2.1.1) assuming a C60+R model and divergence times estimated using Phylobayes (v.4.1e)[86] assuming a CAT+GTR substitution, and a CIR clock model, soft constraints and a birth-death prior on divergence time. Calibrations were taken from previous papers[18,87].

We identified conserved segments across vertebrates, by counting single-copy copy genes derived from OMA clustering sharing the same

set of chromosomal locations in selected species, to identify putative ancestral vertebrate units. We examined conserved syntenic orthology by identifying sets of genes shared by pairs of chromosomes in distinct species using reciprocal best hits computed using Mmseq2[88]. We performed a Fisher test to detect pairs of chromosomes showing significant enrichment, and assigned ancestral linkage groups (ALG) based on comparison with amphioxus and sea scallop. We computed gene family composition and analysed patterns of gene loss and duplications using reconstructed gene trees derived from gene families established with Broccoli[89] and subjected to species-tree aware gene tree inference using Generax[90].

## Hi-C

The Hi-C protocol was performed as described previously with minor modifications[91–93]. Two biological replicates of *L. erinacea* Stg.30 pectoral fin buds, each consisting of ten fins, were fixed in a final concentration of 1% PFA for 10 min at room temperature. Fixation was stopped by placing the samples on ice and by adding 1 M glycine up to a concentration of 0.125 M. The quenched PFA solution was then removed and the tissue was resuspended in ice-cold Hi-C Lysis Buffer (10 mM pH 8 Tris-HCl, 10 mM NaCl, 0.2% NP-40 and 1× Roche Complete protease inhibitor). The lysis was helped with a Dounce Homogenizer Pestle A on ice (series of 10 strokes in 10 min intervals). Nuclei were then pelleted by centrifugation for 5 min, 750 rcf at 4 °C, washed twice with 500 μl of 1× PBS and finally resuspended with water to final volume of 50 μl. A total of 50 μl of 1% SDS was then added and the sample incubated 10 min at 62 °C. The SDS was then quenched by adding 292 μl water and 50 μl of 10% Triton X-100. Chromatin was then digested by adding 50 μl of 10× DpnII buffer and 8 μl of 50 U μl⁻¹ DpnII (NEB, R0543M) followed by incubation at 37 °C overnight in a ThermoMixer with shaking (800 rpm). DpnII was then heat-inactivated at 65 °C for 20 min with no shaking. Chromatin sticky ends were then filled-in and marked with biotin by adding 50 μl of Fill-in Master Mix (5 μl of 10× NEBuffer2, 1.5 μl of 10 mM mix of dCTP, dGTP and dTTP, 37.5 μl of 0.4 mM biotin-dATP (Thermo Fisher Scientific, 19524016) and 10 μl of 5 U μl⁻¹ Klenow (NEB, M0210)) and incubating for 1 h at 37 °C with rotation. Filled-in chromatin was then ligated by adding 500 μl of ligation master mix (100 μl of 10× NEB T4 DNA ligase buffer with ATP (NEB, B0202), 100 μl of 10% Triton X-100, 10 μl of 10 mg ml⁻¹ BSA and 6.5 μl of 400 U μl⁻¹ of T4 DNA ligase (NEB, M0202)) and incubated 4 h at 16 °C with mixing (800 rpm, 30 s pulses every 4 min). Ligated chromatin was then reverse-cross-linked by adding 50 μl of 10 mg ml⁻¹ proteinase K and incubating the sample at 65 °C for 2 h. De-cross-linking was completed by adding 50 μl extra of proteinase K and incubating overnight at 65 °C. DNA from the reverse-cross-linked chromatin was purified using phenol–chloroform extraction and ethanol precipitation. Pelleted DNA was resuspended in 100 μl of TLE. Biotin removal from unligated ends was performed in a final volume of 130 μl with 5 μg of the purified DNA, 13 μl of 10× NEBuffer2.1, 3.25 μl of 1 mM dNTPs, 5 μl of 3 U μl⁻¹ T4 DNA polymerase (NEB, M0203L). The sample was incubated in a thermocycler 4 h at 20 °C and the reaction subsequently stopped by adding EDTA to a final concentration of 10 mM followed by 20 min at 75 °C. A total of 130 μl was used for DNA sonication in a M220 Covaris Sonicator (peak power, 50; duty factor, 20%; cycles/burst, 200; duration, 65 s). After sonication, DNA was size-selected using AMPure XP beads (Agencourt, A63881). In brief, in a first selection, 0.6× bead mix was used and the supernatant was recovered. In the second selection, 1.2× bead mix was used and the bead fraction was recovered. Size-selected DNA was resuspended in 50 μl of TLE and then processed for end repair. End repair was performed by adding 20 μl of the end repair mix (7 μl of 10× NEB ligation buffer, 1.75 μl of 10 mM dNTP mix, 2.5 μl of T4 DNA polymerase (3 U μl⁻¹ NEB M0203), 2.5 μl of T4 PNK (10 U μl⁻¹, NEB M0201) and 0.5 μl of Klenow DNA polymerase (5 U μl⁻¹, NEB, M0210)) and incubating in a thermocycler with the following program: 15 °C for 15 min, 25 °C for 15 min and 75 °C for 20 min. Biotinylated ligation ends were then pulled down using 10 μl of Dynabeads MyOne Streptavidin C1 (Invitrogen, 650.01) per μg of DNA. The beads were washed twice with Tween wash buffer (85 mM Tris-HCl pH 8, 0.5 mM EDTA, 1 M NaCl, 0.05% Tween-20) before being resuspended in 400 μl of 2× bead binding buffer (10 mM Tris-HCl pH 8, 1 mM EDTA, 50 mM NaCl) and incubated for 15 min with rotation with 400 μl of the end repaired sample (70 μl of end repair reaction plus 330 μl of TLE). The beads were then washed once with 400 μl of 1× bead binding buffer and once with 100 μl TLE before being finally resuspended in a final volume of 41 μl. A-tailing was then performed in a total volume of 50 μl by adding 5 μl of 10× NEBuffer2.1, 1 μl of 10 mM dATP and 3 μl of 5 U μl⁻¹ Klenow fragment 3′→5′ exo- (NEB, M0212) in the thermocycler with the following program: 37 °C for 30 min then 75 °C for 20 min. A-tailed sample was then washed with 400 μl of 1× T4 ligase buffer and resuspended in 40 μl of the same buffer to prepare it for the adaptor ligation, which was performed by adding 1 μl of 10× T4 ligation buffer, 4 μl of T4 DNA ligase and 5 μl of 15 μM Illumina paired-end pre-annealed adapters. The reaction was incubated for 2 h at room temperature and the beads were then washed twice with 1× NEBuffer2.1. The beads were resuspended in 50 μl of the final library PCR reaction for library generation (25 μl of NEBNext High-Fidelity 2× PCR Mix, 0.5 μl of PE1 primer 25 μM and 0.5 μl of PE2 primer 25 μM plus milliQ water). The PCR was performed in a thermocycler with the following program: 98 °C for 60 s; 5–10 cycles of 98 °C for 10 s, 65 °C for 30 s, 72 °C for 30 s and 72 °C for 5 min. Test PCRs were used to determine the number of cycles. Final single-sided AMPure XP bead purification was performed to eliminate primer-dimers (1.1× proportion). Final libraries were sent for paired-end sequencing.

## Hi-C analysis

Hi-C paired-end reads were mapped to the skate genome using BWA[94]. Ligation events (Hi-C pairs) were then detected and sorted, and PCR duplicates were removed using the pairtools package (https://github.com/mirnylab/pairtools). Unligated and self-ligated events (dangling and extra-dangling ends, respectively) were filtered out by removing contacts mapping to the same or adjacent restriction fragments. The resulting filtered pairs file was converted to a .tsv file that was used as input for Juicer Tools 1.13.02 Pre, which generated multiresolution .hic files[95]. These analyses were performed using previously published custom scripts (https://gitlab.com/rdacemel/hic_ctcf-null): the hic_pipe.py script was first used to generate .tsv files with the filtered pairs, and the filt2hic.sh script was then used to generate Juicer .hic files. Visualization of normalized Hi-C matrices and other values described below, such as insulation scores, TAD boundaries, aggregate TAD, Pearson's correlation matrices and eigenvectors, were calculated and visualized using FAN-C[96] and custom scripts available in the GitLab repository (https://gitlab.com/skategenome). The observed–expected interchromosomal matrix (Fig. 1d) was calculated counting interchromosomal normalized interactions in the 1 Mb KR normalized matrix (with the two replicates merged). Expected matrix was calculated as if interchromosomal interactions between two given chromosomes were proportional to the total number of interchromosomal interactions of these two chromosomes. A/B compartments were first called in each of the replicates separately using the first eigenvector of the 500 kb KR normalized matrix. Eigenvector correlation was high ($r = 0.91$, Extended Data Fig. 3b) and the replicates were then merged. The first eigenvector was calculated again and oriented according to open chromatin using the amount of ATAC–seq signal in the anterior pectoral fin sample. The same strategy was used to look at compartment differences between anterior and posterior fin Hi-C, but this time using 250 kb resolution (Extended Data Fig. 6). ATAC–seq, percentage of GC, gene models and RNA-seq signal overlaps with compartments were calculated using bedtools intersect[97]. Compartment calling and the different overlaps are available in Supplementary Table 8. The saddle plot was calculated using FAN-C. To define TADs, insulation scores were also calculated separately in the 25 kb resolution KR matrices of

each of the replicates (using FAN-C and as described previously[98] with a window size of 500 kb). Again, correlation between insulation scores of both replicates was high ($r > 0.94$, Extended Data Figs. 4b and 6f). Definitive boundaries and TADs were then calculated in a merged 25 kb matrix with a window size of 500 kb and using a boundary score cut-off of 1 (Supplementary Table 9) or no cut-off for interspecies comparison analyses with mouse and zebrafish. CTCF motifs and their relative orientations were mined inside ChIP–seq peaks in mouse and zebrafish or merged ATAC–seq peaks between the anterior and posterior pectoral fin samples using Clover[99] or FIMO[100] (MA0139.1 Jaspar PWM, PWM score threshold of 8). They were later overlapped with previously calculated boundaries. Boundary feature heat maps from Supplementary Fig. 5 were generated using profileplyr[101] (https://bioconductor.org/packages/release/bioc/html/profileplyr.html) after binning the different signals in 5 kb windowed bigwig files. Chromatin loops were called using HICCUPS[95] with the default parameters in merged replicates of the anterior and posterior fin Hi-C experiments, and in a megamap merging anterior and posterior fin Hi-C maps. A consensus set of loops was then calculated using hicMergeLoops from the HiCExplorer suite[102] and reads were counted in the different replicate 10 kb resolution Hi-C maps to perform the differential loop analysis with EdgeR[103]. Virtual 4C-seqs were plotted from 10-kb-resolution Hi-C matrices using custom scripts.

## HiChIP

HiChIP assays were performed as previously described[104], with some modifications. In brief, 10 anterior and posterior pectoral fins of stg. In total, 30 skate embryos were fixed in a final concentration of 1% PFA for 10 min at room temperature. Fixation was quenched with 1 M glycine up to a concentration of 0.125 M. The tissue was then resuspended in 5 ml cell lysis buffer and homogenized using a Douncer on ice. After the lysis, nuclei were pelleted by centrifuging at 2,500 rcf, and washed in 500 µl of lysis buffer. Chromatin digestion and ligation, ChIP, tagmentation and library preparation were performed as previously described[92]. The antibody used was a ChIP-grade anti-histone H3 trimethyl K4 from Abcam (ab8580). The total amount of antibody used was 20 µg, at a dilution of 1 µg µl⁻¹.

## HiChIP analysis

Paired-end reads from HiChIP experiments were aligned to the skate genome using the TADbit pipeline[105] with the default settings. In brief, duplicate reads were removed, DpnII restriction fragments were assigned to resulting read pairs, valid interactions were kept by filtering out unligated and self-ligated events and multiresolution interaction matrices were generated. Dangling-end read pairs were used to create 1D signal bedfiles that are equivalent to those of ChIP–seq experiments. Coverage profiles were then generated in the bedgraph format using the bedtools genomecov tool (https://bedtools.readthedocs.io/en/latest/content/tools/genomecov.html), and bedgraph to bigwig conversions were also performed for visualization using the bedGraphToBigWig tool from UCSC Kent Utils (https://github.com/ucscGenomeBrowser/kent). 1D signal bedgraph files were used to call peaks with MACS2[106] using the no model and extsize 147 parameters and FDR < 0.01.

FitHiChIP[107] was used to identify 'peak-to-all' interactions at 10 kb resolution using HiChIP-filtered pairs and peaks derived from dangling ends. Loops were called using a genomic distance of between 20 kb and 2 Mb, and coverage bias correction was performed to achieve normalization. FitHiChIP loops with $q$ values smaller than 0.1 were retained for further analyses. Further filtering was performed to enrich enhancer–promoter interactions. First, loops established by two H3K4me3 peaks (likely promoter–promoter interactions) or no H3K4me3 peaks (likely enhancer–enhancer and others) were filtered out. Second, loops related to the H3K4me3 peak of the same gene promoter are grouped together into a common 'regulatory landscape', composed of a promoter anchor and several distal anchors. Then, regulatory landscapes with only one

distal anchor were filtered out. Third, to filter out further spurious interactions, we used the rationale that genomic bins that interact with a given promoter rarely do so in isolation. We therefore calculated a distance cut-off for 'interaction gaps' in regulatory landscapes. Regulatory landscapes containing interaction gaps bigger than the distance cut-off were trimmed and the distal anchors beyond the interaction gap were discarded. The cut-off was determined for each sample independently by calculating the distribution of the biggest gaps (calculating the biggest gap for each regulatory landscape) and setting the cut-off to the sum of the third quartile plus twice the interquartile range (classic outlier definition). Overlaps with ATAC–seq peaks in the pectoral fin were calculated using bedtools intersect (Extended Data Fig. 5a). Inter-TAD loops were also calculated using bedtools intersect -c using the TADs and the loops. Loops intersecting more than one TAD were considered inter-TAD loops. Randomized controls were generated shuffling TAD positions before the intersection using bedtools shuffle. For differential analysis between the anterior and the posterior fin, filtered distal anchors were fused when closer than 20 kb using GenomicRanges reduce[108]. The loops with the merged distal anchors are provided in Supplementary Table 10. To perform the differential analysis, the number of reads supporting the union set of loops was extracted for each of the sample replicates. Correlations shown in Extended Data Fig. 5c and the differential analysis performed using EdgeR[103] (Extended Data Fig. 5d) were calculated with this table. An FDR cut-off of 0.1 was chosen to consider a loop to be significantly stronger in either the anterior or the posterior fin. Custom code used for enhancer–promoter loop filtering and differential analysis is included in the GitLab repository (https://gitlab.com/skategenome).

## RNA-seq

RNA-seq experiments from anterior and posterior pectoral and whole pelvic skate fins were performed as previously described[6]. In brief, two anterior or posterior pectoral and two pelvic fins of stage 31 skate embryos were used for each biological replicate. Total RNA was extracted from each sample using Direct-zol RNA MiniPrep (Zymo Research) and sent for library preparation and sequencing.

## RNA-seq analysis

For the RNA-seq data analysis, we used the nf-core/rnaseq pipeline (v.1.4)[109] for read alignment, read count and quality control of the results. After this, we performed a differential gene expression analysis using the DESeq2 R library (v.1.30.1)[110]. Gene Ontology term enrichment analysis was performed using TopGO R library (v.2.42.0)[111], with the elim algorithm and Fisher test, retaining terms with $P < 0.01$.

## ATAC–seq

ATAC–seq experiments from anterior and posterior regions of pectoral skate fins and whole pelvic fins were performed as previously described[6,112]. After dissecting the pectoral fins, one anterior and one posterior regions were used for each biological replicate. In the case of pelvic fins, one fin was used for each biological replicate. Tissue was homogenized using a Pellet Pestle Motor (Kimble) coupled to a plastic pestle, and treated with lysis buffer. Individual cells were counted, and 75,000 cells were tagmented. ATAC–seq libraries were generated by PCR, using 13 cycles of amplification, purified and sent for external sequencing.

## ATAC–seq analysis

ATAC–seq data analysis was performed using the nf-core/atacseq pipeline (v.1.0.0)[109], which runs Nextflow (v.19.10.0)[113], for quality controls, read alignment against the new skate assembly, filtering, data visualization, peak calling, read count and differential accessibility analysis. To compare whole pectoral and pelvic fin samples, we merged the anterior and posterior pectoral samples into one single pectoral fin sample.

## Microsyntenic pair analysis

The analysis of microsyntenic pairs shared across the gnathostome lineage was based on a previously described analysis[114]. In brief, we used the genome assembly and annotation presented in this paper for the little skate in combination with public assemblies and annotations for mouse and garfish downloaded from ensembl (www.ensembl.org; *Mus musculus*: GRCm38v101; *Lepisosteus oculatus*: LepOcu1v104). Annotations in .gtf format were converted to genepred with gtfToGenePred (UCSC Kent Utils). Then, for each pair of consecutive genes in skates, we determined whether the orthologue pairs of genes in mouse and garfish were also consecutive (allowing 4 intervening gene models as described previously[114]). The intergenic space between pairs of genes categorized as syntenic and non-syntenic in skates was overlapped with TAD boundaries and with TADs again using bedtools intersect. TADs were categorized according to the presence or absence of conserved microsyntenic pairs and then the overlap between the different TADs with ATAC−seq peaks or HiChIP loops was calculated again using bedtools intersect. A list of conserved microsyntenic pairs is provided in Supplementary Table 11 and the code is available in the GitLab repository (https://gitlab.com/skategenome).

## TAD rearrangements in the skate lineage

To identify skate-specific TAD rearrangements, global alignments were performed with lastz[115] against six different gnathostome genomes using as a reference the little skate assembly presented in this study. The chosen species were the thorny skate *Amblyraja radiata*, two species of shark (the white shark *Carcarodon carcarias* and the white-spotted bamboo shark *Chiloscyllium plagiosum*), one chimera (the elephant shark *Callorhinchus milii*) and a bony fish (the spotted gar *Lepisosteus oculatus*). The parameters of lastz were adapted to the phylogenetic distance with skate according to previous recommendations[116] (see assemblies, substitution matrices and lastz parameters used in Supplementary Table 12). Syntenic chains and nets were then devised as proposed elsewhere[117] and further polished using chainCleaner[118]. Synteny breaks were then defined as the junctions between syntenic nets of any level, excluding those that were caused by the end of a scaffold for such genome assemblies that were not chromosome grade (white shark, elephant shark). The overlap between synteny breaks of different species was inferred using bedtools multiinter. Breaks that were found to be common in sharks, chimeras and a bony fish (garfish) were further considered. The distance between candidate synteny breaks and TAD boundaries (Supplementary Table 9) was next determined using bedtools closest -d and breaks that were located closer than 50 kb to a TAD boundary were discarded. Randomized analysis of the overlap between synteny breaks and TAD boundaries (Fig. 4b) was performed, combining bedtools closest and bedtools shuffle. Finally, we selected candidate genes that displayed enhancer−promoter HiChIP interactions in the anterior or the posterior pectoral fin samples that crossed the synteny break, using again bedtools intersect. Enrichment of signalling pathways of candidate genes was performed using the ReactomePA[119] and ClusterProfiler[120] R packages. A list of the final synteny breaks and candidate genes is provided in Supplementary Table 13, and the exact code used is provided at the GitLab code repository (https://gitlab.com/skategenome).

## WISH

Skate and shark embryos were recovered from egg cases at stage 27 and 30 and fixed by 4% PFA at 4 °C overnight. The next day, the embryos were rinsed three times with PBS-0.1% Tween-20, soaked in 100% methanol and stored at −80 °C. WISH was conducted as previously described[1], except for hybridizing the embryos and probes at 72 °C.

## Gain of function analysis

Experiments were performed as previously described[47]. Zebrafish eggs were injected at the one-cell stage with *hoxd13a-GR* mRNA (70 pg per embryo). Dexamethasone at 10 nM (Sigma-Aldrich, D4902) was added to the medium at 24 h after fertilization, and embryos were fixed at 48 h after fertilization.

## RT−qPCR

The pectoral fins of three shark juveniles (*S. retifer*) were dissected out in DEPC-PBS at stage 30. Three replicates were prepared. Total RNA was separately extracted from each replicate by Trizol (Invitrogen). cDNA was synthesized from the total RNA using the iScript cDNA Synthesis Kit (Bio-Rad). Then, quantitative reverse transcription PCR (RT−qPCR) analysis of *gapdh* and *prickle1* was conducted using the KAPA HiFi HotStart ReadyMix PCR Kit (Kapa Biosystems) and the Applied Biosystems 7300 Real time PCR system. A list of the primers used in this study is provided in Supplementary Table 14. The obtained $C_t$ value from RT−qPCR was converted to arbitrary gene expression values.

## Cell elongation analysis

Pectoral fins were dissected from stage 29 skate embryos and fixed by 4% paraformaldehyde overnight. The next day, the fins were rinsed with PBS including 0.1% Triton X-100 and incubated in the blocking buffer (10% sheep serum and 0.1% BSA in PBS-0.1% Triton X-100) at room temperature for 1 h. The blocking buffer was then replaced with blocking buffer including CellMask Deep Red Plasma membrane Stain (1/1,000 dilution, Invitrogen) and DAPI (1:4000 dilution), and incubated at 4 °C overnight. Subsequently, the fins were washed five times for 1 h with PBS-Triton X-100 and mounted onto glass slides. The fins were then scanned using a confocal microscope (Zeiss, LSM510 META). The scanned images were incorporated into Fiji and cell outlines in fin mesenchyme were manually traced. The cell elongation ratio was automatically calculated by the macro 'Tissue Cell Geometry Stats' included in Fiji.

## ROCK inhibitor treatment

To test the function of the PCP pathway in the pectoral fin development, skate embryos were treated with Y27632−a ROCK inhibitor−from stage 29 to stage 31 and investigated for their fin morphology. ROCK inhibitor (500 µl; stock 50 mM, final 50 µM, Selleck chemicals) or DMSO solution (negative control) was added to 500 ml of artificial saltwater (Instant Ocean), and five skate embryos at stage 29 for each condition were kept submerged in these solutions. Once the negative control embryos reached stage 31, all embryos were fixed by 4% PFA and their total body length was measured under a stereomicroscope. The embryos were stained with Alcian Blue as previously described[121] (*n* = 5 per condition).

To locally inhibit the PCP pathway by the ROCK inhibitor, the beads soaked in the inhibitor solution (100 µM or 1 mM in DMSO) or DMSO were repeatedly implanted into the anterior pectoral fin from stage 29 (one bead per week, three times as total for each embryo). The embryos were raised up to stage 31 in artificial saltwater, fixed by 4% PFA and stained with Alcian Blue (the replicates were 9 or 10 embryos per condition).

## Morphometrics analysis of skate fins

Skate embryos at each stage were photographed from the ventral side. A landmark scheme was designed to capture the shape of the pectoral fin (Extended Data Fig. 10). Six homologous landmarks and three curves were assessed in each sample; curves were used to generate sliding semi-landmarks. The samples were digitized in R using the package Stereomorph[122]. Digitized files were then uploaded to ShinyGM[123,124], in which all downstream analyses were performed. The samples were aligned using a generalized Procrustes analysis to account for shape differences due to differences in specimen size, specimen orientation and scaling. A morphospace was generated using these aligned landmark coordinates; deformation grids were generated for the control stage 31 and ROCK-inhibited stage 31 samples (Extended Data Fig. 10). A linear model was run to test for the effect of length, treatment and

stage on shape. Both treatment and stage were significantly associated with shape ($P = 0.002$ and $P = 0.001$, respectively); as expected, total length was not significantly associated with the size-corrected shapes ($P = 0.711$).

## Transgenic enhancer activity assay

Shark and skate *hoxa* enhancers were cloned into pCR8/GW/TOPO vector (Invitrogen) by PCR. A list of the primers is provided in Supplementary Table 14. The cloned enhancers were transferred into the pXIG-cfos-EGFP vector by Gateway LR Clonase II (Invitrogen)[53]. The created vectors were injected into one-cell-stage zebrafish eggs with *tol2* mRNA as previously described[125]. The injected embryos were observed under a stereo-type fluorescent microscope and photographed at 48 h after fertilization.

## Phylome reconstruction

The phylome of *L. erinacea*, meaning the collection of phylogenetic trees for each protein-coding gene in its genome, was reconstructed using an automated pipeline that mimics the steps that one would take to build a phylogenetic tree and based on the PhylomeDB pipeline[126]. First, a database with the proteomes (that is, full set of protein-coding genes) of 21 species was built that included *L. erinacea* (a full list of species included is provided in Supplementary Table 1). A BLASTp search was then performed against this database starting from each of the proteins included in the *L. erinacea* genome. BLAST results were filtered using an *e*-value threshold of $1 \times 10^{-5}$ and a query sequence overlap threshold of 50%. The number of hits was limited to the best 250 hits for each protein. A multiple sequence alignment was performed for each set of homologous sequences. Three different programs were used to build the alignments (Muscle (v.3.8.1551)[127], mafft (v.7.407)[128] and kalign (v.2.04)[129]) and the alignments were performed in forward and in reverse, resulting in six different alignments. From this group of alignments, a consensus alignment was obtained using M-coffee from the T-coffee package (v.12.0)[130]. Alignments were then trimmed using trimAl v1.4.rev15 (consistency-score cut-off 0.1667, gap-score cut-off 0.9)[131]. IQTREE (v.1.6.9)[132] was then used to reconstruct a maximum-likelihood phylogenetic tree. Model selection was limited to five models (DCmut, JTTDCMut, LG, WAG, VT) with freerate categories set to vary between 4 and 10. The best model according to the BIC criterion was used. Then, 1,000 rapid bootstrap replicates were calculated. A second phylome starting from *D. rerio* was also reconstructed according to the same approach. All trees and alignments are stored in phylomedb[126] with phylomeIDs 247 for the *L. erinacea* phylome and 275 for the *D. rerio* phylome (http://phylomedb.org).

## Species tree reconstruction

A species tree was reconstructed using a gene concatenation approach. The trimmed alignments of 102 protein families with a single orthologue per species were concatenated into a single multiple-sequence alignment. IQTREE[132] was then used to reconstruct the species tree using the same parameters as above. The final alignment contained 48,958 positions. The model selected for tree reconstruction was JTTDCMut+F+R5. Moreover, duptree[133] was used to reconstruct a second species tree using a super tree method. Duptree searches for the species tree that minimizes the number of duplications inferred when each gene is reconciled with the species tree. All trees built during the phylome reconstruction process were used to reconstruct this species tree. The two topologies were fully congruent.

## Skate MethylC-seq library preparation

MethylC-seq library preparation was performed as described previously[134]. In brief, 1,000 ng of genomic DNA extracted from the embryonic stage 31 and adult skate pelvic and pectoral fins was spiked with unmethylated λ phage DNA (Promega). DNA was sonicated to ~300 bp fragments using the M220 focused ultrasonicator (Covaris)

with the following parameters: peak incident power, 50 W; duty factor, 20%; cycles per burst, 200; treatment time, 75 s. Sonicated DNA was then purified, end-repaired using the End-It DNA End-Repair Kit (Lucigen) and A-tailed using Klenow fragment (3′→5′ exo-) (New England Biolabs) followed by the ligation of NEXTFLEX Bisulfite-Seq Adapters. Bisulfite conversion of adaptor-ligated DNA was performed using the EZ DNA Methylation-Gold Kit (Zymo Research). Library amplification was performed using KAPA HiFi HotStart Uracil+ DNA polymerase (Kapa Biosystems). Library size was determined using the Agilent 4200 Tapestation system. The libraries were quantified using the KAPA library quantification kit (Roche).

## Skate methylome data analysis

Embryonic stage 31 and adult skate pelvic and pectoral fin DNA methylome libraries were sequenced on the Illumina HiSeq X platform (150 bp, paired-end). Elephant shark *C. milii* raw whole genome bisulphite sequencing data (adult liver) were downloaded from NCBI BioProject (PRJNA379367)[135]. Zebrafish *D. rerio* raw whole genome bisulphite sequencing data (adult liver) were downloaded from the GEO (GSE122723)[136]. Sequenced reads in FASTQ format were trimmed using the Trimmomatic software (ILLUMINACLIP:adapter.fa:2:30:10 SLIDINGWINDOW:5:20 LEADING:3 TRAILING:3 MINLEN:50). Trimmed reads were mapped to the Leri_hhj.fasta genome reference (containing the lambda genome as chrLambda) using WALT[137] with the following settings: -m 10 -t 24 -N 10000000 -L 2000. Mapped reads in SAM format were converted to BAM format; BAM files were sorted and indexed using SAMtools[138]. Duplicate reads were removed using Picard Tools (v.2.3.0; http://broadinstitute.github.io/picard/). Genotype and methylation bias correction were performed using MethylDackel (MethylDackel extract Leri_hhj_lambda.fasta $input_bam -o $output --mergeContext --minOppositeDepth 5 --maxVariantFrac 0.5 --OT 10,110,10,110 --OB 40,140,40,140) (https://github.com/dpryan79/MethylDackel). Methylated and unmethylated calls at each genomic CpG position were determined using MethylDackel (MethylDackel extract Leri_hhj_lambda.fasta $input_bam -o output --mergeContext). DNA methylation profiles at differentially accessible ATAC–seq peaks between embryonic pelvic and pectoral fin samples were generated using deepTools2 computeMatrix reference-point and plotHeatmap[139].

## Reporting summary

Further information on research design is available in the Nature Portfolio Reporting Summary linked to this article.

## Data availability

Raw and processed sequencing data were deposited at the Gene Expression Omnibus (GEO; GSE188980 and GSE190730) and SRA (PRJNA783899). Mouse hindlimb RNA-seq data used for comparative analyses are publicly available at the GEO (GSE104459) and mouse forelimb RNA-seq data at the GEO (GSE136437). Zebrafish and elephant shark bisulphite sequencing data used for comparison were downloaded from NCBI BioProject (PRJNA379367) and the GEO (GSE122723136), respectively. Skate RNA-seq data are publicly available at NCBI BioProject (PRJNA288370 and PRJNA686126).

## Code availability

Code used is available at GitLab (https://gitlab.com/skategenome).

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

**Acknowledgements** We thank R. Schneider, D. Sherwood and the staff of the Marine Biological Laboratory Embryology course for providing laboratory space; L. Bertrand and the staff at Leica Microsystems for microscopy support; D. Remsen, S. Bennett, D. Calzarette, and the staff of the Marine Biological Laboratory and MBL Marine Resources Center for technical and animal husbandry assistance; A. Gillis for support and advice with RNA-seq and skate

functional experiments; and A. Shindo for technical support of image analysis. T.N., D.N. and A.A. were supported by institutional support provided by the Rutgers University School of Arts and Sciences and the Human Genetics Institute of New Jersey, a Whitman Center Fellowship (Marine Biological Laboratory) and the National Science Foundation under grant no. 2210072. D.N. was further supported by the NIH-IRACDA funded INSPIRE program at Rutgers University; F.M. and D.S.R. by funding from the Okinawa Institute for Science and Technology; D.S.R. by the Marthella Foskett-Brown Chair in Computational Biology; D.G.L. and R.D.A. by a grant from the Deutsche Forschungsgemeinschaft (LU 242672-1) and by a Helmholtz ERC Recognition Award grant from the Helmholtz-Gemeinschaft (ERC-RA1045 0033); R.D.A. and C.P. by EMBO Postdoctoral Fellowships (EMBO ALTF 537-2020 and ALTF 346-2020, respectively); P.M.M.G. by a postdoctoral fellowship from Junta de Andalucía (DOC_00397); J.J.T. and J.L.G.-S. by the European Research Council (ERC, grant no. 740041) and the Spanish Ministerio de Economía y Competitividad (grant no. PID2019-103921GB-I00); J.D. by the NIH grant HG003143; F.M. by the Royal Society (URF\R1\191161); V.A.S. by a Wolfson College Junior Research Fellowship and Marine Biological Laboratory Whitman Early Career Fellowship; J.L.-R. by the Spanish Ministerio de Ciencia e Innovacion (PID2020-113497GB-I00); and A.V. and F.D. by NIH grants R01DE028599 and R01HG003988. Research conducted at the E.O. Lawrence Berkeley National Laboratory was performed under US Department of Energy contract DE-AC02-05CH11231, University of California. J.D. is an investigator of the Howard Hughes Medical Institute.

**Author contributions** J.L.G.-S., F.M., J.J.T., T.N., D.S.R. and D.G.L. conceived the study and designed the experiments. F.M., E.d.l.C.-M., N.S. and J.L.G.-S. coordinated the sequencing of the little skate genome and F.M. assembled and annotated the genome. R.D.A., P.M.M.-G., L.Y., J.H.G., J.D. and D.G.L. performed analyses on 3D chromatin organization. F.M. and D.S.R. designed and performed synteny and comparative analyses. M.M.H., F.M. and T.G. performed phylogenetic and phylogenomic analyses. E.d.l.C.-M., C.P., S.N., R.D.A., J.J.T., I.C., L.G.-F., I.S. and J.L.-R. performed and analysed transgenics, ATAC–seq, RNA-seq and Hi-C experiments. V.A.S. and C.H. performed and analysed additional RNA-seq experiments. F.D. and A.V. performed additional functional assays. K.S., P.E.D., A.G.-R. and O.B. performed and analysed DNA methylation experiments. D.N., A.A. and T.N. conducted embryonic experiments of skates and sharks. J.L.G.-S., J.J.T., F.M., D.S.R. and D.G.L. wrote the manuscript with input from all of the authors.

**Competing interests** J.D. is on the scientific advisory board of Arima Genomics and of Omega Therapeutics. The other authors declare no competing interests.

## Additional information
**Correspondence and requests for materials** should be addressed to Ferdinand Marlétaz, Tetsuya Nakamura, Juan J. Tena, Darío G. Lupiáñez or Daniel S. Rokhsar.

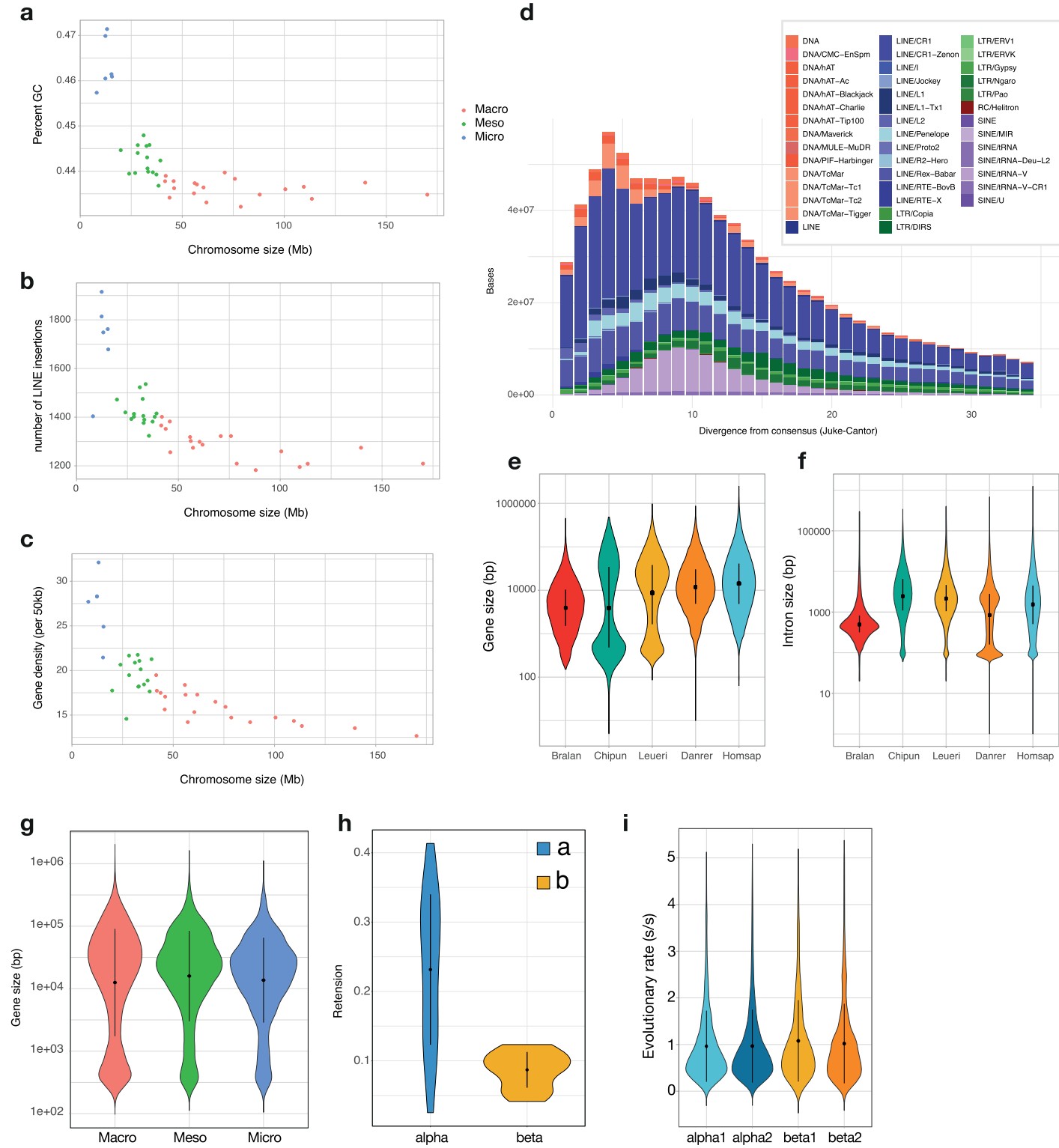

**Extended Data Fig. 1 | Characteristics of skate chromosomes, repeat content and skate genome. a–c**, Characteristics and classification of skate chromosomes according to their size (x-axis) and GC% (a), number of LINE insertions (b) and gene density (c) per 50kb window. **d**, Repetitive landscape computed as JC divergence of repeat occurrence toward the consensus element in the repeat library. **e–f**, Distribution of gene and intron size in selected chordate species: amphioxus (*Branchiostoma floridae*, Bralan), the cloudy catshark (*Chiloscyllium punctatum*, Chipun), the little skate (*Leuraja erinacea*, Leueri), the zebrafish (*Danio rerio*, Danrer) and human (*Homo sapiens*, Homsap). **g**, gene size distribution in three chromosomal categories. **h**, distribution of retention rates inferred for CLGs in the spotted gar (*Lepisosteus oculatus*). **e–h**, mean dot) and standard deviation (bar) are indicated with the violin plot area. **i**. Rates of evolution of genes located in α or β segments estimated as ML distance to the amphioxus outgroup (LG+Γ).

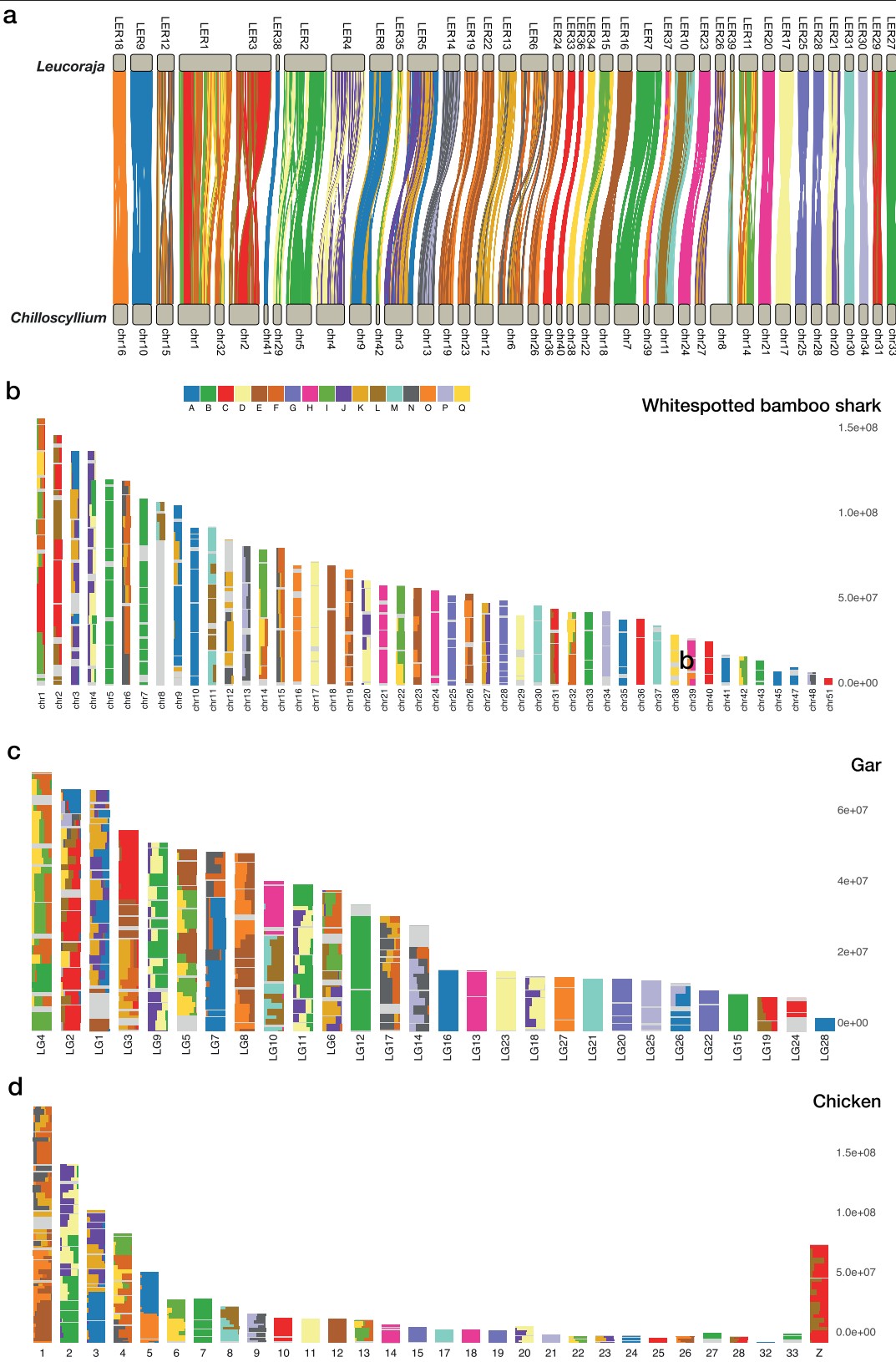

**Extended Data Fig. 2 | Chromosomal architecture and synteny conservation in cartilaginous and bony fishes. a**, Syntenic orthology relationship between skate and bamboo shark highlighting the conservation of chromosomal architecture among chondrichtyans. **b–d**, Organisation of segments derived from each CLG in bamboo shark, gar and chicken genome using the same colour code as in Fig. 1. Each bin along chromosomes represents 20 genes.

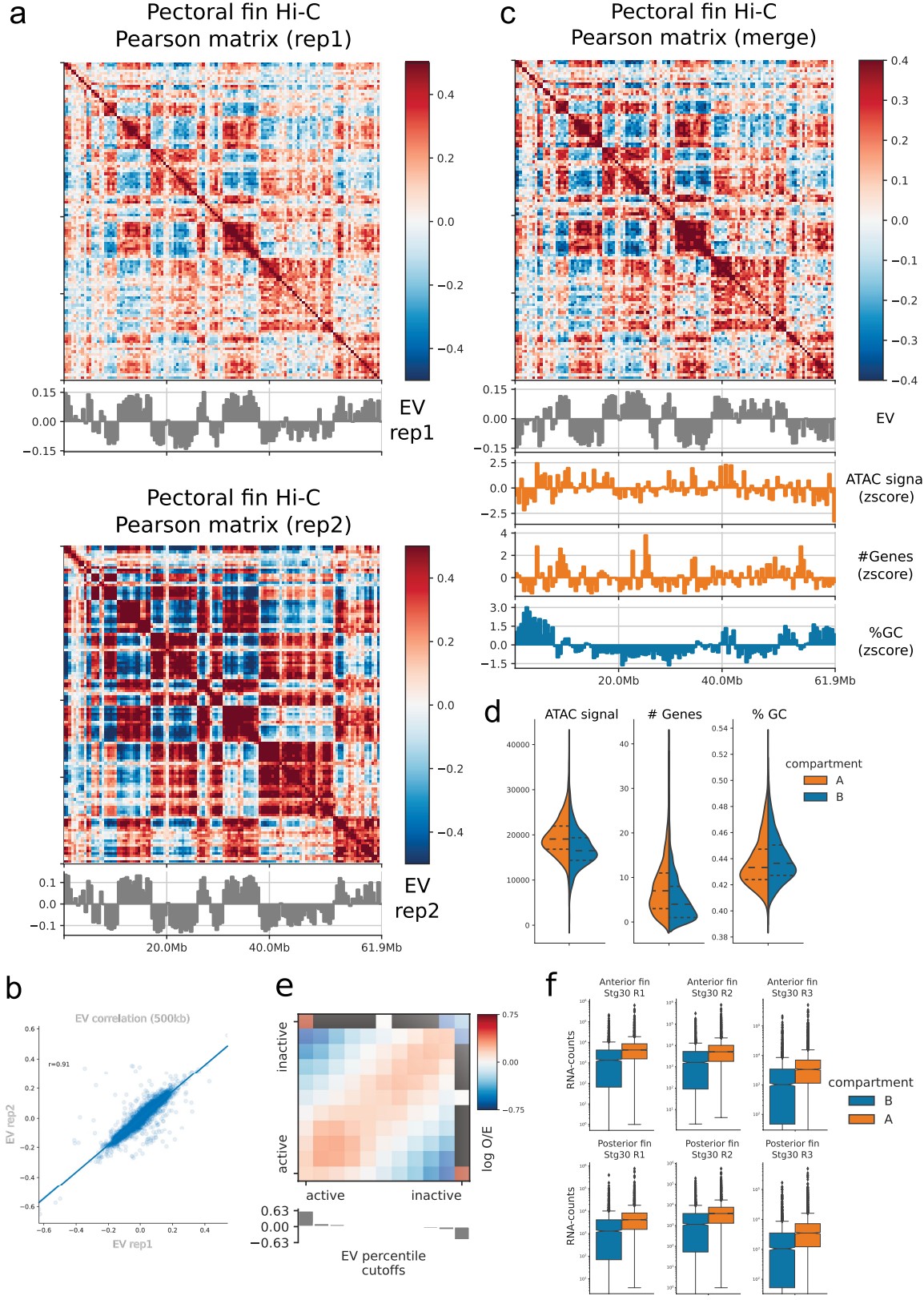

**Extended Data Fig. 3** | See next page for caption.

**Extended Data Fig. 3 | The skate genome is organized in A/B compartments.**
**a.** 500 kb resolution Pearson matrices of a representative chromosome (Leri_11C) and their associated eigenvectors showing marked compartmentalization in A/B compartments in both replicates. **b.** Eigenvector correlation among the two replicates. **c.** Merged Pearson matrix presented together with its eigenvector, the normalized signal for ATAC-seq in anterior pectoral fin, the number of gene models and the percentage of GC content. As shown in **d**, the A compartment in skates correlates with chromatin accessibility and the number of gene models, but no clear correlation was observed with the GC content.

**e.** Saddle plot demonstrating the aggregated enrichment of homotypic A-A and B-B interactions. **f.** Gene expression in either the A or the B compartment as measured with bulk RNA-seq performed in the anterior and posterior portions of the skate pectoral fin at Stg 30. Top: anterior, bottom: posterior, n = 4046 bins of 500kb (A compartment n = 2125, B compartment n = 1921). Boxes correspond to the median and the first and third quartiles (Q1 and Q3). Whiskers extend to the last point within 1.5 times the interquartile range below and above Q1 and Q3, respectively.

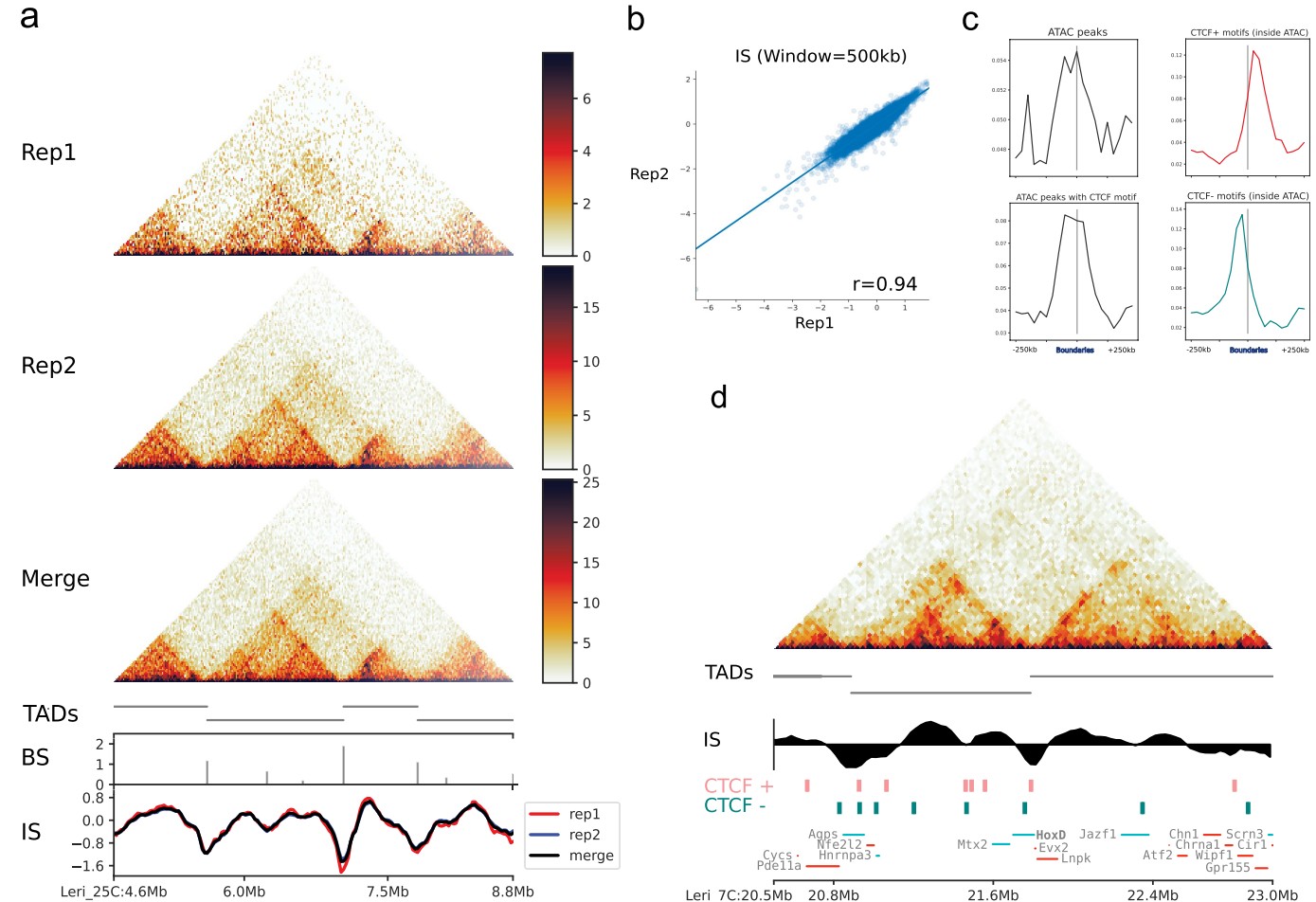

**Extended Data Fig. 4 | Skate chromosomes are organized in TADs flanked by convergent CTCF sites. a**. Hi-C interaction matrices in skate pectoral fins in either of the two replicates and the merge (25kb resolution). The TAD calling performed in the merged matrix and the associated boundary scores (BS) and insulation scores (IS) are shown below (window size of 500kb). **b**. Insulation score correlations between the two replicates. **c**. From top to bottom, enrichment around TAD boundaries (+-250kb) of ATAC-seq peaks and ATAC-seq peaks containing the CTCF motif regardless of the strand, in the plus and in the minus strand. **d**. Hi-C matrix around the HoxD locus showing the conserved bipartite configuration in two TADs with HoxD genes located precisely at the boundary. TADs, insulation scores and ATAC-seq peaks containing the CTCF motif are shown. The tendency of having divergent CTCF sites at insulation minima is observable.

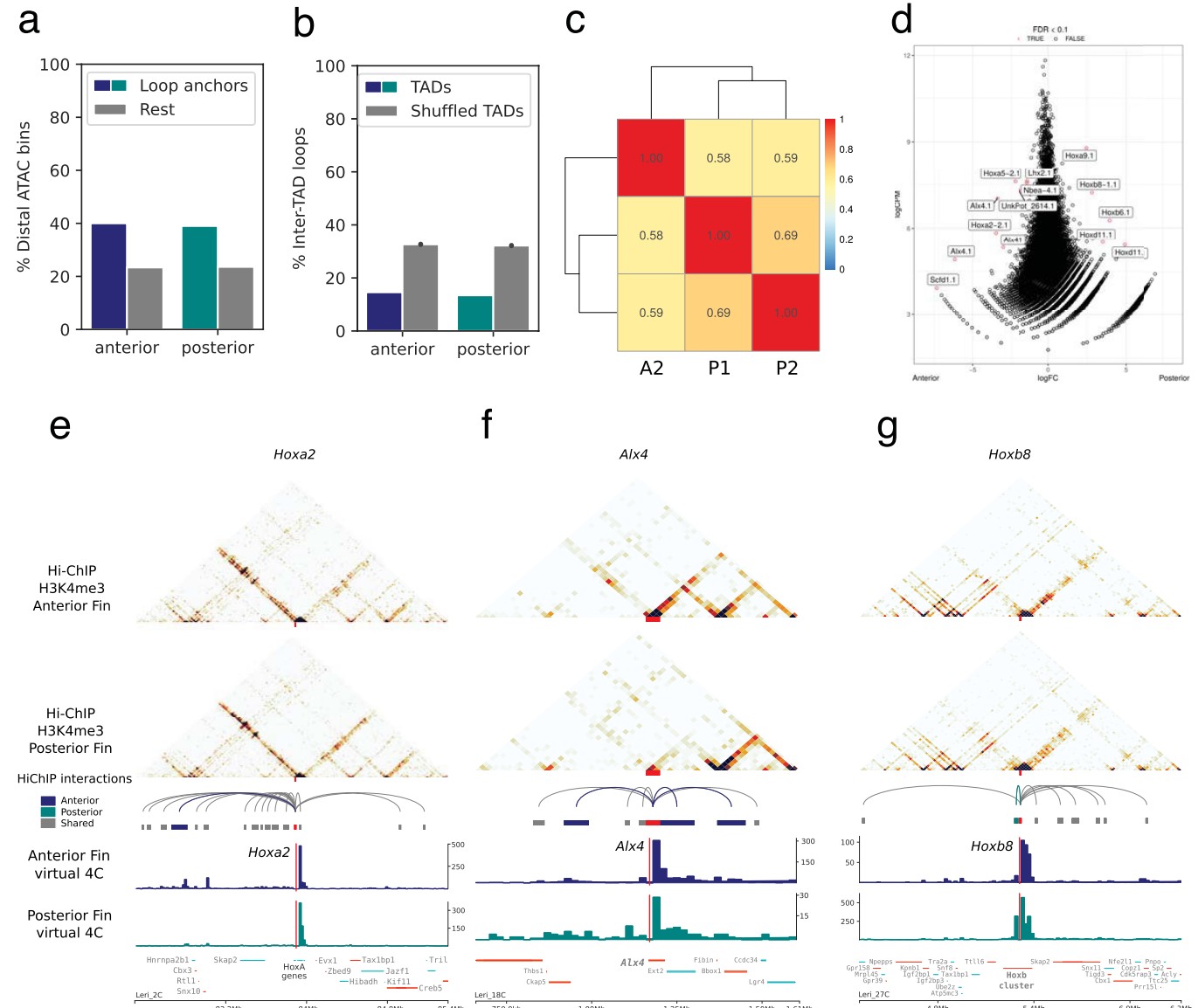

**Extended Data Fig. 5 | H3K4me3 HiChIP unveils the regulatory landscapes of active genes in the anterior and posterior portions of the skate pectoral fin. a**. Proportion of distal loop anchors that also correspond to distal ATAC-seq peaks in the pectoral fin in both the anterior and posterior H3K4me3 HiChIP datasets. **b**. Proportion of inter-TAD interactions calculated in the anterior and posterior HiChIP datasets compared to a random shuffling of the TADs (grey). **c**. Spearman correlation of the three valid replicates (1 for anterior and 2 for posterior fins). The correlation between the matrices is limited to the non-redundant set of interactions (union = 50,601 interactions). **d**. Differential loop analysis derived from read counts in **c**. logFC *vs*. logCPM plot with significant differential loops highlighted in red. **e,f**. Anterior specific contacts in the *Hoxa2* and *Alx4* regulatory landscape (dark blue). **g**. Posterior specific contacts in the *Hoxb8* regulatory landscape (turquoise).

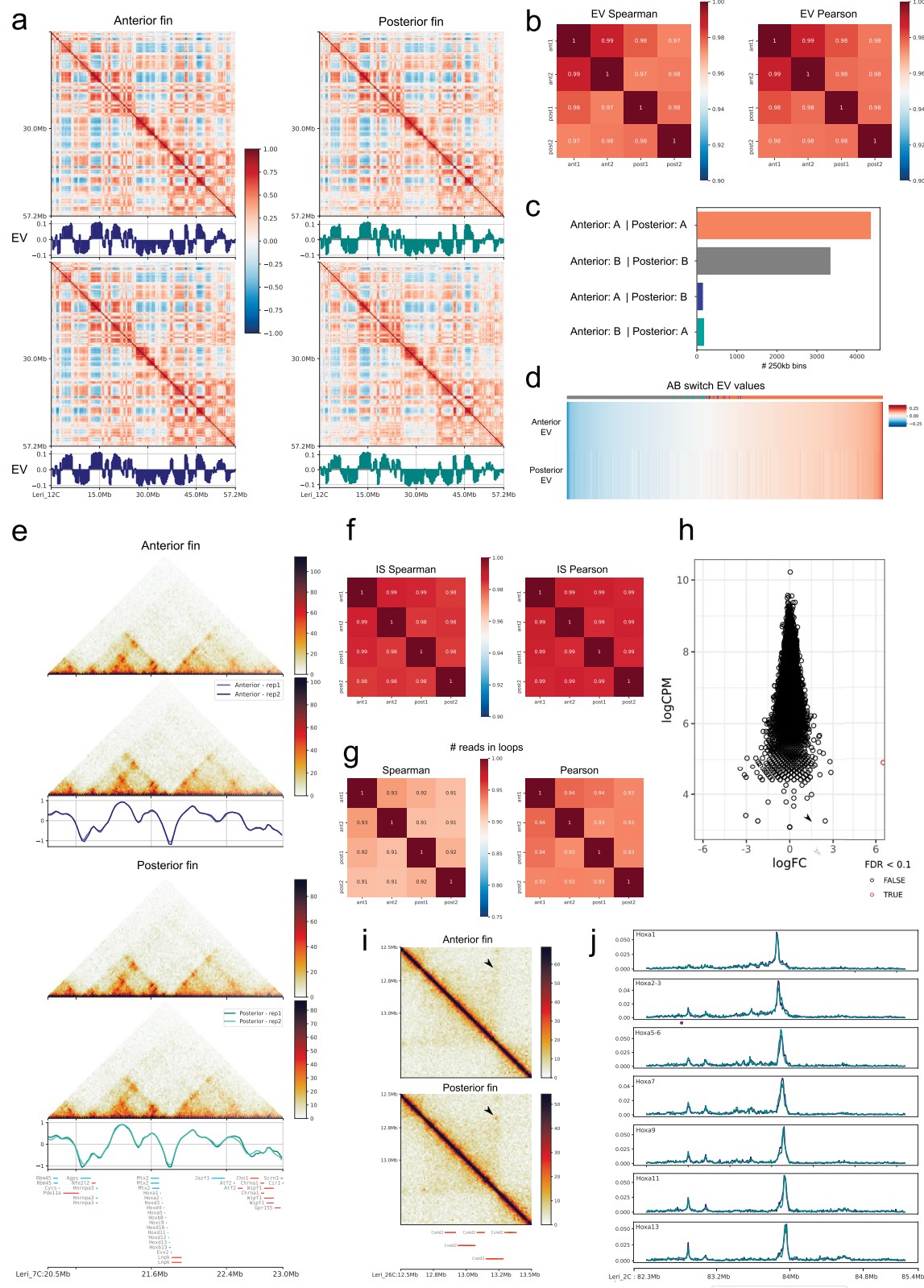

**Extended Data Fig. 6** | See next page for caption.

**Extended Data Fig. 6 | Preformed 3D chromatin folding in anterior *vs*. posterior fin. a**. Pearson matrices and eigenvectors showing A/B compartmentalization of the chromosome Leri_12C of skates in the anterior and posterior portions of the pectoral fin. **b**. Genome-wide eigenvector correlations. **c**. Quantification of A/B compartment switches between anterior and posterior portions of the fin. **d**. Comparison of all EV values between anterior and posterior fin. Heatmaps are sorted according to anterior EV values and compartment switches are indicated in the colour bar on top. Most switches are concentrated towards the centre, where EV values are intermediate. **e**. Comparison of insulation scores and overall TAD structures around the HoxD locus. **f**. Genome wide insulation score correlations. **g**. Correlations of number of reads found inside a consensus set of loops consisting of the union of the loops (see Methods) **h**. Differential loop analysis derived from read counts in **g**. logFC *vs*. logCPM plot with the only significant differential loop highlighted in red. **i**. Snapshot of the Hi-C heatmap around the only significant differential loop located in the *Csmd2* locus. Arrowheads indicate the position of the loop. **j**. Virtual 4C-seq profiles of Hoxa cluster genes derived from the Hi-C experiments. Few differences are appreciated, and no differences are evident in contacts between Hoxa2 and the differential loop predicted by HiChIP (purple asterisk).

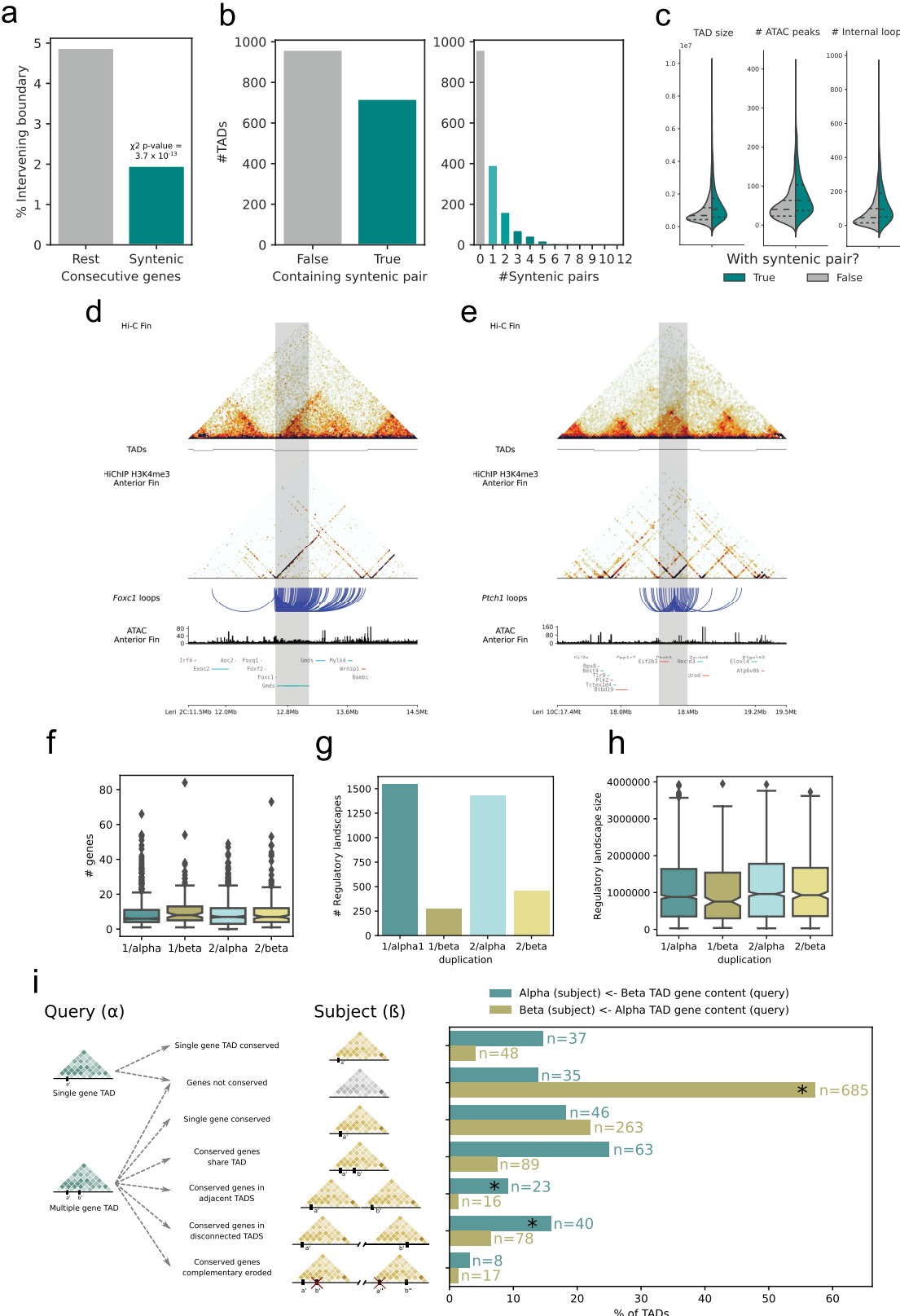

**Extended Data Fig. 7** | See next page for caption.

**Extended Data Fig. 7 | Conservation of vertebrate TADs after the Whole Genome Duplications. a**. Intergenic spaces between microsyntenic pairs conserved across vertebrates (present in skate and osteichthyes, here mouse and garfish) are devoid of TAD boundaries. Syntenic gene pairs n = 3017, non-syntenic n = 25386. Two-sided $\chi^2$ p-value = 3.7 x 10$^{-13}$ **b**. 40% of skate TADs contain a deeply conserved microsyntenic pair. Several of them contain more than one association. **c**. TADs containing deeply microsyntenic associations are bigger, contain more ATAC-seq peaks and more loops as defined using HiChIP (Syntenic TAD n = 718, non-syntenic TAD n = 960). *Foxc1/Gmds* (**d**) and *Ptch1/Eif2b3* (**e**) are examples of deeply conserved microsyntenic associations. Microsyntenic area is shaded in grey. Hi-C, TADs, HiChIP and ATAC-seq data are shown along with the gene tracks. **f**. Gene content of TADs associated to the different paralogous segments of the genome originated after the two rounds of WGD (1 or 2 for the 1R, alpha or beta for the 2R) Boxes correspond to the median and the first and third quartiles (Q1 and Q3). Whiskers extend to the last point within 1.5 times the interquartile range below and above Q1 and Q3, respectively. **g**. Number of regulatory landscapes (defined as the group of interactions anchored by a single gene promoter) belonging to the different paralogous segments of the genome originated after the two rounds of WGD (1 or 2 for the 1R, alpha or beta for the 2R). **h**. Regulatory landscape sizes observed in the paralogous segments of **f** defined as the genomic space spanning from the two more distal loop anchors anchored to a given promoter. Boxplots defined as in **f**. **i**. The fate of the counterparts of alpha TADs was investigated in the beta copy and *vice versa*. TADs with more than one gene conserved allowed us to infer scenarios of TAD fissions-fusions in either or the genome copies. Asterisks (*) highlight complete TAD losses in beta (yellow bar) and TAD fission events in alpha (blue bars).

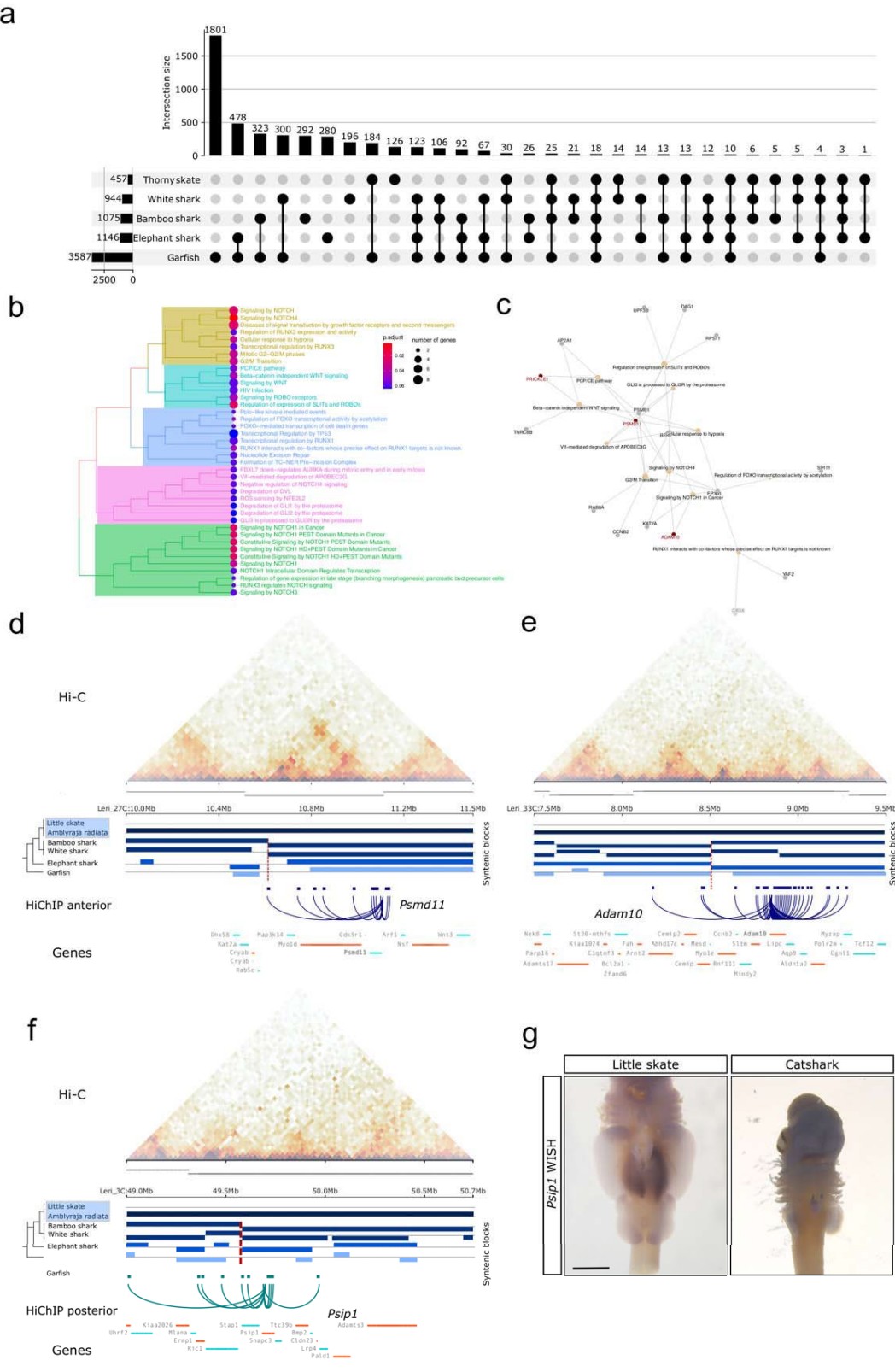

**Extended Data Fig. 8** | See next page for caption.

**Extended Data Fig. 8 | Rearranged TADs in the skate lineage involve PCP-related genes. a**. Extended version of the upset plot presented in Fig. 4a with the quantification of synteny breaks detected in different vertebrate species using the skate genome as a reference. The barplot on top shows the quantification of synteny breaks for the species combination indicated by the dots below. The barplot on the left shows the total quantification of synteny breaks for each individual species. **b**. ReactomePA[119] clustering of significant terms found in the set of candidate genes for regulatory rearrangements in the anterior pectoral fin. P-values are BH corrected p-values obtained with a one-sided Fisher test for term overrepresentation (ReactomePA default). A selection of these terms is shown in Fig. 4c. **c**. Cnetplot showing the relationship of candidate genes with each of the different enriched terms. **d**. Candidate rearrangement at the *Psmd11* locus, implicated in the PCP pathway. Pectoral fin Hi-C map is shown on top together with the TAD predictions. Below, the synteny blocks that are shared with the different species studied and the candidate synteny break is highlighted in red. Finally, arachnogram with the contacts devised from the anterior fin H3K4me3 HiChIP experiment. **e**. Same as in **d**, but for the Notch-signalling related gene *Adam10*. **f**. Same as in **d** and **e** but for the Hox activator *Psip1*. Note that this time the presented H3K4me3 HiChIP is from posterior pectoral fins. **g**. Whole mount *in situ* hybridization against *Psip1* in both the little skate *L. erinacea* and the catshark *S. retifer* shows species-specific expression of *Psip1* in the anterior portion of the skate pectoral fins. *n* = 5 for skates and sharks. The scale bar corresponds to 2 mm.

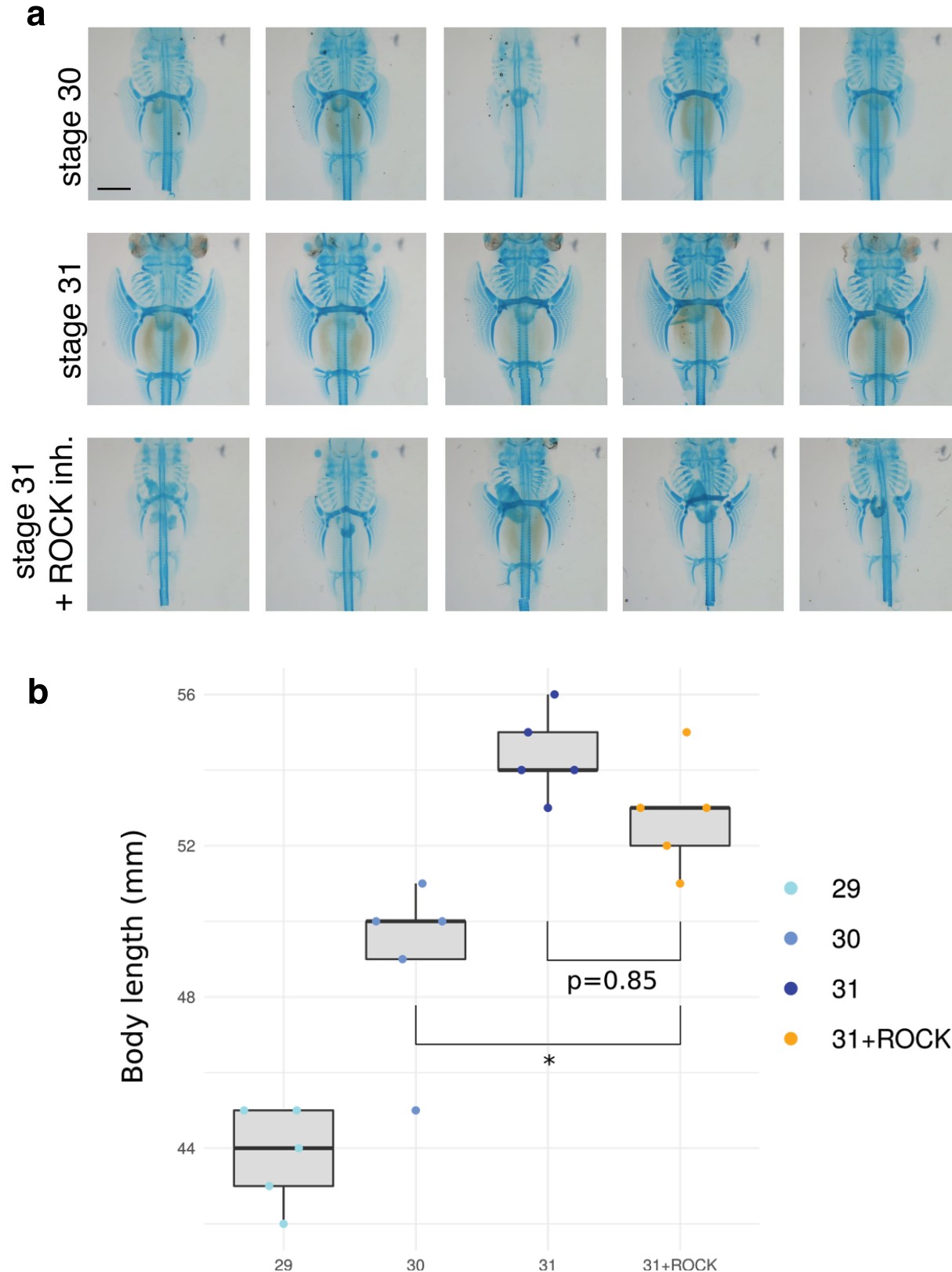

**Extended Data Fig. 9** | See next page for caption.

**Extended Data Fig. 9 | Fin ray development in control and ROCK inhibitor-treated skate embryos. a**. Cartilages in control (stages 30 and 31) and ROCK inhibitor-treated embryos (stage 31) were examined by Alcian blue staining. Five replicates for each condition are shown. The whole-mount staining showed that anterior fin ray development is affected by ROCK inhibitor-treatment with some variations. The number of fin rays attached to propterygium (pro), mesopterygium (meso), and metapterygium (meta) was counted under a stereomicroscope and statistically analysed (Fig. 4). The scale bar is 2 mm. **b**. The total body length of control and ROCK-treated skate embryos. The total body length of control (stages from 29 to 31) and ROCK inhibitor-treated embryos (stage 31). Note that the body length of ROCK inhibitor-treated embryos is longer than stage 30 embryos (* = Bonferroni corrected two-sided t-test p-value = 0.01232), indicating that the embryos with the inhibitor normally developed, and the pectoral fin phenotype was not due to the overall defects of body development. Five replicates for each condition were examined and body length distributions were assumed to be normal. The minima, maxima, and median values of the box and whisker plots of stage 29, 30, 31, and ROCK inhibitor-treated embryos are 42, 45, and 44, 49, 51, and 50, 53, 56, and 54, 51, 55, and 53, respectively. Boxes correspond to the median and the first and third quartiles (Q1 and Q3). Whiskers extend to the last point within 1.5 times the interquartile range below and above Q1 and Q3, respectively.

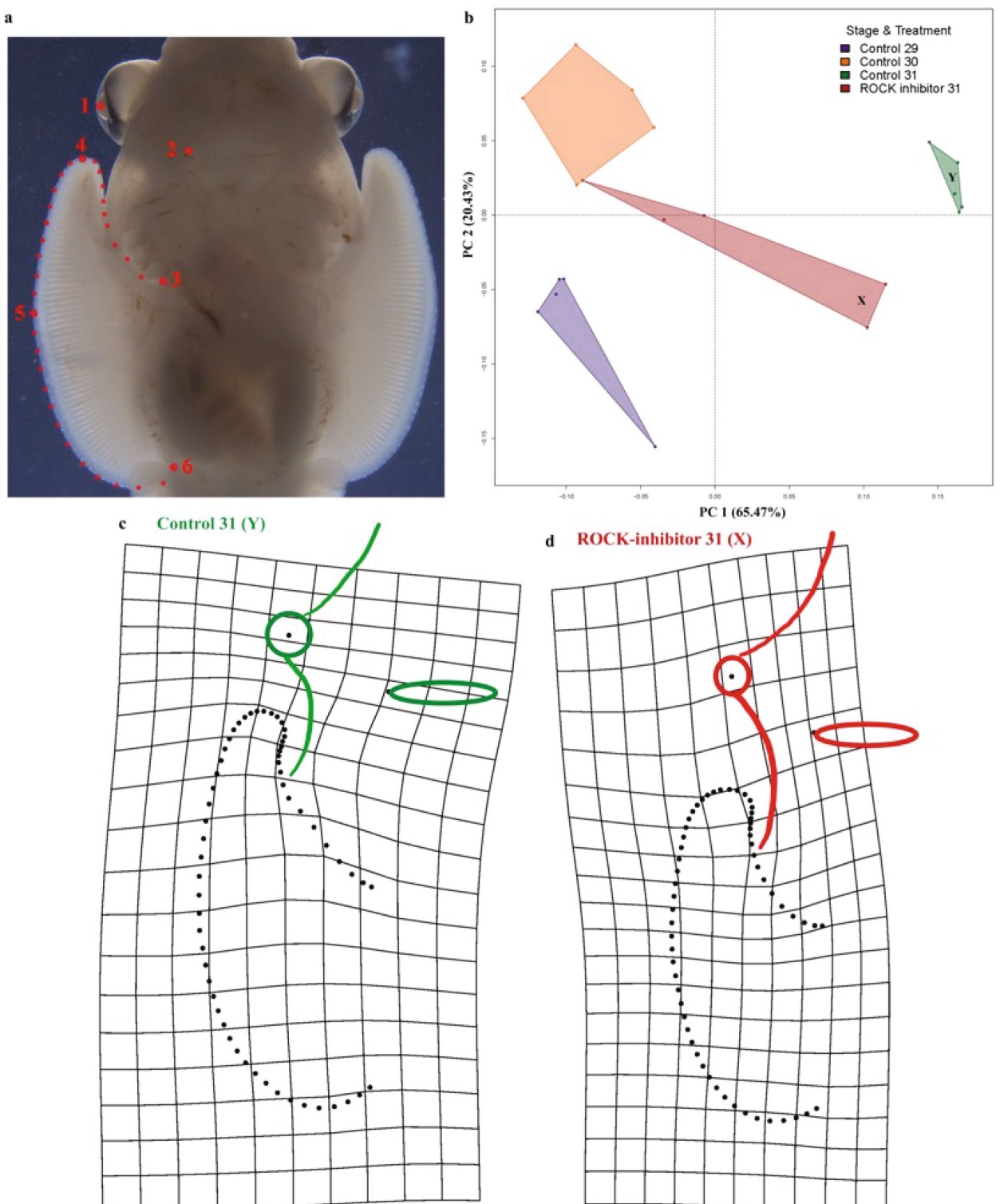

**Extended Data Fig. 10 | Geometric morphometric analyses of the inhibition of the PCP pathway using a rho-kinase (ROCK) inhibitor in stage 31 skate embryos. a**. Schematic of the landmark design used in these analyses, including both landmarks (numbered red points) and semi-landmarks (small red points). **b**. Principal components analysis shows that specimen shapes cluster by treatment and stage. Points X and Y were used to generate the deformation grids showing the shape changes between the area of the PCA plot dominated by control (**c**) and ROCK-inhibited specimens (**d**). Note the inhibition of growth on the anterior region of the pectoral fin in the ROCK-inhibited specimens.

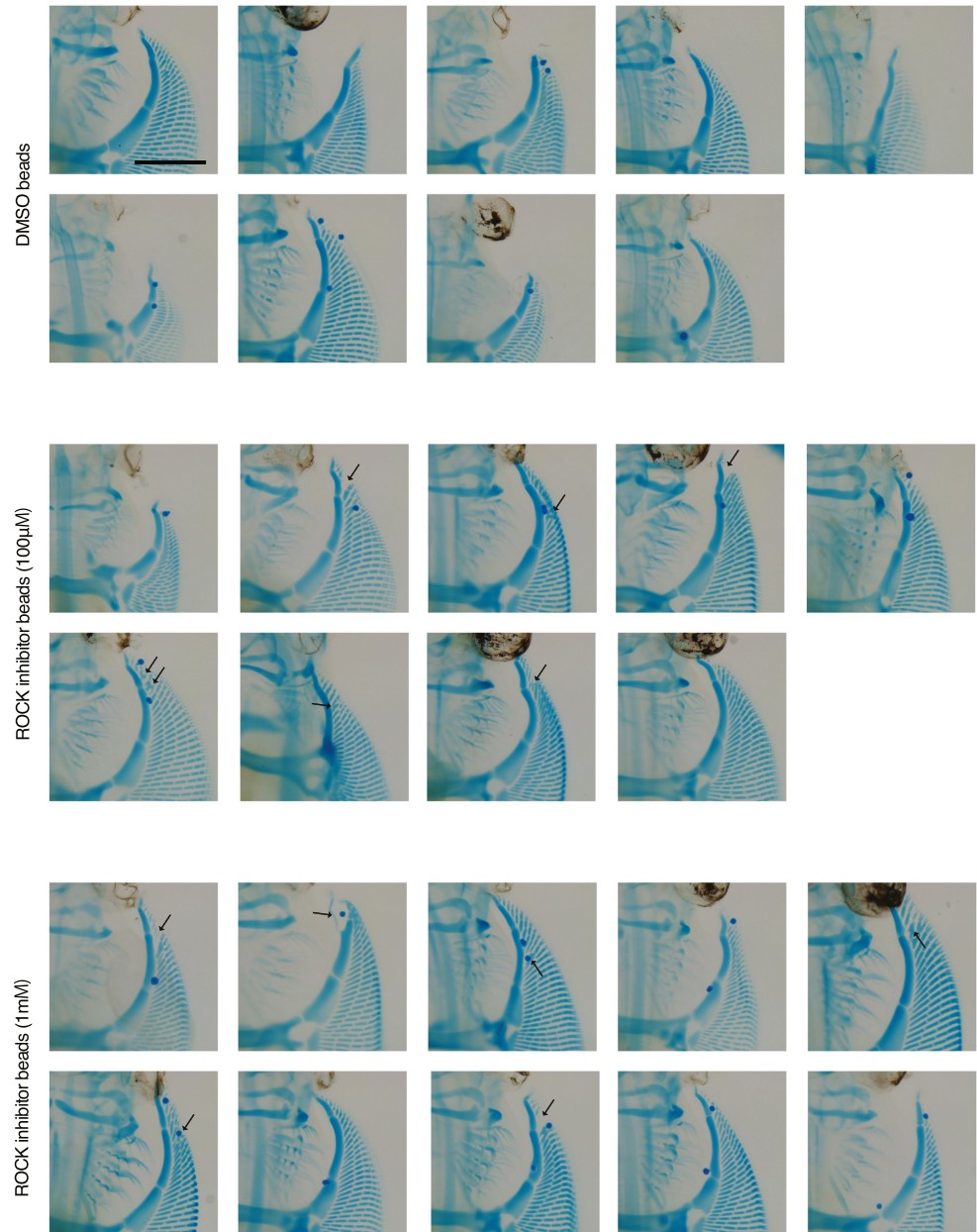

**Extended Data Fig. 11 | Cartilage staining of DMSO or the ROCK inhibitor-beads implanted skate embryos at stage 31.** The beads were repeatedly implanted into the anterior part of the right pectoral fin every two weeks (the total is three times) from stage 29. Some beads were retained until stage 31 (blue dots), while others fell during the treatment The embryos with the ROCK-inhibitor beads exhibited fusion, loss, or disorganized fin ray patterning (arrows, 6/9 for 100 μM and 6/10 for 1 mM). Note that abnormal fin ray patterning was never observed in control animals, indicating that the effects not directly associated with a bead in treated embryos were likely derived from the loss of the bead during the treatment. *N* = 9 for DMSO, 9 for 100 μM inhibitor, and 10 for 1 mM inhibitor beads. The scale bar is 2 mm.

## a

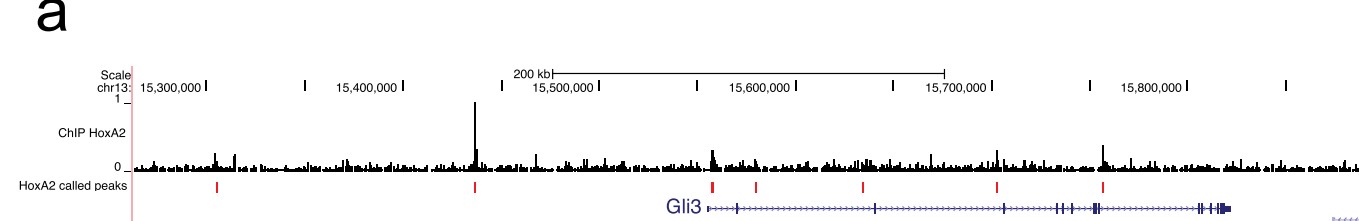

## b

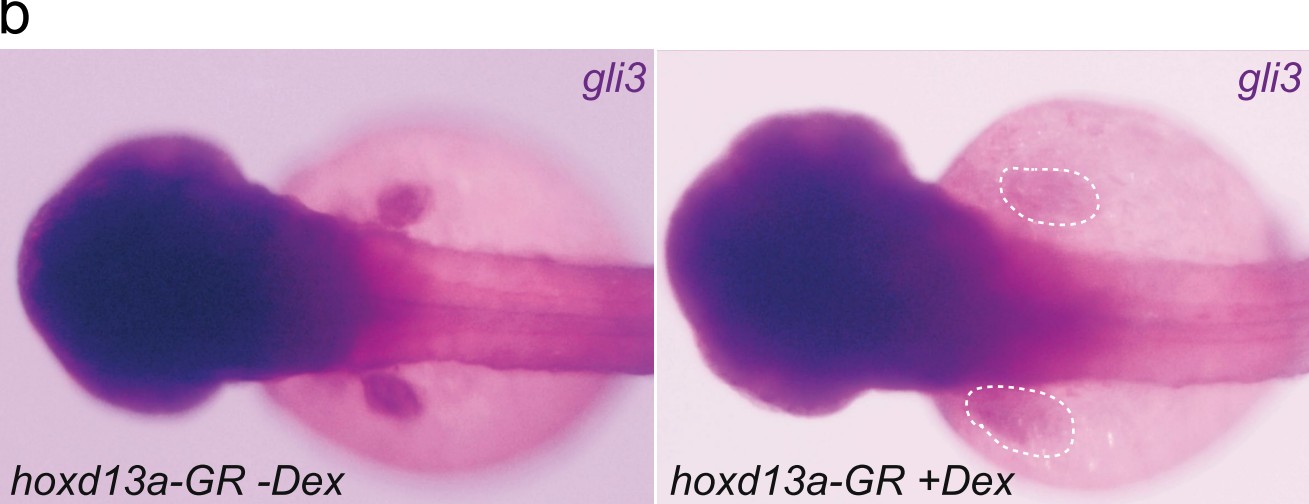

*gli3*

*gli3*

*hoxd13a-GR -Dex*

*hoxd13a-GR +Dex*

**Extended Data Fig. 12 | Genetic interactions among *Hox* and *Gli3* genes.**
**a**. ChIP-seq experiment in mouse embryonic branchial arches performed in
Amin et al. 2015, which shows the binding profile of HoxA2 to *Gli3* genomic
*locus*. **b**. Whole-mount in situ hybridization of *gli3* in zebrafish embryos
inyected with a *hoxd13a-GR* mRNA. Developing fins are indicated with red
arrowheads. In the absence of dexamethasone (left panel), the construct is
inactive and the embryos develop normally (50 out of 57, 88%). Upon treatment
with dexamethasone (right panel), *hoxd13a* is activated and causes a reduction
of *gli3* expression at the developing fin region (mild reduction in 39 out of 93,
42%; strong reduction in 22 out of 93, 24%). Scale bars = 250 µm.

# Reporting Summary

## Statistics

For all statistical analyses, confirm that the following items are present in the figure legend, table legend, main text, or Methods section.

| n/a | Confirmed | |
|---|---|---|
| ☐ | ☒ | The exact sample size (*n*) for each experimental group/condition, given as a discrete number and unit of measurement |
| ☐ | ☒ | A statement on whether measurements were taken from distinct samples or whether the same sample was measured repeatedly |
| ☐ | ☒ | The statistical test(s) used AND whether they are one- or two-sided<br>*Only common tests should be described solely by name; describe more complex techniques in the Methods section.* |
| ☒ | ☐ | A description of all covariates tested |
| ☐ | ☒ | A description of any assumptions or corrections, such as tests of normality and adjustment for multiple comparisons |
| ☐ | ☒ | A full description of the statistical parameters including central tendency (e.g. means) or other basic estimates (e.g. regression coefficient) AND variation (e.g. standard deviation) or associated estimates of uncertainty (e.g. confidence intervals) |
| ☐ | ☒ | For null hypothesis testing, the test statistic (e.g. *F*, *t*, *r*) with confidence intervals, effect sizes, degrees of freedom and *P* value noted<br>*Give P values as exact values whenever suitable.* |
| ☒ | ☐ | For Bayesian analysis, information on the choice of priors and Markov chain Monte Carlo settings |
| ☒ | ☐ | For hierarchical and complex designs, identification of the appropriate level for tests and full reporting of outcomes |
| ☐ | ☒ | Estimates of effect sizes (e.g. Cohen's *d*, Pearson's *r*), indicating how they were calculated |

*Our web collection on statistics for biologists contains articles on many of the points above.*

## Software and code

Policy information about availability of computer code

| Data collection | No software was used for data collection |
|---|---|
| Data analysis | The following public software was used for data analysis:<br><br>- Genomescope (v. 2.0)<br>- Megahit (v. 1.1.1)<br>- dbg2olc (c. 10037fa)<br>- Minimap2 (v. 2.12)<br>- Racon (v. 1.3.1)<br>- BUSCO (v. 3.0.2)<br>- purge_haplotigs (v. 1.0.2)<br>- HiRise (v. 2.0.5)<br>- Juicebox (v. 2.1.10)<br>- HiGlass Browser (v. 1.11.7)<br>- PBjelly (PBSuite v. 15.8.24)<br>- STAR (v. 2.5.2b)<br>- stringtie (v. 1.3.3)<br>- TACO (v. 0.7.3)<br>- Trinity (v. 2.8.4)<br>- GMAP (v. 2018-07-04)<br>- Mikado (v. 1.2.1) |

- Transdecoder (v. 5.5.0)
- Augustus (v. 3.3.3)
- portcullis (v 1.2.0)
- exonerate (v 2.2.0)
- Repeatmasker (v. 4.0.7)
- PASA (v. 2.5.0)
- OMA (v. 2.4.1)
- HMMer (v. 3.1b2)
- MAFFT (v. 7.3)
- BMGE (v. 1.12)
- IQTREE (v. 2.1.1)
- Phylobayes (v. 4.1e)
- Mmseq2 (v. 13-45111)
- Broccoli (v. 1.2)
- GeneRax (v. 2.0.4)
- BWA (v. 0.7.17)
- pairtools (v. 0.3.0)
- Juicer Tools (v. 1.13.02)
- FAN-C (v. 0.9.1)
- FIMO (v. 4.11.2)
- Clover (https://github.com/mcfrith/clover, 5ca3e81725)
- profileplyr (v. 1.13.0)
- HiCExplorer (v. 3.7.2)
- EdgeR (v. 3.36.0)
- TADbit (v 1.0)
- bedGraphToBigWig (kentUtils v4)
- MACS2 (v. 2.2.7)
- FitHiChIP (v. 9.0)
- Bedtools (v. 2.26.0)
- GenomicRanges (v. 1.44.0)
- Nextflow (v19.10.0)
- nf-core/rnaseq (v. 1.4)
- DeSeq2 (v. 1.30.1)
- TopGO (v. 2.42.0)
- nf-core/atacseq (v1.0.0)
- gtfToGenePred (kentUtils v4)
- lastz (v. 1.04.15)
- chainCleaner (https://github.com/hillerlab/GenomeAlignmentTools v. 971d043)
- ReactomePA (v. 1.38.0)
- ClusterProfiler (v. 4.2.2)
- Fiji (v. 20191028-2046)
- Stereomorph (v. 1.6.1)
- ShinyGM (v. 9.11.21)
- Muscle (v. 3.8.1551)
- mafft (v. 7.407)
- kalign (v. 2.04)
- T-coffee (v. 12.0)
- trimAl (v. 1.4.rev15)
- IQTREE (v. 1.6.9)
- duptree (v. 1.48)
- Trimmomatic (v. 0.32)
- WALT (v. 1.01)
- Samtools (v. 1.3)
- Picard (v. 2.3.0)
- MethylDackel (v. 0.6.1)
- deepTools (v. 3.5.0)

Custom code is available at:

https://gitlab.com/skategenome
hic_pipe.py / filt2hic.sh (https://gitlab.com/rdacemel/hic_ctcf-null)

For manuscripts utilizing custom algorithms or software that are central to the research but not yet described in published literature, software must be made available to editors and reviewers. We strongly encourage code deposition in a community repository (e.g. GitHub). See the Nature Portfolio guidelines for submitting code & software for further information.

## Data

Policy information about availability of data

All manuscripts must include a data availability statement. This statement should provide the following information, where applicable:

- Accession codes, unique identifiers, or web links for publicly available datasets
- A description of any restrictions on data availability
- For clinical datasets or third party data, please ensure that the statement adheres to our policy

Raw and processed sequencing data were deposited in GEO (GSE188980 and GSE190730) and SRA (PRJNA783899). Mouse hindlimb RNA-seq data used for comparative analyses are publicly available under GEO accession number GSE104459 and mouse forelimb RNA-seq data under GEO accession number GSE136437. Zebrafish and elephant shark bisulfite sequencing data used for comparison were downloaded from PRJNA379367 and GSE122723136 accession codes respectively. Skate anterior and posterior pectoral fins RNA-seq data are publicly available under BioProject accession code PRJNA288370.

## Human research participants

Policy information about studies involving human research participants and Sex and Gender in Research.

| | |
|---|---|
| Reporting on sex and gender | *Use the terms sex (biological attribute) and gender (shaped by social and cultural circumstances) carefully in order to avoid confusing both terms. Indicate if findings apply to only one sex or gender; describe whether sex and gender were considered in study design whether sex and/or gender was determined based on self-reporting or assigned and methods used. Provide in the source data disaggregated sex and gender data where this information has been collected, and consent has been obtained for sharing of individual-level data; provide overall numbers in this Reporting Summary. Please state if this information has not been collected. Report sex- and gender-based analyses where performed, justify reasons for lack of sex- and gender-based analysis.* |
| Population characteristics | *Describe the covariate-relevant population characteristics of the human research participants (e.g. age, genotypic information, past and current diagnosis and treatment categories). If you filled out the behavioural & social sciences study design questions and have nothing to add here, write "See above."* |
| Recruitment | *Describe how participants were recruited. Outline any potential self-selection bias or other biases that may be present and how these are likely to impact results.* |
| Ethics oversight | *Identify the organization(s) that approved the study protocol.* |

Note that full information on the approval of the study protocol must also be provided in the manuscript.

# Field-specific reporting

Please select the one below that is the best fit for your research. If you are not sure, read the appropriate sections before making your selection.

☒ Life sciences        ☐ Behavioural & social sciences        ☐ Ecological, evolutionary & environmental sciences

For a reference copy of the document with all sections, see nature.com/documents/nr-reporting-summary-flat.pdf

# Life sciences study design

All studies must disclose on these points even when the disclosure is negative.

| | |
|---|---|
| Sample size | Two replicates were used for Hi-C experiments as it is widely recommended in the field (see recent Rao et al. 2017, Cell; Franke et al. 2020, Nature Communications; Valton et al. 2022, Nature Structural & Molecular Biology). Two replicates were also used for HiChIP (Franke et al. 2020) and ATAC-seq (see Marlétaz et al. 2017, Nature) following a similar rationale. Three different catshark embryo pectoral fins were used for RT PCR detection of Prickle1 as it is widely accepted to devise robust statistics. Five replicates were used for the whole mount ISH as it is widely accepted in the field of cartilaginous fish research (see Marconi et al. eLife 2020). For enhancer testing in zebrafish, in F0, GFP positive embryos were selected after transgenesis mix injection, and the ratio of GFP fin positive vs. GFP fin negative embryos was calculated (5/18 for the skate enhancer, 0/31 for the shark enhancer). Regarding the F1 stable transgenic lines, they were considered as stable lines when three independent founders with the same GFP expression pattern were found. For the validated Hox overexpression line 57 embryos for the control experiment and 93 embryos for the Dexamethasone treatment were used, which is in the order of previously used in other similar studies (Tena et al., Dev. Biol. 2007 301:518-31; Freitas et al. Dev. Cell 2012 23:1219-29). These numbers guarantee the robustness and reliability of statistical analyses. For the cell elongation analyses, about 400-550 cells were analyzed in each domain of the pectoral fin (the precise numbers are in the figure caption) as 50-200 cells are typically analyzed as replicates in PCP analysis (see Butler and Wallingford 2018). These numbers were enough to show statistically significant differences among samples. Five replicates were used for the whole body inhibitor treatment as it is widely accepted in the field of cartilaginous fish research (see Marconi et al. eLife 2020, for example). For the beads implantation experiments, we prepared 9-10 replicates for each condition as the locations of the implantation may vary due to the manual surgery. Despite a certain amount of variation, the results showed an obvious difference with/without the inhibitor (6/10 embryos vs 0/10 embryos showed aberrant fins, respectively). For bisulfite sequencing, as per Burger et al NAR 2013, we used single replicates of skate MethylC-seq data (at coverage > 10X), to identify UMRs and |

|  |  |
|---|---|
| | LMRs. |
| Data exclusions | An experiment with two replicates was originally designed for the HiChIPs of the anterior portion of the pectoral fin, but one of the replicates' quality was not high enough and we had to discard it. Given the reliability of the technique and the redundancy and overlapping with the newly presented high-resolution HiC data in anterior and posterior pectoral fins, we decided not to repeat this lost replicate. |
| Replication | All experiments were performed at least in two replicates, refer to sample size section for details. |
| Randomization | Randomization was not directly employed since we were analyzing developmental processes and phenotypes in skate, catshark and zebrafish embryos. No external covariates are expected and since embryos are collected from many different parental individuals, randomization strategies are not obvious and they are not considered necessary with animal models like skates and zebrafish. |
| Blinding | Blinding was not relevant for our study since all comparisons were performed automatically using statistical software not influenced by the investigator. |

# Reporting for specific materials, systems and methods

We require information from authors about some types of materials, experimental systems and methods used in many studies. Here, indicate whether each material, system or method listed is relevant to your study. If you are not sure if a list item applies to your research, read the appropriate section before selecting a response.

## Materials & experimental systems

| n/a | Involved in the study |
|---|---|
| ☐ | ☒ Antibodies |
| ☒ | ☐ Eukaryotic cell lines |
| ☒ | ☐ Palaeontology and archaeology |
| ☐ | ☒ Animals and other organisms |
| ☒ | ☐ Clinical data |
| ☒ | ☐ Dual use research of concern |

## Methods

| n/a | Involved in the study |
|---|---|
| ☒ | ☐ ChIP-seq |
| ☒ | ☐ Flow cytometry |
| ☒ | ☐ MRI-based neuroimaging |

## Antibodies

| | |
|---|---|
| Antibodies used | We have used anti-Histone H3 (tri methyl K4) antibody - ChIP Grade from Abcam (ab8580) for the HiChIPs experiments |
| Validation | This antibody has been tested by the commercial company in experiments of chromatin immunoprecipitation, among others, in cow and human, but it has been predicted to work in many other species of vertebrates, invertebrates and even plants. In our lab we have successfully performed ChIP-seq and HiChIP experiments with this antibody in several species like zebrafish, amphioxus or sea urchin. |

## Animals and other research organisms

Policy information about studies involving animals; ARRIVE guidelines recommended for reporting animal research, and Sex and Gender in Research

| | |
|---|---|
| Laboratory animals | This study did not require the use of adult animals. Skate and elephant shark (Scyliorhinus retifer) embryos were obtained from the Marine Resource Center at Marine Biological Laboratory (MA, USA) and were used for each experiment at Marine Biological Laboratory and Rutgers University. Zebrafish embryos were obtained from AB and Tübingen strains at the fish facility of Centro Andaluz de Biología de Desarrollo (Seville, Spain). Sex was not determined since it is unfeasible for the embryos analyzed. Developmental stages are always stated throughout the manuscript and figure captions. |
| Wild animals | The study did not involve wild animals |
| Reporting on sex | Sex was not determined since it is unfeasible for the embryos analyzed. Developmental stages are always stated throughout the manuscript and figure captions. |
| Field-collected samples | The study did not involve field-collected samples. |
| Ethics oversight | Experiments with skate embryos were performed at the Marine Biological Laboratory and Rutgers University under these protocols: MBL IACUC protocol #18-36 and Rutgers IACUC protocol #201702646. Zebrafish procedures were reviewed and approved by the Ethical Committees from the University Pablo de Olavide, CSIC, and the Andalusian government, and performed in compliance with all relevant ethical regulations. |

Note that full information on the approval of the study protocol must also be provided in the manuscript.

