## [Peer Review File · Nature]

Manuscript Title: The little skate genome and the evolutionary emergence of wing-like fin appendages.

Reviewer Comments & Author Rebuttals

Reviewer Reports on the Initial Version:

Referees' comments:

Referee #1 (Remarks to the Author):

Summary of the key results

This work is a magnum opus of genome sequencing, functional genomics, computational analysis, comparative embryology, and evodevo. The group analyzed the little skate, *Leucoraja erinacea*, a member of the Batoida. The batoids, or rays, comprise the largest extant group of cartilaginous fishes and are distinguished from their sister group, the sharks (Selachii) by their ventrally-placed gills and a greatly flattened body that includes a highly modified pectoral fin unlike that of any other vertebrate. In order to understand this major evolutionary transition (which allowed adaptation to a benthic life history), the group subjected the little skate to all manner of genomic, transcriptomic and embryological analyses. While the transition to a benthic lifestyle involved many anatomical and physiological adaptations, the group focused primarily on the evodevo of the modified pectoral fin. This work used an extensive armamentarium of genomic and computational tools to make testable inferences regarding the deep evolutionary genomic events that led to the formation of the batoid pectoral fin. The findings are spectacular in scope. They described the genome, gene content, chromosome compositions, linkage group comparisons (including those before and after the whole genome duplication events), and really honed down on the genome dynamics within the developing skate pectoral fin. They were able to compile these data to infer strongly that early chromosomal rearrangements – traceable via methods described in recent work on conservation and evolution of chordate linkage groups as well as the genomics of vertebrate polyploidization – altered the TAD genetic regulatory landscape substantially so that new developmental programs arose in the batoids, including the deployment of PCP genes in the anterior pectoral fin and the finding of a likely 2nd organizer for the batoid pectoral fin. The conclusions from the genomic and computational analyses could be experimentally validated using a variety of embryological experiments (ISH, pharmacological inhibitor for PCP) and surrogate expression analysis (zebrafish).

Originality and significance

This is a highly original report that will impact the evodevo field substantially and will also raise the bar for investigators conducting future genomics-inspired biological research. This paper stands out for its totality of approaches. And even though the work could be published as several stand-alone papers, the sum of the findings is much greater than the parts. Kudos to the team for working together to generate such an immense and comprehensive dataset. The use of [chordate] linkage group evolution to pinpoint chromosomal breakpoints and to tie this with local chromatin conformational changes that may have given rise to innovations in gene regulation and major evolutionary transitions, is a powerful approach. It is an approach that will surely be used in addressing age-old problems in evodevo over the next years.

Data & methodology

There is so much data in this paper, from the genome assembly to all the functional genomics and transcriptomic data, to the developmental biology experiments. While all the data are presented in a concise manner in the extended data file, those data are quite dense and cannot be fully validated without rerunning the analyses (which I could not do). So I needed to rely on the analyses by the authors (my mea culpa). In most cases, the data look appropriate and the aesthetically pleasing graphical presentations/summaries are appreciated. For non-genomics types, however, I do feel that some of the figure legends should be more explanatory, e.g., the data in Fig. 4A (and ED Fig. 9A) are not intuitive and a clearer explanation in the legends could help a lot.

Suggested improvements: experiments, data for possible revision

The surrogate expression data in Figure 5d is not easy to interpret due to the large amount of GFP signal in the head. Is it possible to improve on that image? And how many embryos were examined in those experiments?

The Prickle1 ISH in the far anterior pectoral fin of skate (Fig. 5e) is unexpected and the staining is beautiful. Is there also expression in the anterior pelvic fin? Not that it lessens the impact of the paper but their exclusion but were other elasmobranch taxa also included here? Just asking because PCP involvement in anterior wing development is a major finding of this paper.

References: appropriate credit to previous work?

Highly appropriate. A lot of the key references are from pioneering work that the group has previously published on sensitive chromatin confirmation analysis, TADS, deep chordate linkage relationships, skate developmental biology, etc.

Clarity and context: lucidity of abstract/summary, appropriateness of abstract, introduction and conclusions

This is a very clearly written manuscript despite the highly complex nature of the datasets and biological problem. I think the authors could have been more bold in their speculation about the role of PCP in the evodevo of the wing, but they were cautious.

Queries about text

In MAIN, p4, line 5: While wing-like appendages are in skates, they define the batoids. That should probably be batoids instead of skates per se.

"Homology" is used somewhat loosely in the text. In some places it didn't appear clear whether it was orthology or homology in the evolutionary sense.

1st line of p 18 – is "both pectoral fins" correct?

P 18, last line – cownose ray is not a skate, it is another batoid (a myliobatid)

I am fine with revealing my identity on this most excellent and impressive paper.

Chris Amemiya
University of California-Merced

Referee #2 (Remarks to the Author):

Summary

Skates are cartilaginous fish, characterized by enlarged wing-like pectoral fins. These fins have enabled skates to develop a particular swimming style that allows them to thrive at the bottom of seas. In this manuscript, Marlétaz et al. investigate the evolutionary origin of these wing-like fins. The authors present a new assembly of the genome of the little skate *Leucoraja erinacea*, which contains 19 macro chromosomes, 14 meso chromosomes and 7 micro chromosomes. Combined with synteny analyses, the authors show that the little skate genome represents an ancestral condition among jawed vertebrates, situated after the two rounds of whole genome duplication (WGD) that underlie the origin of vertebrates, but before the chromosome fusions that occurred in more distant vertebrates. This makes the little skate an interesting model from an evolutionary perspective. The authors next investigate the 3D organization of the little skate genome, by performing Hi-C in developing pectoral fins. They find that the genome is organized into compartments and TADs, similar to mammalian genomes. The authors find that TAD structures are constrained across evolution. However, they are not completely invariant and the authors suggest that rearrangements of TADs could have played an important role in the evolution of regulatory landscapes and gene expression patterns. To investigate additional regulatory mechanisms underlying the development of the skate's wing-like pectoral fins, the authors performed HiChIP, ATAC-seq and RNA-seq in the anterior and the posterior parts of the pectoral fins. Based on these analyses, the authors propose that the PCP pathway and Hox genes are important drivers of the morphology of skate fins, which they further validate with functional assays in skate and zebrafish.

Overall, I find this an interesting and comprehensive study, which contributes to our understanding of the evolution of the specialized morphology of skate fins and the general role of the 3D organization of the genome in the evolution of specific phenotypes.

Major comments

I generally find the data presented in this manuscript of high quality and convincing. However, I think it is unfortunate that the authors have chosen to use H3K4me3 HiChIP to analyze tissue-specific chromatin interactions in anterior and posterior regions of developing fins. The data seem very sparse, which makes it difficult to identify changes between the two tissue types. Furthermore, as H3K4me3 HiChIP preferentially detects interactions between regions enriched in H3K4me3, the interactions are not unbiased; it is therefore not clear whether differences in HiChIP interactions reflect differences in H3K4me3 enrichment or in interaction frequency.

In lines 301-306, the authors write "Differential analysis revealed similar looping patterns between the anterior and the posterior region of the fin (Pearson correlation >0.96 , with 9 and 5 interactions statistically enriched in the anterior and posterior fins respectively, Extended Data Fig. 7e, f). We found striking differences in promoter interaction between anterior and posterior fins at specific loci including the HoxA locus (Fig. 3d, Extended Data Fig. 7h, i)."

I find this a bit confusing. Does this mean that the HoxA locus is an example of a region containing 1 of the 9 interactions that are statistically different? In any case, the differences in Figure 3d do not seem striking to me. Similarly, the data presented in Figure 5 shows many empty bins and very limited data depth. Furthermore, the authors present 3 valid replicates – 1 for anterior (A1) and 2 for posterior (P1 and P2) fins and show the correlation in Extended Data Figure 7. Based on these analyses it seems that A2 and P2 correlate better than P1 and P2, which does not make sense to me. (It could be that I mis-understood the analyses performed here – I find the legends very confusing.)

I think that there are likely many interesting tissue-specific differences in chromatin interactions between the anterior and the posterior region of the fin for the authors to explore. However, I do not think they can be appreciated from the H3K4me3 HiChIP data, as they are sparse and contain

a H3K4me3 bias. It would be much more convincing if the authors could use Hi-C to analyze the differences between anterior and pectoral fins. Alternatively (or in addition), they could consider doing 4C experiments, a multiplexed Capture-C experiment or another high-resolution targeted 3C method (that does not introduce bias for a specific chromatin mark) for their regions of interest. I think that such experiments would improve this manuscript, since I do not think that the conclusions about tissue-specific 3D regulation are supported by the HiChIP data.

Minor comments

1. The GEO and GitHub links do not work.

2. Lines 167-170: "Skate micro-chromosomes also show a high degree of interchromosomal contacts compared with macro-chromosomes (Fig. 1d, e), as also found in snakes and other tetrapods, which suggests this is a general feature of vertebrate micro-chromosomes." This is indeed clear in the log-transformed observed/expected matrices in Figure 1d. However, I find it surprising that this is not visible in the matrix in Extended Data Figure 4. Can the authors comment on this?

3. Figure 3a. The stripe/bar-structure aligned with the diagonal that is contained within the meta-TAD seems like an unusual feature to me that is potentially of interest. It would be very interesting if the authors could explore this further.

4. Lines 266-267: "At the sub-megabase scale, the skate genome is organized into Topologically Associated Domains (TADs) with a median size of 800 Kb (Extended Data Fig. 6a, b)." It could be interesting to explore some further details about the TAD organization. The authors mention later in the manuscript that they identify 1,678 TADs. What proportion of the genome is covered by TADs? How does this compare to other vertebrates / mammals (based on Hi-C data of similar quality / depth and similar parameters to call TADs)?

5. Figure 3b/Extended Data Figure 6e. The authors show that CTCF motifs are enriched around TAD boundaries. It would be interesting to know what proportion of TADs show CTCF enrichment? How does that compare to other vertebrates / mammals? (Is there reason to think other mechanisms (in addition to cohesin/CTCF-mediated loop extrusion) are involved as well?)

6. Related to the previous comments, I assume that CTCF and cohesin subunits are conserved in *Leucoraja erinacea*? It would be helpful to mention this.

7. "Lines 294-297: "To further explore enhancer-promoter interactions, we used chromatin-conformation capture-immunoprecipitation (HiChIP) to associate H3K4me3-rich active promoters with potential regulatory loci in the anterior and posterior region of the developing pectoral fin, where HoxA and HoxD genes are expressed."

Lines 308-309: "Such interaction changes are consistent with the specific expression of HoxA genes in the anterior portion of the developing pectoral fin."

Lines 444-449: "Importantly, Hox genes also show distinctive differences between pectoral and pelvic fins (Extended Data Table 5) and between mouse and skate paired appendages (Extended Data Fig. 17). In particular, several 3' HoxA and HoxD genes are preferentially expressed in the anterior pectoral fin, while 5' HoxA and HoxD genes are located in the posterior pectoral domains, consistent with previous findings in two different skate species."

I find this a bit confusing; perhaps it would be helpful if the authors could be more clear and specific about the Hox expression patterns when they first mention the Hox genes.

8. Lines 335-336: "We find that although the size of TADs is similar between alpha and beta chromosomes, there are notably fewer TADs in the latter (Fig. 3e, f)."

Does this mean that the entire TAD regions have disappeared or did only the boundaries disappear? If the latter, is this driven by specific changes in CTCF motifs?

9. Figure 4/Extended Data Figure 10. The TAD rearrangement in Figure 4d is convincing; however the examples presented in Extended Data Figure 10 are much less convincing – perhaps this is also something that can be improved with high-quality targeted 3C data.

10. Line 489. I think the authors mean to refer to Extended Data Figures 18 and 21 (and not 18 and 20).

11. Figure 5C. There are some “gaps” in some of the ATAC peaks.

Referee #3 (Remarks to the Author):

In the manuscript “The little skate genome and the evolutionary emergence of wing-like fin appendages” Marlétaz, de la Calle-Mustienes, Acemel, Nakamura, Paliou and colleagues present a chromosome-level genome assembly for the little skate *Leucoraja erinacea*. They first use this new resource for comparative genomics approaches to study gnathostome genome evolution, before producing a whole range of functional genomics data focusing on the development of the skate's wing-like pectoral fin. The authors put a particular emphasis on the 3D-organization of the genome, and how skate-specific synteny breaks across TAD-boundaries could have resulted in the rewiring and/or emergence of novel regulatory landscapes. For two candidates resulting from these analyses, the PCP pathway and anterior Hox genes, the data is complemented by pharmacological inhibition experiments, geometric morphometric analyses and transgenic reporter assays in zebrafish, to evaluate their potential involvement in the development of an anteriorly expanded skate pectoral fin. Overall, the manuscript is clearly written and provides appropriate referencing of the literature.

The amount and quality of the genomics data should make this work of broad interest to the evolutionary genomics community, and will help to elevate the status of *Leucoraja erinacea* as an emerging model organism in comparative developmental biology and functional genomics. As outlined below, though, certain aspects in the authors' analyses need further clarification. Also, the logic for choosing candidate genes and pathways for follow-up experiments is not clear to me, and not all data resulting therefrom – at least in their current form – seem to fully support the conclusions the authors wish to draw.

- Genome ‘content’: in the text, the authors classify 7 chromosomes as ‘micro chromosomes’ – is the 20Mb cut-off due to a particular convention? Chromosome size distribution in Fig. 2a seems continuous, with LER33 and LER34 right around the 20Mb limit. Hi-C contacts (6 ‘outliers’) and GC % (6 ‘outliers’) would argue for a different classification (Fig. 1d,e), as would LINE content (5 ‘outliers’) or gene density (4 ‘outliers’) in Fig. S2b, c. Related to that, can the bi-modal distribution in gene sizes in Fig. S2e (Chipun and Leueri) be explained by shorter genes being overrepresented on micro chromosomes? Likewise, are certain functional classes of genes – e.g. housekeeping genes vs. developmental regulators – overrepresented on micro chromosomes? This would be interesting to know, given the overall denser 3C organization of micro chromosomes and the regulatory interferences potentially resulting therefrom. Lastly, the number of gene losses mentioned in the text (354) and in Fig. S1c (616) do not match.

- Genome evolution: the authors propose that the “skate genome closely resembles that of the most recent jawed vertebrate common ancestor”. Since no real outgroup exists, i.e. a non-gnathostome with a 2R genome, the authors base their argument on chromosome-scale ancestral chordate linkage group (CLG) content, synthetic orthology relationships and gene retention rates. First, in Fig. 2a, it would help the reader if the same color scheme (and size information) for the CLGs would be used as in Fig. 2 of Simakov et al., 2020 NatEcoEvo. Second, is there a way to quantify and make this comparable between species? E.g., total length or % of continuous CLG conservation per chromosome, and compare such metrics for the skate and the 4 species in Simakov et al.? For Fig. 2b, is there another chondrichthyan genome of sufficient quality available,

to include in this analysis and thus balance the phylogenetic representation? For Fig. 2c, please include corresponding gene retention rates for at least one representative osteichthyan.

- 3D chromatin organization: why do the authors think that the anterior and posterior fin have "similar looping patterns", when over 50% appear to be compartment-specific (Fig. S7e)? Also, the difference between Pearson and Spearman correlation seems to indicate that the former is dominated by outliers (Fig. S7f,g). Maybe the normalization/scaling of the data should be re-considered? Minor comment for line 328: for clarity's sake, maybe quickly reiterate which results actually suggest the conservation of the two genome organization mechanisms, e.g. presence of A/B compartments and the orientation bias of CTCF sites?

- Choice for follow-up experiments: synteny breaks and HiChIP data identify a total of 180 genes, whose regulatory interactions might have changed due to skate-specific rearrangements. Following a signaling pathway enrichment analysis, the authors first decide to focus on Prickle1, while seemingly ignoring(?) other candidates from the PCP pathway, their apparent top candidate NOTCH signaling, or other previously implicated pathways (e.g. GLI3, see Nakamura et al. 2015) – why? Likewise, the study of a putative Hoxa2 enhancer is partly justified based on the handpicked(?) example of Psip1. Psip1, however, was reported to activate posterior HoxA genes, via the lncRNA Hottip. Is Hottip present and transcribed in the skate genome? Also, for their Psip1 argument, the authors probably want to cite the 2017 paper of Pradeepa et al., not the 2012 one. In their current form, these two choices thus appear somewhat arbitrary.

- PCP follow-up experiments: why did the authors chose ROCK inhibition, which clearly impacts cellular processes beyond PCP, and why did they administer it systemically, rather than via beads (see Nakamura et al. 2015). To me, 'st.31 + ROCK inh.' embryos shown in Fig. S15 appear closer in overall size and e.g. head width to st.30 control embryos, than st.31. Moreover, ray numbers should also be quantified at st.29, i.e. when treatment started > does ray number not increase after st.29, i.e. upon ROCK inhibition, or are some rays "lost" during the treatment? Geometric morphometric analyses and PCA also seem to suggest that 'st.31 + ROCK inh.' embryos are stuck somewhere on the 'morphogenetic trajectory', going from st.29 to st.31, which could also indicate a general developmental delay upon ROCK inhibition? Lastly, in Fig. S12, the mean axis ratio, or a boxplot, should be included on the scatter plots, and representative images used for the measurements should be provided.

- Hox follow-up experiments: in Fig. 5a, why are there anterior/orange and blue/posterior highlights in the pectoral fin and the mouse forelimb? If I understand the experiment correctly, the comparative RNA-seq analysis was performed between entire(?) pectoral and pelvic fins of the skate, and fore-/hindlimbs from a mouse data set? In Nakamura et al. and Fig. S17a,d we do find an A/P differentiation of the samples, but what corresponding mouse data set would have been used for this? The entire workflow and data sources here are not properly explained, neither in the main text nor the material and methods section. Additionally, Extended Data Table 5 in the Excel file made available to me does not show the differentially expressed genes. More importantly, however, the entire logic for a) using an inducible, highly expressed(?) Hoxd13a transgene to study the effect of endogenous Hoxa2 levels on Gli3 transcription, and b) testing a putative Hoxa2 enhancer in zebrafish, appears flawed to me. If present in the genome, skate Hoxd13 would presumably (see Fig. S17d, Hoxd10-12) be expressed posteriorly, i.e. where Gli3 is expressed. Hoxa13, on the other hand, appears to be expressed anteriorly (Fig. 5b, although I could not find a similar trend for it in Fig. S17d?). What could be the reason for this anterior Hoxa13 pattern, e.g., any clues from the HiChIP data? And why is Hoxa2 still considered the prime candidate for Gli3 downregulation, given the known impact of Hoxa13 on Gli3 expression? (also, see 'Psip1>Hottip>5'HoxA' link, mentioned above) For the enhancer test, the current images without close-up of the fin do not allow to see a potential anterior enrichment of reporter expression in the zebrafish fin, and the skate HoxA2 image appears overexposed (see e.g. fore- and hindbrain signal, or trunk signal). Moreover, were these F0 embryos and what number of replicates were produced and analyzed? Lastly, given the AER-like scenario proposed for the anterior skate fin in Nakamura et al., how do the authors envision this skate enhancer to function in a zebrafish fin, given that the trans-regulatory environment is not skate-like?

Minor comments

- Line 85: Fig. 1b, not a?
- Line 345: "...for other mammalian traits" reads like the skate would also be a mammal?
- Line 461: typo > enhancers
- Line 489: Extended Data 21, not 20?
- Please make sure all Extended Data tables are properly numbered and accounted for

Marlétaz et al: Response to Reviewers

Reviewer: 1

This is a highly original report that will impact the evodevo field substantially and will also raise the bar for investigators conducting future genomics-inspired biological research. This paper stands out for its totality of approaches. And even though the work could be published as several stand-alone papers, the sum of the findings is much greater than the parts. Kudos to the team for working together to generate such an immense and comprehensive dataset. The use of [chordate] linkage group evolution to pinpoint chromosomal breakpoints and to tie this with local chromatin conformational changes that may have given rise to innovations in gene regulation and major evolutionary transitions, is a powerful approach. It is an approach that will surely be used in addressing age-old problems in evodevo over the next years.

We would like to thank the reviewer for the positive appreciation of our work and the comments. We fully agree that the combination of novel technologies and computational approaches opens exciting opportunities to investigate the emergence of traits.

There is so much data in this paper, from the genome assembly to all the functional genomics and transcriptomic data, to the developmental biology experiments. While all the data are presented in a concise manner in the extended data file, those data are quite dense and cannot be fully validated without rerunning the analyses (which I could not do). So I needed to rely on the analyses by the authors (my mea culpa). In most cases, the data look appropriate and the aesthetically pleasing graphical presentations/summaries are appreciated. For non-genomics types, however, I do feel that some of the figure legends should be more explanatory, e.g., the data in Fig. 4A (and ED Fig. 9A) are not intuitive and a clearer explanation in the legends could help a lot.

We have carefully revised all figure legends and expanded them when necessary, including those of Fig. 4A and Extended Data Fig. 9A. Please note that former Extended Data Fig. 9A is Extended Data Fig. 14A in the current version of the manuscript.

The surrogate expression data in Figure 5d is not easy to interpret due to the large amount of GFP signal in the head. Is it possible to improve on that image? And how many embryos were examined in those experiments?

The injected vector, mini-Tol2, has the *midbrain enhancer:egfp* as a positive control of an injection. Thus, successfully injected embryos with the skate or shark *Hox* enhancer vector should show EGFP signal in the midbrain with/without pectoral fin expression. To highlight the anterior pectoral fin expression, we have added an inset of the EGFP reporter signal in the fin of

the F0 animals, as well as a new photo of EGFP expression in the pectoral fin from the skate *Hox*-enhancer stable line at 4 dpf (Fig. 5d).

Regarding the numbers, we found no pectoral-fin positive embryos out of 31 midbrain-EGFP positive embryos from the shark enhancer injection, but identified 5 out of 18 from the skate enhancer injection.

The *Prickle1* ISH in the far anterior pectoral fin of skate (Fig. 5e) is unexpected and the staining is beautiful. Is there also expression in the anterior pelvic fin? Not that it lessens the impact of the paper but their exclusion but were other elasmobranch taxa also included here? Just asking because PCP involvement in anterior wing development is a major finding of this paper.

We observed weak *Prickle1* expression in the anterior pelvic fin and the clasper in addition to the anterior pectoral fin (**Reviewer Fig. 1**). The anterior pelvic fin and clasper also extend laterally and posteriorly during skate development (Maxwell et al., *Anat Rec (Hoboken)*. 2008). Thus, the gained *Prickle1* expression may be also critical for the unique pelvic fin morphology. Importantly, a unique pectoral and pelvic fin morphology has simultaneously evolved during the diversification of batoids, implying the similar/same genetic cascade might have been deployed for paired fin development (Biology of Sharks and Their Relatives, 2nd ed., Carrier, Musick, and Heithaus, 2012). Indeed, another study showed that *Wnt3* or *Hoxa11*, which are critical for the pectoral fin development, are expressed in the anterior tip of the pelvic fins (Barry and Crow, *EvoDevo*, 2017; Nakamura et al. *PNAS*, 2015). Accordingly, we hypothesize that the unique *Prickle1* expression might have driven the pectoral and pelvic fin evolution and diversification among batoid fishes. This hypothesis should be tested by using a variety of batoid species in future studies.

We have added this expanded figure panel into Extended Data Fig. 16 and commented on these findings in the discussion.

Reviewer Figure 1. Expression pattern of *Prickle1* by whole-mount in situ hybridization. Arrows mark expression domains in anterior pectoral and pelvic fins, and in the clasper

In MAIN, p4, line 5: While wing-like appendages are in skates, they define the batoids. That should probably be batoids instead of skates per se.

We thank the reviewer for pointing this out. We have modified the text accordingly

“Homology” is used somewhat loosely in the text. In some places it didn’t appear clear whether it was orthology or homology in the evolutionary sense.

Thanks, we revised the text accordingly. We specify now that the *Hox* clusters in skate and mammals were orthologous (line 287, page 12).

1st line of p 18 – is “both pectoral fins” correct?

Thanks for pointing out this mistake. We have changed the text by “both pectoral and pelvic fins”.

P 18, last line – cownose ray is not a skate, it is another batoid (a myliobatid)

Accordingly, we now specify: “in two different rays and skate species”.

Reviewer: 2

Overall, I find this an interesting and comprehensive study, which contributes to our understanding of the evolution of the specialized morphology of skate fins and the general role of the 3D organization of the genome in the evolution of specific phenotypes.

We would like to thank the reviewer for the positive assessment of our work and the useful suggestions and comments.

I generally find the data presented in this manuscript of high quality and convincing. However, I think it is unfortunate that the authors have chosen to use H3K4me3 HiChIP to analyze tissue-specific chromatin interactions in anterior and posterior regions of developing fins. The data seem very sparse, which makes it difficult to identify changes between the two tissue types. Furthermore, as H3K4me3 HiChIP preferentially detects interactions between regions enriched in H3K4me3, the interactions are not unbiased; it is therefore not clear whether differences in HiChIP interactions reflect differences in H3K4me3 enrichment or in interaction frequency.

We agree with the reviewer that H3K4me3 is not an unbiased technique, and the results may be influenced by differences in promoter activity (H3K4me3 enrichment). Following the reviewer suggestion, we have performed HiC in anterior and posterior regions of the skate pectoral fin. In our opinion, these new data are more robust and clearly improve the manuscript, so we acknowledge the reviewer's suggestion.

From these newly-derived analyses, we reach similar conclusions as for the HiChIP data: there are no major changes in chromatin structure in both regions of the pectoral fin (see also the answers for the following comments). These results are reflected in the new figures 3d-e and Extended Data Fig. 11.

In lines 301-306, the authors write "Differential analysis revealed similar looping patterns between the anterior and the posterior region of the fin (Pearson correlation >0.96, with 9 and 5 interactions statistically enriched in the anterior and posterior fins respectively, Extended Data Fig. 7e, f). We found striking differences in promoter interaction between anterior and posterior fins at specific loci including the HoxA locus (Fig. 3d, Extended Data Fig. 7h, i)."

I find this a bit confusing. Does this mean that the HoxA locus is an example of a region containing 1 of the 9 interactions that are statistically different? In any case, the differences in Figure 3d do not seem striking to me. Similarly, the data presented in Figure 5 shows many empty bins and very limited data depth. Furthermore, the authors present 3 valid replicates – 1 for anterior (A1) and 2 for posterior (P1 and P2) fins and show the correlation in Extended Data Figure 7. Based on these analyses it seems that A2 and P2 correlate better than P1 and P2,

which does not make sense to me. (It could be that I mis-understood the analyses performed here – I find the legends very confusing.)

We apologize for the confusion. Indeed, 2 out of the 9 interactions significantly enriched in the anterior fin involved *Hoxa2* and *Hoxa5* (see Extended Data Table 9). However, based on our new high-resolution Hi-C datasets of the anterior and posterior areas of skate fins, we also think that differences observed in the *HoxA* locus might be caused by differential H3K4me3 binding, rather than differential contacts. We have corrected this paragraph in the new version of the manuscript. Furthermore, in figure 3 we now display the Hi-C profiles of the *HoxA* locus in the anterior and posterior portions of the fin instead of the HiChIP data. Besides, virtual 4C-seq from all *HoxA* promoters derived from the new Hi-C datasets are available in the new Extended Data Fig. 11j. Related to that, and in light of the new Hi-C datasets, the discrepancies between the Pearson and Spearman correlations could be due to outliers in H3K4me3 enrichment (to which Spearman correlation is more robust). We do not observe such discrepancies using our new Hi-C datasets. Therefore we have decided to remove the Pearson correlation from the former Extended Data Fig. 7 (now Extended Data Fig. 10).

I think that there are likely many interesting tissue-specific differences in chromatin interactions between the anterior and the posterior region of the fin for the authors to explore. However, I do not think they can be appreciated from the H3K4me3 HiChIP data, as they are sparse and contain a H3K4me3 bias. It would be much more convincing if the authors could use Hi-C to analyze the differences between anterior and pectoral fins. Alternatively (or in addition), they could consider doing 4C experiments, a multiplexed Capture-C experiment or another high-resolution targeted 3C method (that does not introduce bias for a specific chromatin mark) for their regions of interest. I think that such experiments would improve this manuscript, since I do not think that the conclusions about tissue-specific 3D regulation are supported by the HiChIP data.

As suggested by the reviewer, we have used Hi-C to investigate chromatin interactions in the anterior and posterior skate fin. As previously mentioned, we observe that the few changes observed in HiChIP were not consistent with what we observe now with Hi-C. We have now stressed in the text the main conclusion derived from this new analysis: that 3D chromatin architecture is stable in the different fin regions.

The GEO and GitHub links do not work.

We apologize for the inconvenience. Both links to GEO and GitLab links should be functional now.

Lines 167-170: “Skate micro-chromosomes also show a high degree of interchromosomal contacts compared with macro-chromosomes (Fig. 1d, e), as also found in snakes and other tetrapods, which suggests this is a general feature of vertebrate micro-chromosomes.”

This is indeed clear in the log-transformed observed/expected matrices in Figure 1d. However, I find it surprising that this is not visible in the matrix in Extended Data Figure 4. Can the authors comment on this?

Extended Data Figure 4 (now Extended Data Fig. 5) represents the whole Hi-C dataset binned at 1Mb resolution, while in Fig. 1d all bins from the same chromosomes are merged together making differences in signal more obvious. However, we do observe that the pattern of enhanced micro-chromosome contacts is also visible (albeit fainter) in the old Extended Data Figure 4 (see **Reviewer Fig. 2**). We have now highlighted the microchromosomes in the figure, as well as noted the pattern in the legend.

Reviewer Figure 2: Zoom from the old Extended Data 4 displaying increased interchromosomal interaction between microchromosomes.

Figure 3a. The stripe/bar-structure aligned with the diagonal that is contained within the meta-TAD seems like an unusual feature to me that is potentially of interest. It would be very interesting if the authors could explore this further.

This effect is reminiscent of the structures observed in mitotic cells, in which a stripe signal that is parallel to the diagonal can be noted (Gibcus *et al.*, *Science*, 2018). At the stage in which Hi-C libraries were prepared, the skate fin is in a proliferative state. Therefore, we believe that this signal may be caused by a certain percentage of mitotic cells. These might be interesting findings to pursue in future research, but fall beyond the scope of the current study.

Lines 266-267: “At the sub-megabase scale, the skate genome is organized into Topologically Associated Domains (TADs) with a median size of 800 Kb (Extended Data Fig. 6a, b).”

It could be interesting to explore some further details about the TAD organization. The authors mention later in the manuscript that they identify 1,678 TADs. What proportion of the genome is

covered by TADs? How does this compare to other vertebrates / mammals (based on Hi-C data of similar quality / depth and similar parameters to call TADs)?

This is an important point and we thank the reviewer for bringing it into discussion. We have defined TADs as the genomic space between boundaries. Therefore, using our definition we cannot accurately determine which proportion of the genome is covered by TADs. However, we have observed that TAD structures are pervasive throughout the genome and therefore this definition is instrumental. According to the reviewer's suggestion, we have now compared TADs identified in skate with TADs called in an equivalent manner in mammals and zebrafish Hi-C data with similar depth. However, it is important to note that using universal parameters for TAD calling across species appears problematic, as they fail to identify structures that are observed at plain sight. To overcome this problem, we have adjusted the parameters manually for each species, based on the accuracy to detect the structures from Hi-C maps. With this analysis, we observe that skate TADs fall in an intermediate regime of TAD sizes between zebrafish and mammals, in concordance with the smaller genome size of zebrafish. This new analysis is now displayed in Extended Data Fig. 8 and mentioned accordingly in the text.

5. Figure 3b/Extended Data Figure 6e. The authors show that CTCF motifs are enriched around TAD boundaries. It would be interesting to know what proportion of TADs show CTCF enrichment? How does that compare to other vertebrates / mammals? (Is there reason to think other mechanisms (in addition to cohesin/CTCF-mediated loop extrusion) are involved as well?)

We thank the reviewer for suggesting this valuable addition. We have now dedicated Extended Data Fig. 9 to explore this question. We observed that divergently oriented CTCF motifs are observed in skates in similar proportions compared to other vertebrates (mouse and zebrafish, where CTCF depletion or KOs led to overall loss of TADs). We found at least one “canonical” oriented CTCF in 80% of the boundaries and around 30% of them display the CTCF divergent pattern (similar to 90% and 40% observed in mouse cortical neurons). It is important to note that here we analyze all boundaries identified by the FAN-C algorithm regardless of the boundary score. Therefore, many of these boundaries may fall into the category of “sub-TAD” boundaries, where the divergent CTCF pattern is generally not observed. Besides this, we also found enriched ATAC signal, A compartment and RNA production in boundaries suggesting that additional mechanisms might play a role in insulation. In contrast, only 8% of the boundaries were associated with switches between A and B compartments. We comment on these new findings in the main text.

6. Related to the previous comments, I assume that CTCF and cohesin subunits are conserved in *Leucoraja erinacea*? It would be helpful to mention this.

The Cohesin complex gathers 4 subunits, one of which has two paralogues in vertebrates. We identified all of them in the skate genome: SMC1A (LE04772), SMC3 (LE22652), SCC1 (LE24269) and 2 copies of SCC3 (LE19510 and LE11912). The CTCF gene also has been

steadily identified in the skate (LE18077). This information has now been added to the main text.

7. “Lines 294-297: “To further explore enhancer-promoter interactions, we used chromatin-conformation capture-immunoprecipitation (HiChIP) to associate H3K4me3-rich active promoters with potential regulatory loci in the anterior and posterior region of the developing pectoral fin, where HoxA and HoxD genes are expressed.”

Lines 308-309: “Such interaction changes are consistent with the specific expression of HoxA genes in the anterior portion of the developing pectoral fin.”

Lines 444-449: “Importantly, Hox genes also show distinctive differences between pectoral and pelvic fins (Extended Data Table 5) and between mouse and skate paired appendages (Extended Data Fig. 17). In particular, several 3’ HoxA and HoxD genes are preferentially expressed in the anterior pectoral fin, while 5’ HoxA and HoxD genes are located in the posterior pectoral domains, consistent with previous findings in two different skate species.”

I find this a bit confusing; perhaps it would be helpful if the authors could be more clear and specific about the Hox expression patterns when they first mention the Hox genes.

We apologize for the lack of clarity of this section. We have reordered the text and rewritten some sentences in order to improve readability. In particular, specific information about the Hox expression patterns is now provided when they are first mentioned (lines 307-309, page 13). In addition, a fragment of the results section that was specially confusing and has been completely reworded (lines 502-521, pages 19-20).

8. Lines 335-336: “We find that although the size of TADs is similar between alpha and beta chromosomes, there are notably fewer TADs in the latter (Fig. 3e, f).”

Does this mean that the entire TAD regions have disappeared or did only the boundaries disappear? If the latter, is this driven by specific changes in CTCF motifs?

Those events are really ancient and it is challenging to find arguments for either of the scenarios, as these analyses need to rely on the identification of coding sequences. Nonetheless, we investigated this further (Extended Data Fig. 13D). We only found evidence of 63 TAD boundaries that separate genes in alpha that are together in beta, indicating either the removal of a boundary in beta or the appearance of a boundary in alpha. Conversely, 94 boundaries (31 more) separate genes in beta that could not be found in alpha. In contrast, the gene content of 685 TADs in alpha could not be found in any TAD of beta. We then can conclude that entire TAD disappearances explain the fewer number of TADs in beta. On the one hand, this could be caused by a complete loss of these TADs due to rapid genome reorganizations. Alternatively, the genes that allow the identification of these TADs may have been lost by pseudogenization and erosion. In this second scenario, the disappearance of genes may have relaxed evolutionary constraints on the boundaries of their respective TADs. Thus, these “gene-free” TADs may have fused with neighboring domains. However, discriminating between these two hypothesis leading to complete TAD losses is challenging. We have highlighted these new findings in the text:

Furthermore, we could confirm that the lower number of TADs in beta could not be explained by TAD fusions in beta or converse boundary gains in alpha (**Extended Data Fig. 13d**) [...]. Whether TAD losses in beta were caused by the deletion of whole redundant TADs or the progressive erosion and pseudogenization of their genes it is difficult to ascertain.

9. Figure 4/Extended Data Figure 10. The TAD rearrangement in Figure 4d is convincing; however the examples presented in Extended Data Figure 10 are much less convincing – perhaps this is also something that can be improved with high-quality targeted 3C data.

We chose the rearrangements of former Extended Data Figure 10 (now Extended Data Figure 15) because they involved genes belonging to the terms enriched in the Reactome analysis. We agree with the reviewer that in the former panel 10a, the newly incorporated region into TADs is not huge (still bigger than 50kb, which is the cutoff) and that in panel 10b the breakpoint may correspond with a sub-TAD boundary. Nonetheless we believe that the presence of HiChIP interactions across the predicted synteny breakpoints is reliable and points to a functional effect. Below, we show four additional examples that are more clear and that could be potentially included in the figure (**Reviewer Figure 3**). However, we prefer to maintain the original panels in actual Extended Data Figure 15 because of their relationship with the enrichment analysis.

Reviewer Figure 3: Additional examples of rearranged TADs in the batoid lineage.

10. Line 489. I think the authors mean to refer to Extended Data Figures 18 and 21 (and not 18 and 20).

The figure numbers have been updated accordingly.

11. Figure 5C. There are some “gaps” in some of the ATAC peaks.

We thank the reviewer for pointing to this detail. In this case, the gaps resulted from the preparation of the figure. The figure has now been corrected.

Reviewer: 3

The amount and quality of the genomics data should make this work of broad interest to the evolutionary genomics community, and will help to elevate the status of *Leucoraja erinacea* as an emerging model organism in comparative developmental biology and functional genomics.

As outlined below, though, certain aspects in the authors' analyses need further clarification. Also, the logic for choosing candidate genes and pathways for follow-up experiments is not clear to me, and not all data resulting therefrom – at least in their current form – seem to fully support the conclusions the authors wish to draw.

We would like to thank the reviewer for the positive comments and the suggestions for improvement.

- Genome 'content': in the text, the authors classify 7 chromosomes as 'micro chromosomes' – is the 20Mb cut-off due to a particular convention? Chromosome size distribution in Fig. 2a seems continuous, with LER33 and LER34 right around the 20Mb limit. Hi-C contacts (6 'outliers') and GC % (6 'outliers') would argue for a different classification (Fig. 1d,e), as would LINE content (5 'outliers') or gene density (4 'outliers') in Fig. S2b, c.

We thank the reviewer for pointing out the discrepancy between the main and extended data figure in the size cut-off for the different categories. Our primary criterion for microchromosome assignment were Hi-C contacts, for which the 6 small chromosomes display a clear divergent pattern (Figure 2a). We believe that this likely reflects some biological properties for this subset of chromosomes that is not shared by others. We have now adjusted the plots in Figure 2a and Extended Data Figure 2a, b, c to match this classification. Nevertheless, we would like to emphasize that microchromosomes vary in size across species, and that for instance some of the chromosomes classified here as 'mesochromosomes' are in one-to-one correspondence with chromosomes classified as microchromosomes in *gar*. This suggests that while chromosomal identity is conserved, there are fluctuations in size, GC and gene content across species.

Related to that, can the bi-modal distribution in gene sizes in Fig. S2e (Chipun and Leueri) be explained by shorter genes being overrepresented on micro chromosomes? Likewise, are certain functional classes of genes – e.g. housekeeping genes vs. developmental regulators – overrepresented on micro chromosomes? This would be interesting to know, given the overall denser 3C organization of micro chromosomes and the regulatory interferences potentially resulting therefrom.

We thank the reviewer for raising these interesting points. Following these suggestions, we have now examined the distribution of gene size, which we have added as a panel on Extended Data Figure 2 (panel g). However, this analysis does not suggest an enrichment of smaller gene size

in the micro-chromosomes. We have also examined GO enrichment of the 3 chromosomal categories, and we did not find a particular enrichment of a meaningful broad functional category on micro-chromosomes, with the exception of 'carbohydrate binding' (adjusted p-value $\sim 1e-4$). However, this enrichment only corresponds to 9 genes which seems quite anecdotal.

Lastly, the number of gene losses mentioned in the text (354) and in Fig. S1c (616) do not match.

We thank the reviewer for pointing out this slight discrepancy. We adjusted the text to reflect the numbers displayed in Extended Data Figure 1c.

- Genome evolution: the authors propose that the “skate genome closely resembles that of the most recent jawed vertebrate common ancestor”. Since no real outgroup exists, i.e. a non-gnathostome with a 2R genome, the authors base their argument on chromosome-scale ancestral chordate linkage group (CLG) content, synthetic orthology relationships and gene retention rates. First, in Fig. 2a, it would help the reader if the same color scheme (and size information) for the CLGs would be used as in Fig. 2 of Simakov et al., 2020 NatEcoEvo. Second, is there a way to quantify and make this comparable between species? E.g., total length or % of continuous CLG conservation per chromosome, and compare such metrics for the skate and the 4 species in Simakov et al.? For Fig. 2b, is there another chondrichthyan genome of sufficient quality available, to include in this analysis and thus balance the phylogenetic representation? For Fig. 2c, please include corresponding gene retention rates for at least one representative osteichthyan.

We decided to adjust our color scheme according to the journal instructions to use 'color blind' friendly color schemes and the necessity to provide better contrast for visualization. The representation that we use for Figure 2a is similar to the one depicted in Simakov 2020 (binned CLG content scaled by size). To help compare the genome architectures with previously described species, we now provide the same representation of Figure 2a for gar and chicken (Extended Data Figure 3c and d), as well as for the whitespotted bamboo shark, another chondrichthyan with a chromosome-scale assembly, as suggested by the reviewer (Extended Data Figure 3b). As requested, we also generated a similar synteny plot between skate and shark than that of Figure 2b for the bamboo shark (Extended Data Figure 3a) that indicates the near identical chromosomal content between skate (only skate chromosome 1 and 2 are not in 1:1 correspondence between these two species). The new Extended Data Figure 3b that compares skate and shark chromosomal architecture also shows that most of their chromosomes have a similar CLG content.

We have also included gene retention rates for the gar as Extended Data Figure 2h. As the retention rates are computed per chromosome, the lower value reflects the increased mixing of CLGs in the gar karyotype, and not a higher loss of paralogues.

- 3D chromatin organization: why do the authors think that the anterior and posterior fin have “similar looping patterns”, when over 50% appear to be compartment-specific (Fig. S7e)? Also, the difference between Pearson and Spearman correlation seems to indicate that the former is dominated by outliers (Fig. S7f,g). Maybe the normalization/scaling of the data should be re-considered? Minor comment for line 328: for clarity’s sake, maybe quickly reiterate which results actually suggest the conservation of the two genome organization mechanisms, e.g. presence of A/B compartments and the orientation bias of CTCF sites?

Following the suggestions of Reviewer 2 (see specific comments above), we have now performed Hi-C in replicates in the anterior and posterior areas of the skate fin. These new results are in agreement with our previous HiChIP analysis, in terms of overall similarity in interaction patterns. Some of the changes identified by HiChIP were not visible in Hi-C data suggesting that they may originate from differences in H3K4me3 occupancy rather than bona fide differences in genome folding (see previous comments).

Regarding the Venn diagrams represented in former Extended Data Figure 7, we acknowledge that they might have been misleading. In our experience we find that loop detection algorithms from HiChIP and Hi-C are helpful but not entirely reproducible, as they are sensitive to small differences in coverage and are prone to false negatives. Therefore, assessing the similarity between two samples based solely on the intersection of the loops tends to overestimate the differences. We believe that generating a consensus set of loops, quantifying the reads in the different replicates and performing statistical tests a posteriori is a more robust methodology that matches better with the naked eye observation of the data (pointing to fewer, but clearly differential folding patterns). In our initial submission, we provided Venn diagrams based on loop detection algorithms, which showed a prominent number of tissue-specific interactions. However, these differences were minor when we applied the second approach. Importantly, the analysis of our new Hi-C datasets is consistent with this notion, showing that the differential interactions between the anterior and posterior fin are minor. Thus, we have opted to remove the Venn diagrams from Extended Data Figure 7 (now named Extended Data Figure 10). Given the discrepancy between Spearman and Pearson correlations, which we do not observe in anterior and posterior fin Hi-C datasets (see Extended Data Figure 11f and g), we agree that Pearson correlations in HiChIP can be dominated by outliers, namely due to genomic regions with high H3K4me3 occupancy. Therefore we have also eliminated Pearson correlations from Extended Data Figure 10.

- Choice for follow-up experiments: synteny breaks and HiChIP data identify a total of 180 genes, whose regulatory interactions might have changed due to skate-specific rearrangements. Following a signaling pathway enrichment analysis, the authors first decide to focus on Prickle1, while seemingly ignoring (?) other candidates from the PCP pathway, their apparent top candidate NOTCH signaling, or other previously implicated pathways (e.g. GLI3, see Nakamura et al. 2015) – why? Likewise, the study of a putative Hoxa2 enhancer is partly justified based on the handpicked (?) example of Psip1. Psip1, however, was reported to

activate posterior HoxA genes, via the lncRNA Hottip. Is Hottip present and transcribed in the skate genome? Also, for their *Psip1* argument, the authors probably want to cite the 2017 paper of Pradeepa et al., not the 2012 one. In their current form, these two choices thus appear somewhat arbitrary.

We apologize for this lack of clarity. We have now expanded in the text our rationale for selecting these genes in the text. We performed *in situ* hybridization to screen 8 candidate genes from the 180 that turned up in our genomically-derived list. We also performed whole-mount *in situ* hybridization (WISH) of *Plec*, *Dchs* and *Scrib* which are related to cytoskeleton, limb development and PCP pathway respectively and were found close to rearrangement breakpoints. These are now listed in the Methods section. From the screening candidates, only *Psip1* and *Prickle1* showed clear anterior-specific expression in our in-situs, suggesting they were good follow-up candidates (see the arrows in the **Reviewer Figure 4**). Since *Gli3* was already implicated and had been analyzed previously (Nakamura *et al*, *PNAS*, 2015), we opted to investigate the potential involvement of *Prickle1*, one of the main components in the PCP pathway, in the pectoral fin expansion of skates. The focus of this candidate was further supported by our cell elongation analysis (Extended Data Figure 18), which showed distinct cell morphologies and suggest that the PCP pathway is acting preferentially in the anterior developing fin of skates. We also apologize for the inaccurate citation of Pradeepa et al. 2017, which has now been corrected. Regarding *Hottip*, we could not identify this lncRNA in skates due to lack of sequence conservation and neither did we identify transcription upstream from *Hoxa13* in the RNA-seq datasets from pectoral fins from Nakamura et al. 2015. However, we can not completely rule out the existence of an homologous or analogous lncRNA in skates that interacts with *Psip1*.

Reviewer Figure 4: WISH screening of genes for specific anterior pectoral fin expression. Arrows mark fin-specific staining

- PCP follow-up experiments: why did the authors chose ROCK inhibition, which clearly impacts cellular processes beyond PCP, and why did they administer it systemically, rather than via beads (see Nakamura et al. 2015). To me, 'st.31 + ROCK inh.' embryos shown in Fig. S15 appear closer in overall size and e.g. head width to st.30 control embryos, than st.31. Moreover, ray numbers should also be quantified at st.29, i.e. when treatment started > does ray number not increase after st.29, i.e. upon ROCK inhibition, or are some rays "lost" during the treatment? Geometric morphometric analyses and PCA also seem to suggest that 'st.31 + ROCK inh.' embryos are stuck somewhere on the 'morphogenetic trajectory', going from st.29 to st.31, which could also indicate a general developmental delay upon ROCK inhibition? Lastly, in Fig. s12, the mean axis ratio, or a boxplot, should be included on the scatter plots, and representative images used for the measurements should be provided.

In model organisms, both inhibitors and genetic approaches, such as up- or down-regulation of pathway components by transgenesis, have been widely used to study the function of the PCP pathway. In particular, ROCK inhibitor has been used as a PCP inhibitor in the long history of

developmental biology (for example, see López-Escobar *et al.*, *Development* 2018; Nishimura *et al.*, *Cell* 2012; LaMonica *et al.*, *Dev Biol.* 2009). Although the manipulation of the expression levels of PCP components might have been a preferred approach, transgenic technologies have not yet been established in skates. Therefore, there were no other feasible choices for this type of experimental validation.

Our initial systemic treatment using the ROCK inhibitor revealed a general growth delay in skate embryos. However, the effect is more prominent in the anterior fin, as we could show by quantification and morphometrics analysis, suggesting the PCP pathway could be involved in its growth. To strengthen these results, as per reviewer suggestion, we have now conducted inhibitor beads implantations at the anterior pectoral fin (Extended Data Figure 24). Although this approach would be restricted to the fin tissue, it is worth noting that the expected effect is very local and only surrounding the implanted bead. In contrast to systemic treatments, these experiments are also very challenging because the beads might drop during the treatment due to the skate movement.

Nevertheless, we observed that the ROCK-inhibitor beads affected fin ray patterning locally. These effects were noticeable in the form of fusion or loss of fin rays around the beads. Furthermore, we also observe such effects in some regions of treated embryo fins where we do not observe the presence of beads. The fact that such effects are never observed in control animals, suggest that these might correspond to regions where a bead was implanted but fell off during the treatment. Thus, these new experiments, together with our cell elongation analysis and our systemic treatment experiments, support our initial observations that the PCP pathway regulates the fin growth and ray patterning. This new data is displayed in the already mentioned new supplementary figure (Extended Data Figure 24) and mentioned in the main text.

As also suggested by the reviewer, we have now counted the number of fin rays in the pectoral fin at stage 29. At this stage, the tribasal bones (pro-, meso-, and meta-ptyrgium) are not clearly identified by cartilage staining. Accordingly, we were able to count only the total number of fin rays compared to the ray count in three fin domains as shown in Fig. 5. The total number of fin rays at stage 29 varies from 55 to 62 (average 58.4, $N=7$), which is the similar range of the total number of fin rays at stage 30 and 31 if we would sum the number of fin rays in the three domains (Fig. 5). Thus, the number of fin rays does not significantly increase from stage 29 to 31, but the ray distribution and patterning change during the fin expansion. Accordingly, we conclude that the fin ray number decreases, particularly in the anterior pectoral fin, by the ROCK inhibitor treatment from stage 29 to 30 and 31, due to the retarded growth of the pectoral fin. This data is displayed in a new supplementary figure (Extended Data Figure 19) and mentioned in the main text.

We also revised Extended Data Figure 18, which currently contains representative images of the membrane and nucleus staining and the mean axis ratios.

- Hox follow-up experiments: in Fig. 5a, why are there anterior/orange and blue/posterior highlights in the pectoral fin and the mouse forelimb? If I understand the experiment correctly, the comparative RNA-seq analysis was performed between entire (?) pectoral and pelvic fins of the skate, and fore-/hindlimbs from a mouse data set? In Nakamura et al. and Fig. S17a,d we do find an A/P differentiation of the samples, but what corresponding mouse data set would have been used for this? The entire workflow and data sources here are not properly explained, neither in the main text nor the material and methods section.

We apologize for the confusion. In Fig. 5a the color coding did not indicate the comparison of particular datasets. In the manuscript, we refer to the expression of genes that show an inverted antero-posterior expression pattern in catshark and mouse (e.g. *Gli3*). For this reason, we wanted to remind the reader of the homologous appendage regions between the two skate and mouse. We have opted now to remove the color coding from the Fig. 5a.

Regarding the RNA-seq datasets, we have performed two different analyses. Again, we apologize for the lack of clarity. First, we generated RNA-seq data from pectoral and pelvic fin tissues and compared them with corresponding publicly-available datasets from mouse fore- and hindlimb tissues. We now specifically mention this in lines 461-463, page 18: “we compared our list of differentially expressed genes in skate fins against a list that compares mouse fore- and hindlimb RNA-seq datasets”. We also provide the data sources in the main text and in the data availability section.

Second, we used the available RNA-seq data from Nakamura et al. (*PNAS*, 2015) from anterior and posterior pectoral fins. We integrated these datasets with the ATAC-seq data generated from the same tissues, to assign differential CREs to differential genes. We have provided a more detailed explanation by adding the sentence “To further explore the transcriptional changes associated with the skate pectoral fin, we analyzed publicly-available anterior and posterior pectoral fin RNA-seq data” (lines 490-491, page 19). We also provide the reference and the accession number at the data availability section.

Additionally, Extended Data Table 5 in the Excel file made available to me does not show the differentially expressed genes.

We apologize for this mistake. Extended Data Tables have been checked and corrected accordingly.

More importantly, however, the entire logic for a) using an inducible, highly expressed (?) *Hoxd13a* transgene to study the effect of endogenous *Hoxa2* levels on *Gli3* transcription, and b) testing a putative *Hoxa2* enhancer in zebrafish, appears flawed to me. If present in the genome, skate *Hoxd13* would presumably (see Fig. S17d, *Hoxd10-12*) be expressed posteriorly, i.e. where *Gli3* is expressed. *Hoxa13*, on the other hand, appears to be expressed anteriorly (Fig.

5b, although I could not find a similar trend for it in Fig. S17d?). What could be the reason for this anterior *Hoxa13* pattern, e.g., any clues from the HiChIP data? And why is *Hoxa2* still considered the prime candidate for *Gli3* downregulation, given the known impact of *Hoxa13* on *Gli3* expression? (also, see 'Psp1>Hottip>5'HoxA' link, mentioned above)

We agree with the reviewer that this section of the manuscript was not sufficiently clear. In this case, naming this enhancer *Hoxa2* is not related to a particular functional link with this gene, but due to genomic proximity. In fact, our 3D interaction data suggest a strong co-regulation (and probably redundant functions) of all anterior and some posterior genes, especially *Hoxa13*. Therefore, we believe that this enhancer (and probably others) drive the expression of several *Hox* genes specifically in skates to the anterior region of the pectoral fin, leading to the downregulation of *gli3* expression. We have changed the name of the enhancer to avoid confusion, and added a supplementary figure with contacts based on HiChIP, Hi-C, and virtual 4C (from HiChIP) from the region where the enhancer is located (Extended Data Fig. 31a).

Although other *Hoxd* posterior genes are highly expressed in the posterior region of the fin, *Hoxd13* is not. In Nakamura et al. PNAS, 2015, WISH experiments for several *Hoxd* genes are displayed. While *Hoxd10* and *Hoxd12* display a clear and strong posterior signal, the signal from *Hoxd13* appears to be weaker and more diffuse. Since WISH experiments are not quantitative, we have compared the expression levels of genes from the *Hoxa* and *Hoxd* clusters in both the anterior and posterior regions of the pectoral fin (Extended Data Fig. 31b). This new figure shows that *Hoxd13* is expressed at very low levels in the posterior region. Overall, the data highlights that the levels of *Hox13* paralogs have a bias towards the anterior fin. This bias is primarily caused by the expression pattern of *Hoxa13* that, based on our chromatin interaction data, is likely due to the skate-specific *Hoxa* enhancer.

Hoxa13 does not appear in the differential expression plot of the old Extended Data Fig. 1b7 (now Extended Data Fig 25b) because the difference between the anterior and posterior fin expression is not so prominent (see also new Extended Data Fig 31b).

Given the known functional redundancy between *Hoxd13* and *Hoxa13* proteins (Fromental-Ramain et al. *Development*, 1996; Sheth et al. *Cell Rep*, 2016; Nakamura et al. *Nature*, 2016), and that the *hoxd13a-GR* construct has already been tested in a previous publication (Freitas et al., *Dev Cell*, 2012), we decided to use this construct to test our hypothesis that an increase in Hox protein levels would lead to downregulation of *Gli3*.

Finally, regarding transgenesis assays in zebrafish, trans-species transgenesis assays have been previously performed with success, proving that orthologous enhancers from different vertebrate species can drive similar expression patterns when tested in zebrafish (i.e. see de la Calle et al., *Gen Res* 2005).

For the enhancer test, the current images without close-up of the fin do not allow to see a potential anterior enrichment of reporter expression in the zebrafish fin, and the skate *HoxA2* image appears overexposed (see e.g. fore- and hindbrain signal, or trunk signal).

Moreover, were these F0 embryos and what number of replicates were produced and analyzed?

We added a close-up image of the skate *Hoxa* enhancer-GFP transgenic embryo in Fig. 5

These embryos were derived from the F0. The number of replicates for these transgenic fish were 15 out of 42 mid brain EGFP-positive embryos with the skate *HoxA* enhancer compared to 0 out of 31 mid brain EGFP-positive embryos with shark *HoxA* enhancer. This information is now also included in the caption of Fig. 5.

Furthermore, we have also added a figure from an embryo derived from the stable F1 line, which shows the anterior fin activity of the skate enhancer.

Lastly, given the AER-like scenario proposed for the anterior skate fin in Nakamura et al., how do the authors envision this skate enhancer to function in a zebrafish fin, given that the trans-regulatory environment is not skate-like?

Our data suggests that the skate *HoxA* enhancer (previously referred as *HoxA2*) originated *de novo* in the lineage of elasmobranchs. As this enhancer is active in the native context of the skate pectoral fin formation as well as in the zebrafish pectoral fin, we anticipate that it is likely activated by the conserved tool kit genes that are essential for fin/limb development. This may be evolutionarily more efficient than creating a novel enhancer and trans-regulatory environment from scratch. The identification of such an environment regulating skate *HoxA* enhancer activity will be definitely a target in our future studies.

- Line 85: Fig. 1b, not a?

We have now corrected this mistake.

- Line 345: "...for other mammalian traits" reads like the skate would also be a mammal?

We have now corrected this mistake.

- Line 461: typo > enhancers

We have now corrected this mistake.

- Line 489: Extended Data 21, not 20?

We have now corrected this mistake.

- Please make sure all Extended Data tables are properly numbered and accounted for

We have checked all the tables and their call in the text.

Reviewer Reports on the First Revision:

Referees' comments:

Referee #1 (Remarks to the Author):

This work is a tour-de-force that combines natural history, comparative embryology, functional genomics and extensive computational analysis in order to make evodevo inferences about a longstanding evolutionary question: How did a flattened body plan emerge within the batoid lineage of elasmobranchs. The work uses a bevy of tools to assess how the dynamic state of chromatin and the gene expression patterns within the little skate pectoral fins are aligned with the changes in karyotype and rearrangements that have occurred in the skate genome to give rise to the greatly modified pectoral fins. This paper is so extensive that it could easily have comprised four or five different reports. But when all the data are combined as in this paper, the results and storyline are so much better formulated. The authors have thoughtfully addressed criticisms of the previous submission and have rerun certain experiments. The result is a paper that more clearly shows the power of this multi-pronged approach for addressing an age-old biological question. While the paper still suffers a bit from the sheer amount of data and analyses, the authors have done a good job in distilling the text to emphasize the major biological problem at hand and highlighting the novel findings from their work. This is an amazing story and one worth publication in Nature.

Referee #2 (Remarks to the Author):

The authors have addressed all my concerns.

Referee #3 (Remarks to the Author):

In the revised version of their manuscript the authors have successfully addressed most of my previous concerns. Importantly, now with several sections of the text edited and re-written, the entire workflow and logic of the manuscript appears much easier to follow for me. Moreover, the authors provide additional data and quantifications to strengthen many of their original conclusions. Some of the functional follow-up experiments still appear to suffer from the limited toolkits available in such non-traditional model organisms. However, the authors make the most of this difficult setting and their obtained results – cumulatively – are at the very least in agreement with the conclusions they wish to draw. Moreover, as mentioned in my previous review, the quality and amount of the functional genomics resources alone should make this manuscript of broad interest to the comparative genomics and EvoDevo communities. Accordingly, I am happy to endorse it for publication in Nature.